# Cathepsin L-dependent positive selection shapes clonal composition and functional fitness of CD4⁺ T cells

Elisabetta Petrozziello[1], Amina Sayed[1], João A. Freitas [1], Christine Federle[1], Jelena Nedjic[1], Sarina Ravens [2], Batuhan Akçabozan [1], Anna M. Schulz [3,11], Dietmar Zehn [3], Marc Schmidt-Supprian [4], Reinhard Obst[1], Immo Prinz[2,5], Martijn Verdoes [6], Jan Kisielow [7,12], Thomas Reinheckel [8], Tobias Straub [9], Stephen R. Daley [10,13] & Ludger Klein [1,13] ✉

The physiological significance of thymic positive selection and its reliance on a single stromal cell type, cortical thymic epithelial cells, remain incompletely understood. The lysosomal cysteine protease cathepsin L (CTSL) has been implicated in generating major histocompatibility complex class II-bound peptides in cortical thymic epithelial cells for efficient CD4⁺ T cell differentiation. Here, we addressed the extent and nature of the CD4⁺ T cell repertoire changes associated with CTSL deficiency. In the absence of CTSL, a highly selective loss of T cell receptors resulted in a markedly reduced repertoire diversity. However, a similarly large proportion of nominally 'CTSL-independent' T cell receptors were retained. Clones representative of the second category experienced weaker positive selection signals in the absence of CTSL, which were sufficient for further maturation yet imprinted aberrant responsiveness to agonist stimulation and impaired homeostatic behavior. Together, these findings demonstrate that CTSL is crucial for both shaping full repertoire diversity and optimizing CD4⁺ T cell functionality.

During thymic selection, thymocytes test their T cell receptor (TCR) on self-peptide major histocompatibility complex (pMHC) ligands presented by thymic antigen-presenting cells (APCs). 'Weak' TCR–pMHC interactions promote developmental progression and CD4⁺/CD8⁺ T cell lineage commitment (termed positive selection), whereas 'strong' signals trigger negative selection[1]. Although negative selection eliminates autoreactive T cells[2], how positive selection shapes a 'useful' repertoire remains unclear[3].

Traditionally, positive selection was thought to enforce self-MHC restriction. However, T cells recognizing specific pMHC ligands arise at similar frequencies regardless of the selecting MHC allele[4], and the repertoire's apparent self-MHC restriction may indirectly result from negative selection of overly MHC-reactive TCRs[5]. Increasing evidence

also indicates that positive selection imprints T cell functionality, such as tuning of the inhibitory TCR rheostat CD5 (ref. 6). CD5 levels are thought to reflect the strength of the selecting TCR–pMHC interactions and have been linked to responsiveness to foreign antigens[3,7,8], contribution to primary versus memory responses[9] and differentiation potential into helper T (T_H) cell subsets or regulatory T (T_reg) cells[10–14].

Positive selection relies on a single stromal cell type, cortical thymic epithelial cells (cTECs). This specialized role appears to stem from unique pathways of self-antigen handling and processing, likely generating a partially 'private' pMHC ligandome[1,15]. For MHC class I (MHCI), cTECs express 'thymoproteasomes' containing the β5t subunit, which is absent from other APCs. β5t deficiency profoundly affects thymic development of CD8⁺ T cells[15]. For MHC class II (MHCII), cTECs

use autophagy-associated mechanisms for unconventional endogenous MHCII loading. Disruption of these pathways perturbs CD4[+] T cell selection[16–18]. Mice lacking the thymus-specific serine protease PRSS16 exhibit impaired polyclonal CD4[+] T cell responses and diminished positive selection of some transgenic TCRs[19,20]. Of the pathways implicated in shaping the cTEC pMHC class II (pMHCII) ligandome, the most profound reduction in the thymic CD4[+] T cell population is caused by ablation of cathepsin L (CTSL)[21]. Cathepsins are a family of lysosomal proteases. CTSL is strongly expressed in cTECs but is barely present in other MHCII[+] APCs, whereas cathepsin S exhibits the opposite expression pattern. Both enzymes serve dual functions in the MHCII pathway by degrading the MHCII-associated invariant chain (Ii) and processing antigens to generate peptides for MHCII loading[22]. Importantly, the Ii-degrading function of CTSL alone cannot explain its requirement for efficient CD4[+] T cell selection[23]. Limited resolution and cell numbers have so far precluded a comprehensive characterization of the cTEC pMHCII ligandome beyond relatively abundant peptides[24]. However, analysis of pMHCII ligands in fibroblasts engineered to express CTSL or cathepsin S indicated qualitative and quantitative differences[25], suggesting that CTSL-dependent 'private' pMHCII ligands on cTECs may be crucial for CD4[+] T cell positive selection. However, the impact of CTSL on the diversity and functionality of the CD4[+] T cell repertoire remains unclear.

Here, we analyzed the CD4[+] T cell repertoire selected in the absence of CTSL and characterized the clonal composition and reactivity of residual CD4[+] T cells specific for a prototypical foreign antigen. Using high-resolution repertoire sequencing and re-expression of selected clones in TCR transgenic mice, we found that CTSL shapes CD4[+] T cell selection by promoting full repertoire diversity and fine-tuning CD4[+] T cell functionality.

## Results

### CTSL deficiency impairs positive selection of CD4[+] T cells

Full genomic deletion of *Ctsl* in *Ctsl*[−/−] mice has pleiotropic effects, most prominently alopecia and epithelial hyperplasia in the skin[21], reflecting 'nonimmune' functions. To selectively delete *Ctsl* in TECs, we generated *Ctsl*[ΔTEC] mice, which carry a conditional *Ctsl* allele and a *Foxn1-cre* transgene (Supplementary Fig. 1a,b). Compared to *Ctsl*[+/+] mice, *Ctsl*[ΔTEC] mice showed a reduction in CD4 single-positive (CD4SP) thymocytes, as described in *Ctsl*[−/−] mice[21] (Fig. 1a,b and Extended Data Fig. 1a). The phenotypic segregation of CD4SP thymocytes into three consecutive maturation stages (CD69[+]MHCI[−] semimature, CD69[+]MHCI[+] mature 1 and CD69[−]MHCI[+] mature 2; SM, M1 and M2, respectively)[26] was largely preserved, although there was a reduction in the most mature M2 stage (Extended Data Fig. 1b). The proportion of Foxp3[+] T$_{reg}$ cells among CD4SP thymocytes was unchanged; however, there was a trend toward an increased proportion of CD73[+]CCR7[−] reimmigrants from the periphery (Extended Data Fig. 1c,d). Peripheral CD4[+] T cell populations were diminished and contained more 'memory-like' Foxp3[−]CD44[+]CD62L[−] cells and Foxp3[+] T$_{reg}$ cells (Fig. 1c and Extended Data Fig. 1e–g). CD8[+] T cells developed in normal proportions, and the thymic architecture was indistinguishable from *Ctsl*[+/+] mice (Fig. 1a,b and Supplementary Fig. 1e). *Ctsl*[ΔTEC] mice did not show skin defects (Supplementary Fig. 1c,d).

Total MHCII on cTECs remained unchanged (Fig. 1d). However, the fraction of pMHCII ligands on *Ctsl*[ΔTEC] cTECs that consisted of complexes with the invariant chain-derived peptide CLIP, as detected using the monoclonal antibody (mAb) 15G4, was increased, and, conversely, non-CLIP pMHCII ligands, as indicated by the mAb BP107, were reduced (Fig. 1e,f). Nevertheless, compared to *H2-Ab1*[−/−] (hereafter *MHCII*[−/−]) cTECs, non-CLIP pMHCII ligands still constituted a major fraction of the pMHCII ligandome in *Ctsl*[ΔTEC] cTECs (Fig. 1f). For instance, cTECs from *Ctsl*[ΔTEC] C57BL/6 × BALB/c F1 mice presented substantial amounts of the 'frequent' non-CLIP ligand I-A[b]:Eα$_{52–68}$, recognized by the mAb Y-Ae, albeit moderately less than *Ctsl*[+/+] control mice (Fig. 1g). These

effects were seen in cTECs but not medullary TECs (mTECs; Fig. 1e–g and Extended Data Fig. 1h,i), consistent with the differential expression of CTSL between cTECs and mTECs.

The diminished CD4SP thymocyte population in *Ctsl*[−/−] mice has been suggested to result, at least in part, from positive selection of an altered TCR repertoire that is hypersusceptible to negative selection[23]. To test whether interference with negative selection by hematopoietic APCs 'rescued' the CD4SP compartment, we reconstituted *Ctsl*[+/+], *Ctsl*[ΔTEC] or *Ctsl*[−/−] mice with *H2-Ab1*[+/+] (hereafter *MHCII*[+/+]) or *MHCII*[−/−] bone marrow (BM). *MHCII*[−/−] → *Ctsl*[+/+] chimeras harbored a significantly increased percentage of CD4SP thymocytes compared to *MHCII*[+/+] → *Ctsl*[+/+] controls (Fig. 1h), reflecting diminished negative selection[27]. By contrast, the CD4SP compartment of *MHCII*[−/−] → *Ctsl*[ΔTEC] or *MHCII*[−/−] → *Ctsl*[−/−] chimeras was not, or only marginally, increased compared to the respective *MHCII*[+/+] BM controls (Fig. 1h and Extended Data Fig. 1j). To interfere with negative selection by mTECs, we generated *Ctsl*[+/+], *Ctsl*[ΔTEC] and *Ctsl*[−/−] mice carrying a transgene (*Ciita*[kd]) that mediates tissue-specific knockdown of C2TA, a transcription factor that controls multiple MHCII pathway components, leading to reduced MHCII expression on mTECs[28]. The *Ciita*[kd] transgene resulted in a significantly increased CD4SP compartment in *Ctsl*[+/+] mice[28] but did not 'rescue' the CD4SP compartment in *Ctsl*[ΔTEC] or *Ctsl*[−/−] mice (Fig. 1i and Extended Data Fig. 1k).

The proportion of TCRβ[+]CD69[+] cells among CD4[+]CD8[+] (double-positive; DP) thymocytes (representing signal-selection intermediates) was reported to be normal in *Ctsl*[−/−] mice[23]; however, TCRβ[+]CD69[+] DP cells also include CD8[+] T cell lineage selection intermediates that engage pMHCI ligands. To exclude these, we generated MHCI-deficient *B2m*[−/−]*Ctsl*[ΔTEC] mice, which recapitulated the reduced CD4SP compartment associated with CTSL deficiency (Extended Data Fig. 1l). In these mice, where positively selecting interactions could be exclusively attributed to pMHCII ligands, the proportion of 'signaled' TCRβ[+]CD69[+] DP cells was reduced to about half that of *B2m*[−/−]*Ctsl*[+/+] controls (Fig. 1j). CD5 expression on these DP cells (Extended Data Fig. 1m) and bulk *Ctsl*[ΔTEC] CD4SP cells was lower than on their counterparts in *Ctsl*[+/+] mice (Fig. 1k), suggesting that positive selection occurred through relatively weak interactions or nonselection of 'natural' CD5[hi] clones. Together, these findings indicate that the contraction of the CD4[+] T cell compartment in CTSL-deficient mice was not secondary to selection of an altered repertoire that was overly susceptible to negative selection but most likely reflected a bona fide numerical constraint in positive selection as a consequence of an altered cTEC pMHCII ligandome.

### CTSL deficiency causes 'clonal holes' and 'newcomers'

We next addressed whether the bottleneck in positive selection in *Ctsl*[ΔTEC] mice was TCR selective, leading to the disappearance of some clones while allowing others to persist within the repertoire. Across seven transgenic MHCII-restricted TCRs with diverse antigen specificities (*OT-II* (chicken ovalbumin), *Dep* (human C-reactive protein), *AND* and *AD10* (pigeon cytochrome *c*), *LLO56* and *LLO118* (*Listeria monocytogenes* listeriolysin O; LLO) and *PLP1* (myelin proteolipid protein)), all exhibited a profound blockade in the emergence of CD4SP thymocytes in *Ctsl*[ΔTEC] mice, whereas the MHCI-restricted *OT-I* TCR was efficiently selected (Fig. 2a,b and Extended Data Fig. 2a–e). For each MHCII-restricted TCR transgene, CD5 expression on DP thymocytes, which is upregulated concomitant with positive selection, was substantially reduced in *Ctsl*[ΔTEC] mice (Fig. 2c and Extended Data Fig. 2b), suggesting that these TCRs did not, or did not sufficiently, interact with pMHCII ligands to elicit positive selection. When normally selected in *Ctsl*[+/+] mice, *OT-II*, *AND*, *Dep* or *AD10* CD4SP thymocytes each displayed distinct CD5 levels, which varied widely between these clones and spanned the entire range of CD5 expression observed in polyclonal CD4SP cells (Fig. 2d). Thus, nonselection in *Ctsl*[ΔTEC] mice was not tied to specific CD5 characteristics and, by inference, was not confined to a particular window in the affinity range of positively selecting TCR–pMHC interactions.

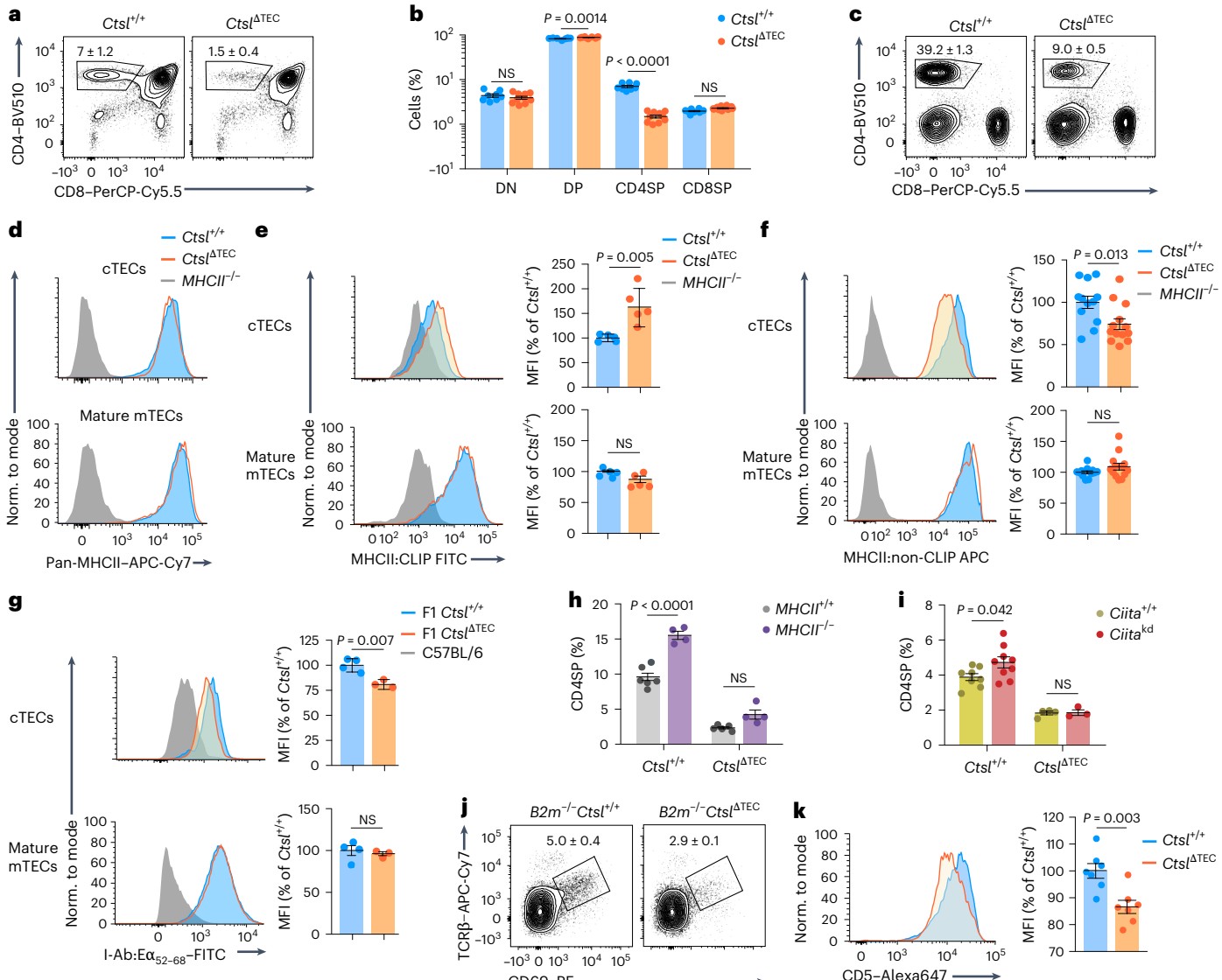

**Fig. 1 | CTSL deficiency alters the cTEC pMHCII ligandome and impairs CD4⁺ T cell positive selection. a**, Representative flow cytometry plots of thymocyte subsets in $Ctsl^{+/+}$ ($n = 8$) and $Ctsl^{\Delta TEC}$ mice ($n = 10$); frequency ± s.e.m. of CD4SP thymocytes is indicated. **b**, Percentages ± s.e.m. of thymocyte subsets as in **a**; NS, not significant; DN, double negative. **c**, Representative flow plots of lymph node cells from $Ctsl^{+/+}$ ($n = 3$) and $Ctsl^{\Delta TEC}$ ($n = 3$) mice; frequency ± s.e.m. of CD4⁺ T cells is indicated. **d**, Representative flow cytometry plots of MHCII expression on CD45⁻EpCAM⁺Ly51⁺ cTECs or CD45⁻EpCAM⁺Ly51⁻CD80⁺ mature mTECs from $Ctsl^{+/+}$ and $Ctsl^{\Delta TEC}$ mice ($n = 20$ each) and $MHCII^{-/-}$ mice ($n = 3$ and 2) as background. **e**, Representative flow plots and mean fluorescence intensity (MFI) ± s.e.m. relative to $Ctsl^{+/+}$ cells of MHCII:CLIP on cTECs or mature mTECs from $Ctsl^{+/+}$ ($n = 5$) and $Ctsl^{\Delta TEC}$ mice ($n = 5$) and $MHCII^{-/-}$ mice ($n = 2$) as background. **f**, Representative flow plots and MFI ± s.e.m. relative to $Ctsl^{+/+}$ cells of MHCII:non-CLIP on cTECs or mature mTECs from $Ctsl^{+/+}$ ($n = 12$) and $Ctsl^{\Delta TEC}$ mice ($n = 13$) and $MHCII^{-/-}$ mice as

background. **g**, Representative flow cytometry plots and MFI ± s.e.m. relative to $Ctsl^{+/+}$ cells of I-Aᵇ:Eα₅₂₋₆₈ on cTECs or mature mTECs from $Ctsl^{+/+}$ ($n = 4$) and $Ctsl^{\Delta TEC}$ ($n = 3$) mice on the C57BL/6 × BALB/c F1 background and C57BL/6 (Eα⁻) mice as background controls. **h**, CD4SP thymocyte percentages ± s.e.m. in $Ctsl^{+/+}$ mice ($n = 6$ or 4) or $Ctsl^{\Delta TEC}$ mice ($n = 5$ or 4) reconstituted with BM from $MHCII^{+/+}$ or $MHCII^{-/-}$ mice. **i**, CD4SP thymocyte percentages ± s.e.m. in $Ctsl^{+/+}$ mice ($n = 8$ or 9) or $Ctsl^{\Delta TEC}$ mice ($n = 4$ or 3) on a wild-type or $Ciita^{kd}$ transgenic background. **j**, Representative flow cytometry plots of TCRβ and CD69 surface expression on DP cells from $B2m^{-/-}Ctsl^{+/+}$ ($n = 9$) and $B2m^{-/-}Ctsl^{\Delta TEC}$ mice ($n = 13$); frequency ± s.e.m. of TCRβ^int CD69⁺ cells is shown ($P < 0.001$). **k**, Representative flow cytometry analysis and MFI ± s.e.m. relative to $Ctsl^{+/+}$ cells of CD5 expression on CD4SP thymocytes from $Ctsl^{+/+}$ and $Ctsl^{\Delta TEC}$ mice ($n = 7$ each). $P$ values in **b** were determined by two-way analysis of variance (ANOVA) and Sidak's test for multiple comparisons and in **e**–**k** by Student's two-tailed $t$-test.

To globally characterize TCR repertoire perturbations caused by CTSL deficiency, we crossed $Ctsl^{\Delta TEC}$ mice with mice expressing a transgenic TCRβ chain (hereafter $Tcrb^{Fixed}$), enabling high-throughput sequencing of variable TCRα chains paired with the 'fixed' β-chain. $Tcrb^{Fixed}Ctsl^{\Delta TEC}$ mice had fewer CD4SP thymocytes and peripheral CD4⁺ T cells (Fig. 3a and Extended Data Fig. 3a–c), with reduced CD5 expression on CD4SP cells (Fig. 3b) compared to $Tcrb^{Fixed}Ctsl^{+/+}$ mice. $Tcra$ sequencing was performed on CD4SP cells in the most mature M2 stage[26], ensuring that the repertoire was fully shaped by thymic selection. Sampling depth approached saturation for

both genotypes (Fig. 3c). Repertoire diversity, as assessed using the Shannon index, was significantly lower in $Tcrb^{Fixed}Ctsl^{\Delta TEC}$ mice than in $Tcrb^{Fixed}Ctsl^{+/+}$ mice (Fig. 3d). Based on the Morisita–Horn index, repertoires were highly stereotypical between genotype-matched replicates but varied significantly between genotypes (Fig. 3e). Of 9,626 'recurrent' TCRs (defined as TCRs found in three or more of all six samples (three $Tcrb^{Fixed}Ctsl^{\Delta TEC}$ and three $Tcrb^{Fixed}Ctsl^{+/+}$)), 4,765 were shared between the two genotypes (Fig. 3f). Almost half (3,848 of 8,613) of all recurrent TCRs in the $Tcrb^{Fixed}Ctsl^{+/+}$ repertoire were entirely absent from the $Tcrb^{Fixed}Ctsl^{\Delta TEC}$ repertoire (Fig. 3f), and

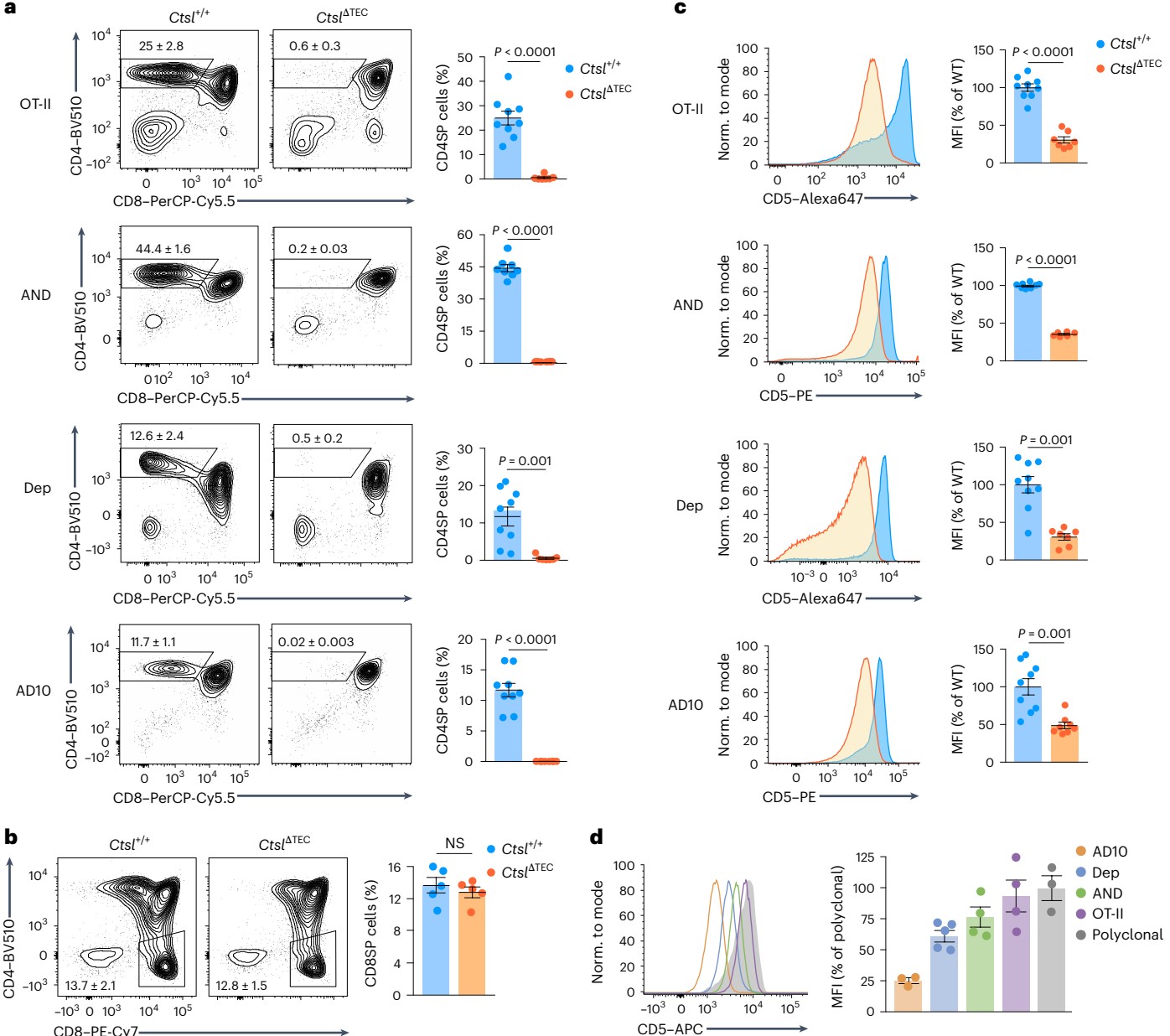

**Fig. 2 | CTSL is essential for positive selection of multiple MHCII-restricted transgenic TCRs. a**, Representative flow cytometry plots of thymocyte subsets and percentages ± s.e.m. of CD4SP cells in *Ctsl*[+/+] and *Ctsl*[ΔTEC] mice reconstituted with BM from *Rag1*[−/−]*OT-II* (*Ctsl*[+/+], n = 9; *Ctsl*[ΔTEC], n = 7), *Rag1*[−/−]*AND* (*Ctsl*[+/+], n = 8; *Ctsl*[ΔTEC], n = 6), *Rag1*[−/−]*Dep* (*Ctsl*[+/+], n = 9; *Ctsl*[ΔTEC], n = 7) or *Rag1*[−/−]*AD10* (*Ctsl*[+/+], n = 9; *Ctsl*[ΔTEC], n = 8) TCR transgenic donors. **b**, Representative flow cytometry plots of the thymus and percentages ± s.e.m. of CD8SP cells in *Ctsl*[+/+] and *Ctsl*[ΔTEC] mice reconstituted with BM from *OT-I*[Tg]*Rag1*[−/−] donors

(n = 5 each). **c**, Representative flow cytometry plots and MFI ± s.e.m. of CD5 expression in DP thymocytes from BM chimeras as in **a**, relative to cells selected in *Ctsl*[+/+] chimeras; WT, wild-type. **d**, Representative flow cytometry plots and MFI ± s.e.m. of CD5 expression in CD4SP thymocytes from *Rag1*[−/−]*Ctsl*[+/+]*AD10* (n = 3), *Rag1*[−/−]*Ctsl*[+/+]*Dep* (n = 5), *Rag1*[−/−]*Ctsl*[+/+]*AND* (n = 4) and *Rag1*[−/−]*Ctsl*[+/+]*OT-II* (n = 4) TCR transgenic mice relative to polyclonal CD4SP thymocytes (n = 3). *P* values in **a** and **b** were determined by Student's two-tailed *t*-test and in **c** by Welch's two-tailed *t*-test.

these 'CTSL-dependent' clones disproportionately contributed to the diversity of the 'normal' repertoire (Fig. 3g). Conversely, about 20% (1,013 of 5,778) of clones in the *Tcrb*[Fixed]*Ctsl*[ΔTEC] repertoire were not found in the 'normal' *Tcrb*[Fixed]*Ctsl*[+/+] repertoire (Fig. 3f). These 'newcomer' TCRs displayed a bias toward more distal TCRα variable (V) and joining (J) elements and increased nucleotide additions or deletions at the V–J joint (Extended Data Fig. 3d–f), suggesting a selection bias for unusual TCR features. Thus, the loss of TCRs in the absence of CTSL was highly selective, affecting roughly half of the 'normal' TCR repertoire, whereas a similarly large array of seemingly CTSL-independent TCRs was retained.

## CTSL deficiency blunts CD4⁺ T cell responses

Having identified 'clonal holes' with various TCR transgenes and in the *Tcrb*[Fixed] repertoire, we asked whether CTSL deficiency caused corresponding 'antigenic holes' in a fully polyclonal setting. MHCII tetramer (Tet) staining revealed a marked reduction (or near absence) of cells recognizing the epitopes 2W, human invariant chain residues 277–285 (huCLIP) and $PLP_{11-19}$ in *Ctsl*[ΔTEC] mice (Fig. 4a,b). However, numbers of $LLO_{190-201}$:I-A[b]-specific CD4⁺ T cells were comparable between *Ctsl*[ΔTEC] and *Ctsl*[+/+] mice, both among peripheral CD4⁺ T cells and thymic CD4SP cells (Fig. 4c and Extended Data Fig. 4). Given the approximately four-fold reduction in total CD4⁺ T cell counts in *Ctsl*[ΔTEC] mice (Extended Data

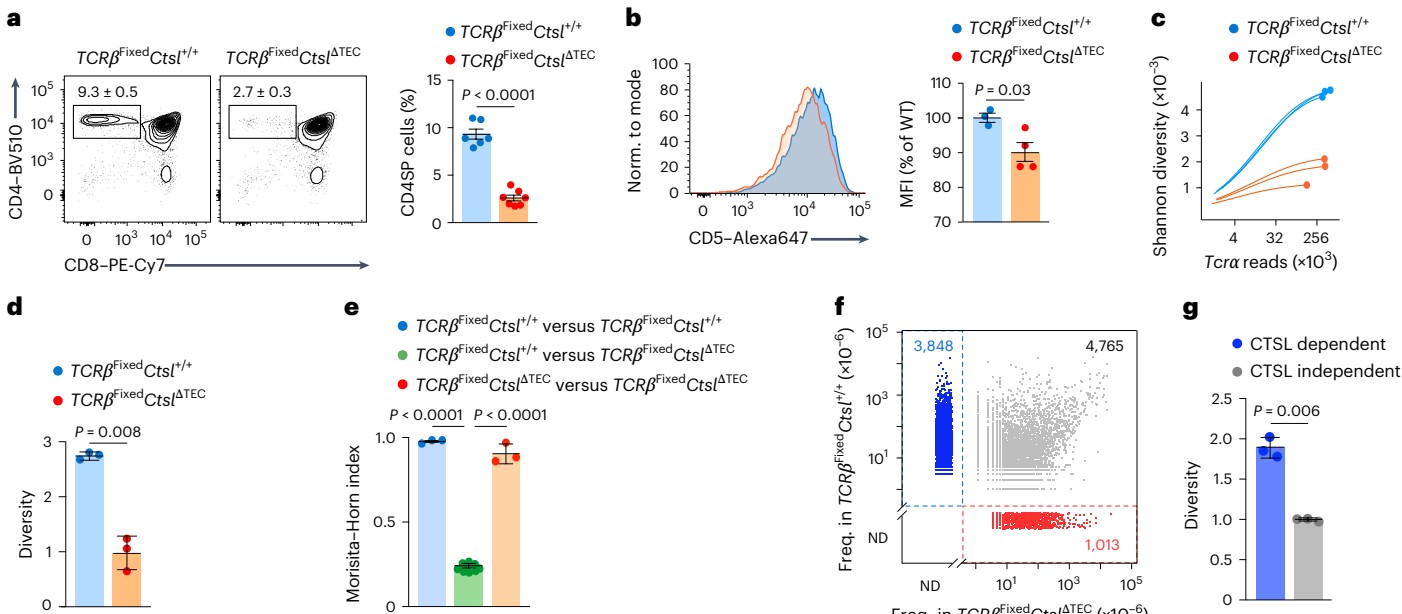

**Fig. 3 | CTSL deficiency results in nonselection of ~50% of TCR clonotypes and enables emergence of 'newcomer' TCRs. a**, Representative flow cytometry of thymocyte subsets and frequency ± s.e.m. of CD4SP thymocytes in $Tcrb^{Fixed}Ctsl^{+/+}$ ($n = 6$) and $Tcrb^{Fixed}Ctsl^{\Delta TEC}$ mice ($n = 7$). **b**, Representative flow cytometry analysis and MFI ± s.e.m. of CD5 expression on CD4SP cells from $Tcrb^{Fixed}Ctsl^{+/+}$ ($n = 3$) and $Tcrb^{Fixed}Ctsl^{\Delta TEC}$ mice ($n = 4$), relative to $Tcrb^{Fixed}Ctsl^{+/+}$ samples. **c**, Analysis of sequencing depth by simulation of Shannon diversity as a function of the number of $Tcra$ reads in bulk TCR-sequencing datasets generated with CD4$^+$CD8α$^-$CD69$^-$MHCI$^+$CD25$^-$FoxP3$^-$ M2 CD4SP cells from $Tcrb^{Fixed}Ctsl^{+/+}$ ($n = 3$ with cells pooled from two to three mice) and $Tcrb^{Fixed}Ctsl^{\Delta TEC}$ mice ($n = 3$ with cells pooled from two to three mice). All mice were on a $Tcra^{+/-}Foxp3^{GFP}$ background to exclude dual TCR expression and enable exclusion of Foxp3$^+$ cells. **d**, Shannon diversity analysis (mean ± s.e.m.) of bulk TCR-sequencing datasets as in **c**. **e**, Repertoire similarity comparison by Morisita–Horn index (mean ± s.e.m.) for all pairwise comparisons between TCRα datasets from $Tcrb^{Fixed}Ctsl^{+/+}$ and $Tcrb^{Fixed}Ctsl^{\Delta TEC}$ mice as in **c** ($n = 3$ for $Tcrb^{Fixed}Ctsl^{+/+}$ versus $Tcrb^{Fixed}Ctsl^{+/+}$; $n = 9$ for $Tcrb^{Fixed}Ctsl^{+/+}$ versus $Tcrb^{Fixed}Ctsl^{\Delta TEC}$; $n = 3$ for $Tcrb^{Fixed}Ctsl^{\Delta TEC}$ versus $Tcrb^{Fixed}Ctsl^{\Delta TEC}$). **f**, Scatter plot of the mean frequency of 'recurrent' TCRs in the $Tcrb^{Fixed}Ctsl^{+/+}$ versus $Tcrb^{Fixed}Ctsl^{\Delta TEC}$ repertoire as in **c**. Recurrent TCRs ($n = 9,626$) were defined as clonotypes found in three or more of all six samples regardless of genotype. TCRs exclusively found in $Ctsl^{+/+}$ samples are highlighted in blue (CTSL-dependent TCRs; $n = 3,848$), and TCRs exclusively found $Ctsl^{\Delta TEC}$ samples are highlighted in red ('newcomer TCRs'; $n = 1,013$); ND, not detected. **g**, Shannon diversity analysis (mean ± s.e.m.) of the CTSL-dependent (blue in **f**) or CTSL-independent (gray in **f**) subrepertoires within the $Tcrb^{Fixed}Ctsl^{+/+}$ repertoire. $P$ values in **a**, **b**, **d** and **g** were determined by Student's two-tailed $t$-test and in **e** by one-way ANOVA with a Tukey's test for multiple comparisons.

Fig. 1e), the frequency of LLO-specific cells was thus even increased by a corresponding factor. To assess the functionality of these cells, we immunized mice with LLO peptide in adjuvant. At day 7 after challenge, ~1 × 10$^4$ LLO-Tet$^+$ T cells were detectable in the draining lymph nodes of $Ctsl^{+/+}$ mice (Fig. 4d), reflecting a >20-fold expansion following antigen exposure. By contrast, despite higher precursor frequencies, LLO-Tet$^+$ cells were approximately tenfold less abundant in $Ctsl^{\Delta TEC}$ mice (Fig. 4d).

We performed adoptive co-transfer experiments to determine whether the diminished expansion of LLO-Tet$^+$ cells in $Ctsl^{\Delta TEC}$ mice reflected a cell-intrinsic defect or an indirect consequence of lymphopenia. Bulk M2 thymocytes from CD45.2 $Ctsl^{\Delta TEC}$ and CD45.1 $Ctsl^{+/+}$ donors were co-transferred at a 1:1 ratio (thereby establishing a ~4:1 ratio of $Ctsl^{\Delta TEC}$ to $Ctsl^{+/+}$ LLO-Tet$^+$ cells in the input population) into CD45.1/CD45.2 $Ctsl^{+/+}$ recipients that had been intravenously (i.v.) immunized with LLO peptide plus polyinosinic–polycytidylic acid (poly(I:C)) 6 h before. At day 7 after challenge, $Ctsl^{+/+}$CD45.1$^+$ donor-derived cells accounted for 2.6 ± 0.3% of splenic LLO-Tet$^+$ cells, whereas $Ctsl^{\Delta TEC}$CD45.2$^+$ donor-derived cells contributed only marginally (0.006 ± 0.004%; Fig. 4e). This corresponded to a ratio of $Ctsl^{\Delta TEC}$ to $Ctsl^{+/+}$ LLO-Tet$^+$ cells of ~1:400 (Fig. 4f), indicating a marked competitive disadvantage in antigen-driven expansion and/or defective homeostatic maintenance of cells selected in $Ctsl^{\Delta TEC}$ donors. Thus, positive selection in $Ctsl^{\Delta TEC}$ mice not only created 'antigenic gaps' but also resulted in impaired expansion and/or persistence of retained cells following antigen encounter.

## Nonselection affects clones across the CD5 range

Polyclonal 'natural' CD5$^{hi}$ clones have been reported to respond more robustly to immunization than CD5$^{lo}$ clones[8], suggesting a direct link between the modalities of positive selection and responsiveness to foreign antigens[3,8]. To assess whether the nonselection of CTSL-dependent clones in the $Ctsl^{\Delta TEC}$ thymus correlated with their CD5 level, we generated TCR inventories from sorted $Tcrb^{Fixed}Ctsl^{+/+}$ CD4SP cells at both extremes of the CD5 spectrum (Fig. 5a). The TCR compositions were remarkably stereotypic within the four replicates of CD5$^{lo}$ or CD5$^{hi}$ CD4SP cells, respectively, yet highly distinct between the two groups, as evidenced by Morisita–Horn comparisons (Fig. 5b). This supports the notion that partitioning of a given clone into the CD5$^{lo}$ or CD5$^{hi}$ subset of the CD4$^+$ T cell compartment is not stochastic but specified by TCR identity. We classified TCRs found in three or more of four CD5$^{lo}$ samples and absent from all four CD5$^{hi}$ datasets as 'natural CD5$^{lo}$ TCRs' and those exhibiting a reciprocal pattern as 'natural CD5$^{hi}$ TCRs'. Cross-comparison with the recurrent TCRs in our previously established $Tcrb^{Fixed}Ctsl^{+/+}$ versus $Tcrb^{Fixed}Ctsl^{\Delta TEC}$ datasets revealed that 70% of the natural CD5$^{lo}$ TCR clones and 43.5% of the natural CD5$^{hi}$ TCR clones were not selected in the absence of CTSL compared to a loss of 44.7% across all TCRs (Fig. 5c). Thus, at the global repertoire level, nonselection in the absence of CTSL was more pronounced among the natural CD5$^{lo}$ subrepertoire, yet affected clones across the entire spectrum of natural CD5 expression.

We next assessed whether the diminished LLO-specific response in $Ctsl^{\Delta TEC}$ mice reflected a CTSL dependency of 'good-responder' TCR clones within the 'normal' repertoire. To this end, we i.v. immunized $Tcrb^{Fixed}Ctsl^{+/+}$ mice with LLO and performed $Tcra$ sequencing on the expanded LLO-Tet$^+$ population at day 7 after challenge, identifying the top ten expanded clonotypes (Fig. 5d). These clonotypes were cross-referenced with our previously established inventories of

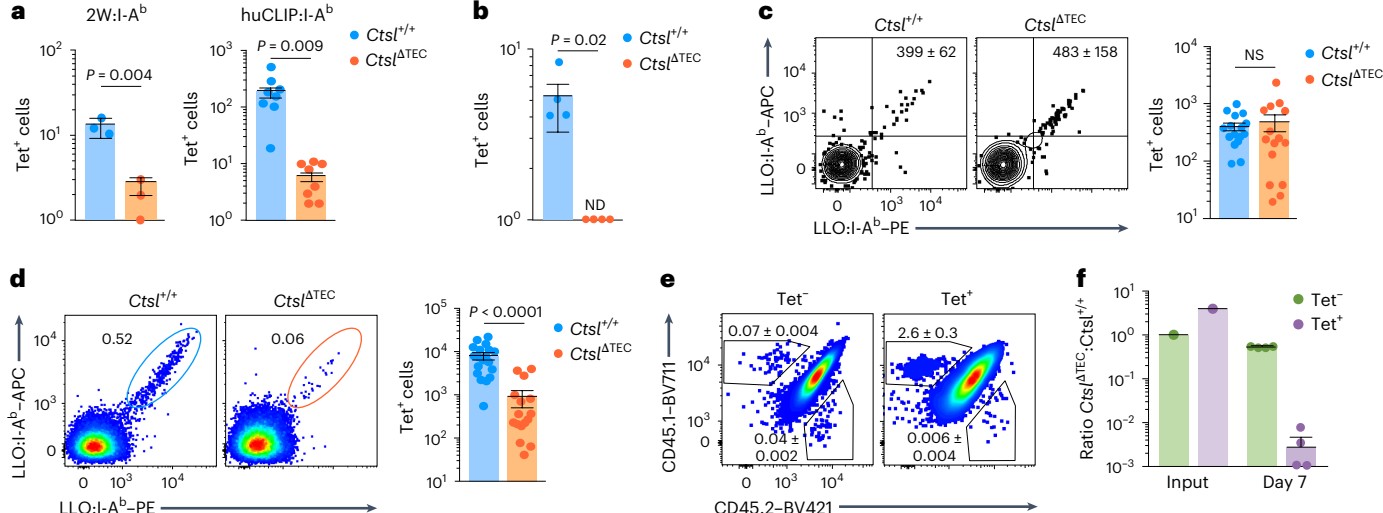

**Fig. 4 | CTSL deficiency imparts 'antigenic holes' in the repertoire and blunts CD4⁺ T cell responsiveness to immunization. a**, Total number ± s.e.m. of 2W- or huCLIP-specific CD4⁺ T cells in the naive repertoire of $Ctsl^{+/+}$ ($n = 3$ for 2W; $n = 8$ for huCLIP) and $Ctsl^{\Delta TEC}$ mice ($n = 3$ for 2W; $n = 8$ for huCLIP), quantified by flow cytometric analysis after magnetic enrichment from pooled spleen and lymph node (LN) cells using 2W:I-A^b and huCLIP:I-A^b Tets, respectively. **b**, Total number ± s.e.m. of PLP$_{11-19}$-specific CD4⁺ T cells in the naive repertoire of $Ctsl^{+/+}Plp1^{-/-}$ ($n = 4$) and $Ctsl^{\Delta TEC}Plp1^{-/-}$ mice ($n = 4$), quantified as in **a** using a PLP$_{11-19}$:I-A^b Tet. **c**, Representative flow cytometry plots of LLO$_{190-201}$-specific CD4⁺ T cells gated on CD11b⁻CD11c⁻B220⁻F4/80⁻CD4⁺ T cells after magnetic enrichment from pooled spleen and lymph node cells using the LLO Tet and total number ± s.e.m. of LLO-Tet⁺ cells quantified as in **a** in the naive repertoire of $Ctsl^{+/+}$ ($n = 16$) and $Ctsl^{\Delta TEC}$ mice ($n = 15$). **d**, Representative flow cytometry plots gated on activated CD11b⁻CD11c⁻B220⁻F4/80⁻CD44⁺CD4⁺ T cells without magnetic enrichment and total number ± s.e.m. of LLO$_{190-201}$-specific CD4⁺ T cells in pooled spleen and lymph node cells of $Ctsl^{+/+}$ ($n = 19$) and $Ctsl^{\Delta TEC}$ mice ($n = 16$) at day 7 after subcutaneous peptide immunization with adjuvant. **e**, Representative flow cytometry plot of CD45.1 versus CD45.2 on LLO-Tet⁻ and LLO-Tet⁺ spleen cells in CD45.1/CD45.2 wild-type recipients ($n = 4$) of a 1:1 mixture of bulk M2SP cells from CD45.1 $Ctsl^{+/+}$ and CD45.2 $Ctsl^{\Delta TEC}$ donors at day 7 after i.v. challenge with LLO peptide plus poly(I:C) pregated on activated CD11b⁻CD11c⁻B220⁻F4/80⁻CD44⁺CD4⁺ T cells as in **d**. **f**, Ratio between CD45.2⁺ $Ctsl^{\Delta TEC}$ and CD45.1⁺ $Ctsl^{+/+}$ donor-derived cells among LLO-Tet⁻ and LLO-Tet⁺ in the input population (same 1:1 M2 CD4SP cell suspension administered to $n = 4$ recipient mice) and at day 7 after immunization (mean ± s.e.m.), as in **e**. Data in **e** and **f** are representative of two experiments. $P$ values in **a**–**d** were determined by Student's two-tailed $t$-test.

CTSL-dependent and CTSL-independent TCRs in the naive repertoire of $Tcrb^{Fixed}Ctsl^{+/+}$ mice, revealing that five TCRs could be assigned to the CTSL-dependent category and five to the CTSL-independent category (Fig. 5d). Thus, a significant proportion of 'good-responder' TCRs were retained in the $Ctsl^{\Delta TEC}$ repertoire, suggesting that the blunted response of LLO-Tet⁺ cells in $Ctsl^{\Delta TEC}$ mice could not be explained solely by the physical absence of all such clones.

### CTSL specifies selection signals in CTSL-independent clones

To assess whether and which LLO-Tet⁺ clonotypes were retained in the absence of CTSL in a fully TCRαβ polyclonal setting, we performed single-cell TCR sequencing of LLO-Tet⁺ cells sorted from $Ctsl^{+/+}$ and $Ctsl^{\Delta TEC}$ mice, yielding 316 and 156 paired TCRαβ clonotypes, respectively (Fig. 6a). Most of these were detected only once within each genotype; however, five TCRs were shared between $Ctsl^{+/+}$ and $Ctsl^{\Delta TEC}$ mice (Fig. 6a). To explore the characteristics of such CTSL-independent CD4⁺ T cell clones, we generated two TCR transgenic mouse lines, hereafter $Lm54^{Tg}$ and $Lm6^{Tg}$. $Lm54^{Tg}$ or $Lm6^{Tg}$ mice gave rise to CD4SP cells on both the $Rag1^{-/-}Ctsl^{+/+}$ and $Rag1^{-/-}Ctsl^{\Delta TEC}$ backgrounds (Fig. 6b,c). However, CD4SP cells were reduced for both TCRs on the $Ctsl^{\Delta TEC}$ background (markedly for $Lm6$ cells and subtly for $Lm54$ cells; Fig. 6b,c and Extended Data Fig. 5a,b), with a corresponding reduction in peripheral CD4⁺ T cell numbers (Fig. 6d,e), indicating a graded impairment in positive selection in the absence of CTSL, even for these apparently CTSL-independent TCRs.

Although CD4SP cells were reduced in $Rag1^{-/-}Ctsl^{\Delta TEC}Lm54^{Tg}$ mice compared to $Rag1^{-/-}Ctsl^{+/+}Lm54^{Tg}$ mice (hereafter $Ctsl^{\Delta TEC}Lm54^{Tg}$ and $Ctsl^{+/+}Lm54^{Tg}$, respectively), their segregation into the SM, M1 and M2 subsets was virtually identical (Fig. 6f). To determine whether selection of $Lm54$ thymocytes in the presence or absence of CTSL affected the underlying TCR signals (despite seemingly identical developmental progression downstream of the positive selection checkpoint), we assessed the expression of the nuclear receptor Nur77, whose levels reflect ongoing or recent TCR signaling[29]. Although $Lm54$ thymocytes in $Ctsl^{+/+}$ mice showed typical dynamic Nur77 modulation, with upregulation in the signaled DP cells and return to baseline in M2 CD4SP cells, Nur77 expression was markedly attenuated in cells selected in $Ctsl^{\Delta TEC}$ mice (Fig. 6g), consistent with weaker selecting TCR–pMHC interactions in the absence of CTSL. In line with this, both $Lm54$ and $Lm6$ thymocytes selected in $Ctsl^{\Delta TEC}$ mice showed markedly lower CD5 upregulation than their counterparts selected in $Ctsl^{+/+}$ mice (Fig. 6g,h), and this pattern was also observed for CD6, whose expression likewise correlates with TCR signal strength[30] (Extended Data Fig. 5c). Thus, even for TCR clones that appeared CTSL independent in their repertoire seeding, CTSL was still crucial for calibrating the intensity of the positive selection signal.

### Functional tuning by CTSL regulates homeostatic fitness

We next compared gene expression profiles in M2 CD4SP cells from $Ctsl^{\Delta TEC}Lm54^{Tg}$ and $Ctsl^{+/+}Lm54^{Tg}$ mice. Genes more highly expressed in $Ctsl^{\Delta TEC}$ cells included the genes encoding the two subunits of co-receptor CD8, as well as ion channels or solute carriers such as TMIE and SLC16A5 (Fig. 7a), which typically peak in DP cells, whereas $Ccr8$, which is upregulated in CD4SP cells[31], was reduced, consistent with 'less mature' traits in thymocytes selected in the absence of CTSL. Gene set enrichment analysis (GSEA) further revealed underrepresented transcripts in $Ctsl^{\Delta TEC}$ cells linked to translation, mTORC1 signaling and Myc targets (Fig. 7b), and these correlated with reduced cell size and diminished upregulation of CD44, a marker of mTOR signaling[32,33] (Extended Data Fig. 6a,b), suggesting reduced metabolic activity. Consistent with this, ex vivo assessment of basal protein biosynthesis showed diminished translation in the absence of CTSL, first emerging in CD69⁺ signaled DP cells and persisting through the M2 CD4SP cell stage (Fig. 7c).

After 18 h of stimulation with plate-bound I-A^b:LLO$_{190-201}$ in vitro, M2 CD4SP cells from $Ctsl^{\Delta TEC}Lm54^{Tg}$ mice upregulated the activation

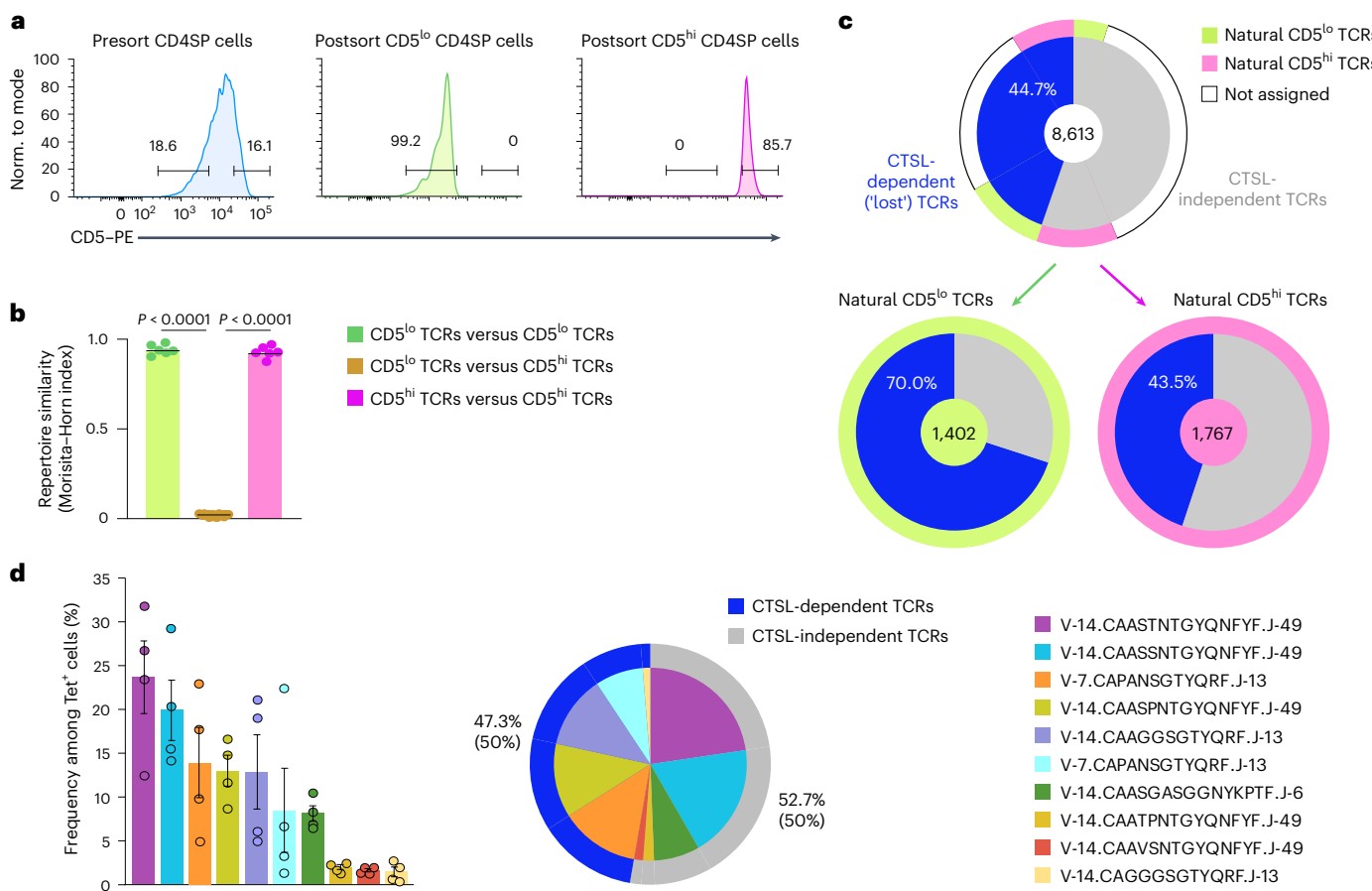

**Fig. 5 | CTSL deficiency impedes selection of clones at both extremes of the CD5 spectrum but retains a substantial proportion of nominal 'good-responder' TCRs. a**, Representative flow cytometry plot of CD5 expression on total presort M2 CD4SP cells, postsort 'natural' CD5$^{lo}$ cells and postsort CD5$^{hi}$ cells from $Tcrb^{Fixed}Ctsl^{+/+}$ mice, with sorting gates marking ~15% of cells at the low and high extremes of the CD5 spectrum. Histograms of postsort CD5$^{lo}$ and CD5$^{hi}$ cells show before bulk $Tcra$ sequencing ($n = 4$ each). **b**, Subrepertoire similarity comparison by Morisita–Horn index (mean ± s.e.m.) for all pairwise comparisons between $Tcra$ datasets from CD5$^{lo}$ and CD5$^{hi}$ M2 CD4SP cells as in **a** ($n = 6$ for CD5$^{lo}$ versus CD5$^{lo}$; $n = 16$ for CD5$^{lo}$ versus CD5$^{hi}$; $n = 6$ for CD5$^{hi}$ versus CD5$^{hi}$). **c**, Pie charts showing the proportion of CTSL-dependent and CTSL-independent TCRs among 8,613 recurrent TCRs in the $Tcrb^{Fixed}Ctsl^{+/+}$ CD4SP repertoire (top) and among 1,402 natural CD5$^{lo}$ TCRs (bottom left) and 1,767 natural CD5$^{hi}$ TCRs (bottom right). Top,

colored segments in the outer ring indicate the subset of TCRs cross-assigned to the natural CD5$^{lo}$ or natural CD5$^{hi}$ subrepertoires. **d**, Relative percentages ± s.e.m. of the ten most abundant clonotypes among expanded LLO-Tet$^+$CD4$^+$ T cells sorted from the spleens of $Tcrb^{Fixed}Ctsl^{+/+}$ mice at day 7 after i.v. challenge with LLO peptide plus poly(I:C), as quantified by bulk $Tcra$ sequencing ($n = 4$) (left). The pie chart shows the relative contribution of the ten most abundant clonotypes among expanded LLO-Tet$^+$CD4$^+$ T cells in LLO-immunized $Tcrb^{Fixed}Ctsl^{+/+}$ mice (middle). Colored segments in the outer ring indicate cross-assignment of these clones to CTSL-dependent and CTSL-independent TCRs among recurrent TCRs in the $Tcrb^{Fixed}Ctsl^{+/+}$ CD4SP repertoire. Percentages indicate the proportion of $Tcra$ reads (top) or clones (bottom, in parentheses). $P$ values in **b** were determined by one-way ANOVA with a Tukey's test for multiple comparisons.

marker CD69 with an approximately fivefold lower half-maximum inhibitory concentration than their counterparts from $Ctsl^{+/+}Lm54^{Tg}$ mice (Fig. 7d), indicating enhanced TCR sensitivity. Elevated responsiveness to TCR stimulation was likewise observed in polyclonal $Ctsl^{ΔTEC}$ M2 CD4SP cells (Extended Data Fig. 6c). By contrast, short-term stimulation with phorbol 12-myristate 13-acetate (PMA), which bypasses upstream TCR signaling, elicited comparable levels of ERK phosphorylation in M2 CD4SP cells selected in $Ctsl^{ΔTEC}Lm54^{Tg}$ and $Ctsl^{+/+}Lm54^{Tg}$ mice (Extended Data Fig. 6d), suggesting that the hyperresponsiveness of CD4SP cells selected in the absence of CTSL was confined to the proximal TCR signaling cascade. Flow cytometric analysis showed that, in addition to CD5 and CD6, two further negative regulators of TCR signaling (PD-1 and BTLA[34,35]) also exhibited reduced expression downstream of positive selection in $Ctsl^{ΔTEC}Lm54^{Tg}$ mice compared to $Ctsl^{+/+}Lm54^{Tg}$ mice (Fig. 7e). These findings suggest that the observed TCR-proximal hyperresponsiveness of cells selected in the absence of CTSL may be attributable to reduced expression of multiple TCR signaling attenuators.

Gene expression profiling of M2 CD4SP cells from $Ctsl^{ΔTEC}Lm54^{Tg}$ or $Ctsl^{+/+}Lm54^{Tg}$ mice after stimulation with plate-bound I-A$^b$:LLO$_{190–201}$ revealed a relative enrichment of transcriptional modules related to mRNA export, processing and splicing (processes potentially more directly linked to TCR activation) in cells selected in the absence of CTSL (Fig. 7f). By contrast, these cells exhibited comparably less efficient implementation of transcriptional programs related to key metabolic pathways crucial for sustaining T cell activation, including proliferation, glycolysis, mTORC1 signaling, cholesterol homeostasis and fatty acid metabolism (Fig. 7f). Many of these differences were not exclusively triggered following TCR activation but were already evident before stimulation and became more pronounced following TCR engagement (Fig. 7f), suggesting that these traits had been differentially 'imprinted' during positive selection in the presence or absence of CTSL.

Given the aberrant 'tuning' of multiple basal metabolic programs when $Lm54$ CD4$^+$ T cells were selected in the absence of CTSL, we next examined additional hallmarks associated with CD4$^+$ T cell

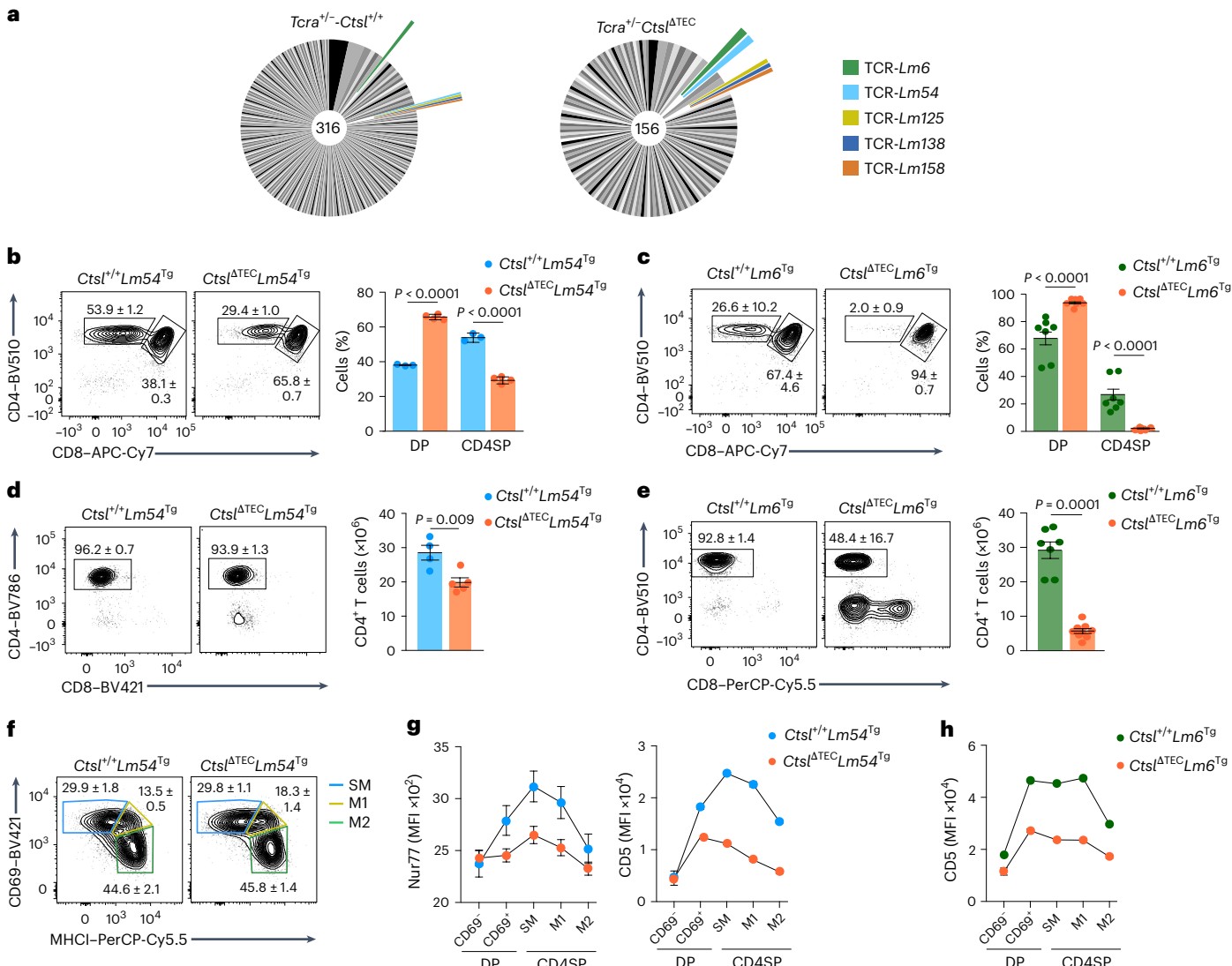

**Fig. 6 | LLO-specific TCRs retained in the absence of CTSL indicate a role for CTSL in calibrating positive selection signal strength. a**, Pie charts representing LLO-specific clonotypes in the fully polyclonal repertoire of $Tcra^{+/-}Ctsl^{+/+}$ mice ($n = 316$ TCRs) and $Tcra^{+/-}Ctsl^{\Delta TEC}$ mice ($n = 156$ TCRs), identified by single-cell $Tcra$ and $Tcrb$ sequencing of sorted LLO-Tet$^+$ cells ($n \geq 42$ mice per genotype; clonotypes aggregated across 19 experiments). Segments in shades of gray represent TCRs exclusively found in one genotype; colored segments represent 'public' TCRs shared between genotypes. Segment size is proportional to the number of mice in which a given TCR was detected. **b**, Representative flow cytometry plots and percent ± s.e.m. of thymocyte subsets in $Ctsl^{+/+}Lm54^{Tg}$ ($n = 3$) and $Ctsl^{\Delta TEC}Lm54^{Tg}$ mice ($n = 4$; hereafter $Ctsl^{\Delta TEC}Lm54^{Tg}$ and $Ctsl^{+/+}Lm54^{Tg}$, respectively). **c**, Representative flow cytometry plots and percent ± s.e.m. of thymocyte subsets in $Rag1^{-/-}Ctsl^{+/+}Lm6^{Tg}$ ($n = 7$) and $Rag1^{-/-}Ctsl^{\Delta TEC}Lm6^{Tg}$ mice ($n = 9$; hereafter $Ctsl^{\Delta TEC}Lm6^{Tg}$ and $Ctsl^{+/+}Lm6^{Tg}$, respectively).

**d**, Representative flow cytometry plots and number ± s.e.m. of lymph node CD4$^+$ T cells in $Ctsl^{+/+}Lm54^{Tg}$ ($n = 4$) and $Ctsl^{\Delta TEC}Lm54^{Tg}$ mice ($n = 5$). **e**, Representative flow cytometry plots and number ± s.e.m. of lymph node CD4$^+$ T cells in $Ctsl^{+/+}Lm6^{Tg}$ ($n = 7$) and $Ctsl^{\Delta TEC}Lm6^{Tg}$ mice ($n = 9$). **f**, Representative flow cytometry plots of MHCI and CD69 surface expression on CD4SP cells from $Ctsl^{+/+}Lm54^{Tg}$ ($n = 9$) and $Ctsl^{\Delta TEC}Lm54^{Tg}$ mice ($n = 10$). The percent ± s.e.m. of SM, M1 and M2 cells is indicated. **g**, Nur77 and surface CD5 expression (MFI ± s.e.m.) at consecutive DP and CD4SP stages of differentiation in $Ctsl^{+/+}Lm54^{Tg}$ ($n = 4$ or 5) and $Ctsl^{\Delta TEC}Lm54^{Tg}$ ($n = 4$ or 5) mice, assessed by intracellular staining and flow cytometry (Nur77) or flow cytometry (CD5). **h**, CD5 surface expression (MFI ± s.e.m.) at consecutive DP and CD4SP cell differentiation stages in $Ctsl^{+/+}Lm6^{Tg}$ ($n = 4$) and $Ctsl^{\Delta TEC}Lm6^{Tg}$ mice ($n = 5$), as assessed by flow cytometry. $P$ values in **b** and **c** were determined by two-way ANOVA and a Sidak's test for multiple comparisons. Data in **d** and **e** were analyzed by Student's two-tailed $t$-test.

survivability. Flow cytometric analysis showed that, across all maturation stages downstream of positive selection, expression of the interleukin-7 receptor (IL-7R) and BCL-2 (both key orchestrators of CD4$^+$ T cell survival[36]) was reduced in $Ctsl^{\Delta TEC}Lm54^{Tg}$ mice compared to $Ctsl^{+/+}Lm54^{Tg}$ mice (Fig. 7g).

We therefore next assessed whether M2 CD4SP cells from these two genotypes differed in their homeostatic properties. Following 24-h in vitro culture without added growth factors or TCR stimulation, cells selected in the absence of CTSL exhibited substantially higher apoptosis (Fig. 7h). Administration of high-dose IL-7 rescued this in vitro survival defect (Fig. 7i). To address whether selection in the absence

of CTSL impaired homeostasis in vivo, we conducted competitive co-transfer experiments using M2 CD4SP cells from $Ctsl^{\Delta TEC}Lm54^{Tg}$ and $Ctsl^{+/+}Lm54^{Tg}$ mice. Two weeks after co-transfer at a 1:1 ratio into sublethally irradiated wild-type recipients, the donor cell ratio had markedly shifted in disfavor of cells selected in the absence of CTSL (Fig. 7j). Analogous co-transfers of polyclonal M2 CD4SP cells from $Ctsl^{\Delta TEC}$ and $Ctsl^{+/+}$ donors to wild-type recipients recapitulated these observations and revealed a tendency toward diminished homeostatic proliferation of $Ctsl^{\Delta TEC}$ cells (Extended Data Fig. 6e,f).

To disentangle the complex interplay between tonic TCR–pMH-CII interactions and competition for soluble cues such as IL-7, which

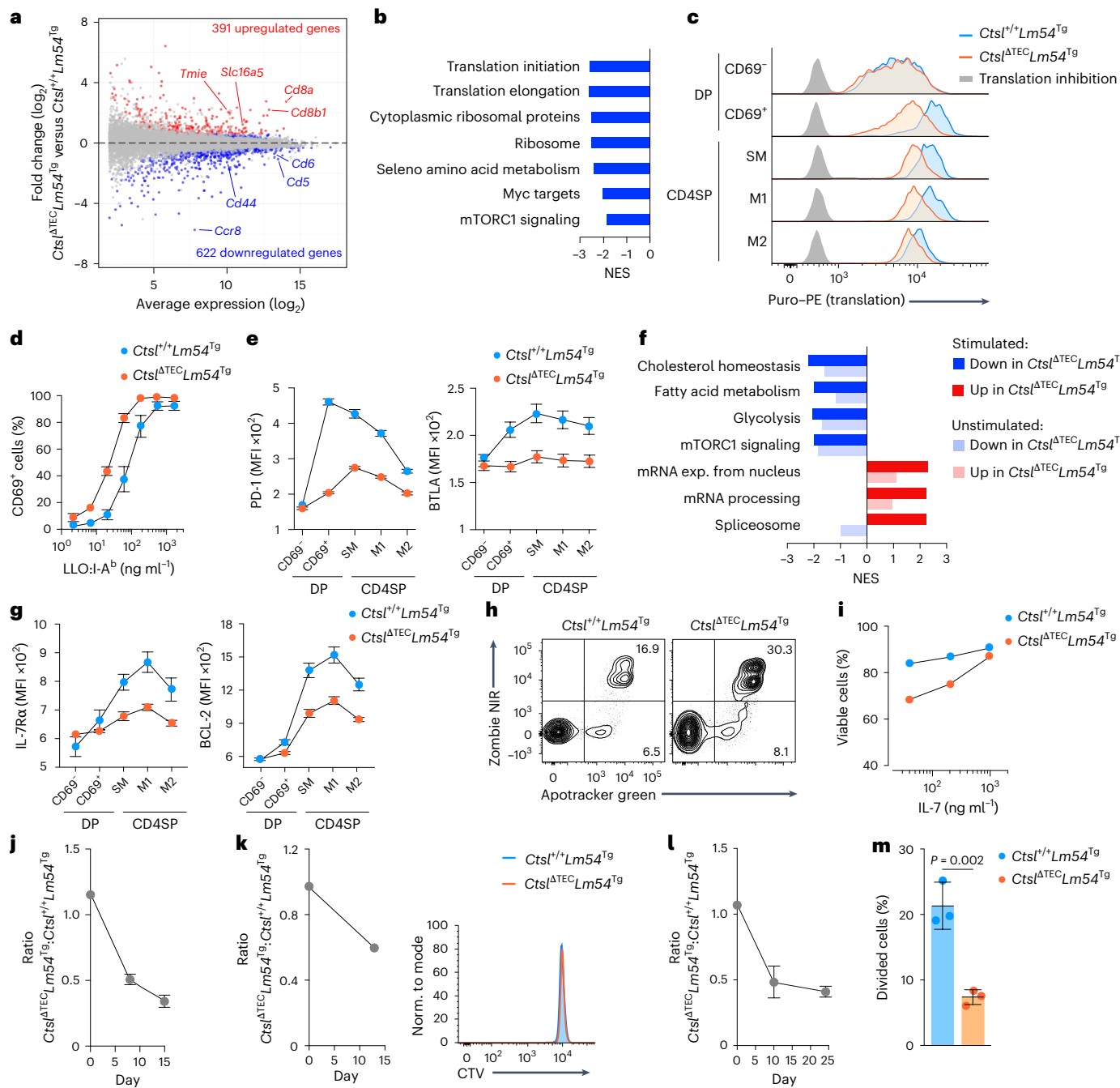

**Fig. 7 | CTSL deficiency causes aberrant functional tuning and impaired homeostatic fitness of CD4+ T cells. a**, MA plot of gene expression in *Ctsl*[+/+]*Lm54*[Tg] (*n* = 4) versus *Ctsl*[ΔTEC]*Lm54*[Tg] (*n* = 5) M2 CD4SP cells; differentially expressed genes (adjusted *P* value of <0.1) are shown in color. **b**, GSEA showing gene sets underrepresented (normalized enrichment score (NES) < −1.8) in *Ctsl*[ΔTEC]*Lm54*[Tg] cells as in **a**. **c**, Representative flow cytometry plots showing translation at thymocyte stages in *Ctsl*[+/+]*Lm54*[Tg] and *Ctsl*[ΔTEC]*Lm54*[Tg] mice (*n* = 3 each). A translation inhibitor (background) was included. **d**, CD69 expression (MFI ± s.e.m.) in M2 CD4SP cells from *Ctsl*[+/+]*Lm54*[Tg] and *Ctsl*[ΔTEC]*Lm54*[Tg] mice (*n* = 4 each) after 18 h of stimulation with plate-bound LLO[190–201]:I-A[b]. **e**, PD-1 and BTLA expression (MFI ± s.e.m.) at consecutive DP and CD4SP stages of differentiation in *Ctsl*[+/+]*Lm54*[Tg] (*n* = 3) and *Ctsl*[ΔTEC]*Lm54*[Tg] mice (*n* = 4), as assessed by flow cytometry. **f**, GSEA of *Ctsl*[+/+]*Lm54*[Tg] versus *Ctsl*[ΔTEC]*Lm54*[Tg] M2 CD4SP cells (*n* = 2 each) after 4 h of stimulation as in **d** or unstimulated as in **a**; only gene sets with a normalized enrichment score (NES) of >| 2 | in stimulated cells are shown. **g**, IL-7Rα and BCL-2 expression (MFI ± s.e.m.) at consecutive DP and CD4SP

stages of differentiation in *Ctsl*[+/+]*Lm54*[Tg] and *Ctsl*[ΔTEC]*Lm54*[Tg] mice (*n* = 5 each). Samples were analyzed by flow cytometry. **h**, Representative flow cytometry plots showing apoptosis in *Ctsl*[+/+]*Lm54*[Tg] and *Ctsl*[ΔTEC]*Lm54*[Tg] M2 CD4SP cells (*n* = 5 each) after 24 h of culture in normal medium. **i**, Viability (mean ± s.e.m.) of *Ctsl*[+/+]*Lm54*[Tg] and *Ctsl*[ΔTEC]*Lm54*[Tg] M2 CD4SP cells (*n* = 3 each) after 24 h of culture with IL-7, as assessed by flow cytometry. **j**, Donor cell ratio ± s.e.m. in the spleens of 4.5-Gy-irradiated recipients (*n* = 4) after transfer of a 1:1 mixture of *Ctsl*[+/+]*Lm54*[Tg] and *Ctsl*[ΔTEC]*Lm54*[Tg] M2 CD4SP cells, as assessed by flow cytometry. **k**, Ratio ± s.e.m. of donor cells in the spleens of *MHCII*[−/−] recipients (*n* = 3) after transfer of a 1:1 mixture of CellTrace Violet (CTV)-labeled *Ctsl*[+/+]*Lm54*[Tg] and *Ctsl*[ΔTEC]*Lm54*[Tg] M2 CD4SP cells. Histogram (right) shows representative CTV profiles on day 13 after transfer. **l**, Ratio ± s.e.m. of donor cells in the blood of *Rag1*[−/−] recipients (*n* = 4) after transfer of a 1:1 mixture of *Ctsl*[+/+]*Lm54*[Tg] and *Ctsl*[ΔTEC]*Lm54*[Tg] M2 CD4SP cells. **m**, Percent divided cells ± s.e.m. in an experimental replicate as in **l** with CellTrace Violet-labeled donor cells (*n* = 3 *Rag1*[−/−] recipients; day 9). Data were analyzed by Student's two-tailed *t*-test.

together sustain naive CD4[+] T cell maintenance and may also drive homeostatic proliferation[36], we repeated the co-transfer of M2 CD4SP cells from $Ctsl^{\Delta TEC}Lm54^{Tg}$ and $Ctsl^{+/+}Lm54^{Tg}$ mice using $MHCII^{-/-}$ or $Rag1^{-/-}$ recipients. In $MHCII^{-/-}$ hosts (where homeostatic MHCII contacts are abolished but survival factors such as IL-7 are readily available due to the absence of endogenous CD4[+] T cells), cells selected in the absence of CTSL again exhibited a competitive disadvantage, although neither donor population showed evidence of proliferation (Fig. 7k). This indicated a diminished survivability of cells selected in the absence of CTSL that was not explained by altered responsiveness to tonic TCR signaling and/or reduced homeostatic proliferation. In $Rag1^{-/-}$ recipients, which similarly provide ample access to soluble survival factors owing to their lack of an endogenous CD4[+] T cell compartment but, in contrast to $MHCII^{-/-}$ recipients, permit homeostatic TCR–pMHCII contacts, the donor cell ratio once more shifted in disfavor of cells selected in the absence of CTSL (Fig. 7l). However, unlike in $MHCII^{-/-}$ recipients (Fig. 7k), a significantly smaller proportion of these cells underwent at least one cell division (Fig. 7m), revealing defective homeostatic proliferation as an additional layer of impaired functional fitness. Thus, CD4[+] T cells selected in $Ctsl^{\Delta TEC}$ mice in a seemingly CTSL-independent manner (based on their progression to the mature CD4SP stage) retained a lasting functional imprint of their selection history, manifesting as a metabolically less poised state and diminished homeostatic responsiveness to both 'tonic' TCR signals and TCR-independent survival cues.

## Discussion

Our findings highlight two key roles for CTSL in shaping the CD4[+] T cell compartment by establishing full repertoire diversity and optimizing the fitness of clones that enter the repertoire in a seemingly CTSL-independent manner. We refer to the complete loss of certain clones as 'essential CTSL dependency' and to impaired functionality of retained clones as 'functional CTSL dependency'.

Essentially CTSL-dependent clones contributed disproportionately to repertoire diversity, conceivably reflecting reliance of low-abundance clones on few (or even a single) selecting pMHCII ligand(s). High-abundance clones may be more flexible in their range of selecting pMHCII ligands. If CTSL is required for only a subset of these ligands, the frequency of TCR–pMHC contacts may be reduced, allowing such clones to be retained in the repertoire, albeit with defects characteristic of functionally CTSL-dependent clones. CD5 expression in bulk CD4SP cells was diminished in the absence of CTSL. This was not due to preferential loss of 'natural' CD5[hi] clones, suggesting that reduced signal strength affected the full range of selecting interactions. As a consequence, natural CD5[lo] clones may more frequently fail to reach signaling thresholds for positive selection. Indeed, the proportion of essentially CTSL-dependent clones was higher among natural CD5[lo] clones. By contrast, higher-affinity natural CD5[hi] clones may persist through compensatory sensitization of proximal TCR signaling. This may involve reduced expression not only of CD5 but also of other negative regulators such as CD6, PD-1 and BTLA, all evident in clones $Lm54$ and $Lm6$.

Although reduced peptide diversity may explain some consequences of CTSL deficiency, this does not preclude the possibility that the key role of CTSL lies in generating 'qualitatively special' peptides optimal for selection. Despite the loss of ~50% of TCRs in $Tcrb^{Fixed}Ctsl^{\Delta TEC}$ mice, numerous 'newcomer TCRs' emerged, suggesting that the pMHCII ligandome remained diverse yet was enriched in otherwise absent or outcompeted 'newcomer ligands'. The unusual V–J features of newcomer TCRs imply that they arose through an atypical selection process and may not be functionally equivalent to normally selected CD4[+] T cells. As cTECs express at least two other cathepsins, cathepsin B and D[24], these may generate diverse, yet 'suboptimal', selecting peptides in the absence of CTSL. Circumstantial support for this notion comes from experiments showing that occupancy of <5% of MHCII molecules on cTECs by pMHC ligands generated by the normal proteolytic machinery, including CTSL, was sufficient to sustain CD4[+] T cell numbers near wild-type levels[37].

Essential and functional CTSL dependency likely both contribute to the blunted anti-LLO CD4[+] T cell response in $Ctsl^{\Delta TEC}$ mice. Half of the ten 'best-responder' clones in $Tcrb^{Fixed}Ctsl^{+/+}$ mice were physically absent from the repertoire of $Tcrb^{Fixed}Ctsl^{\Delta TEC}$ mice. We deem it likely that both in $Tcrb^{Fixed}$ mice and in the fully αβ polyclonal repertoire, a sizeable proportion of clones retained in the absence of CTSL are functionally compromised. This is exemplified by the clones $Lm54$ and $Lm6$, whose altered features when selected in the absence of CTSL may be the rule rather than the exception. However, their marked homeostatic defects complicate efforts to resolve how functional CTSL dependency affects responsiveness and effector functions following immunogenic challenge with cognate antigen.

The exact mechanism underlying the impaired homeostatic fitness of functionally CTSL-dependent clones remains unclear. These clones appeared to progress 'on autopilot' through differentiation stages downstream of positive selection in the absence of CTSL yet exhibited alterations in multiple cellular programs. A downshift in CD5 expression may directly contribute to their inferior homeostatic behavior. In normally selected thymocytes, elevated CD5 expression calibrates the NF-κB pool, promoting viability and responsiveness despite attenuating TCR signals[38]. NF-κB also establishes proliferation competence[26] and augments IL-7 responsiveness, triggering prosurvival transcriptional programs including BCL-2 (ref. 39). IL-7R expression is fine-tuned by TCR signals during thymic development[40], and reduced IL-7R and BCL-2 expression in functionally CTSL-dependent clones suggests that selection in the absence of CTSL led to diminished IL-7 responsiveness and compromised survivability.

The 'altered peptide' model, proposed over three decades ago, suggested that developing thymocytes engage unique pMHC combinations for positive selection[41]. This model was later abandoned when it was found that several abundant MHCII-associated peptides were shared between cTECs and other APCs[42], and the 'affinity model' has become the prevailing concept to explain the dual role of self-pMHC ligands in both positive and negative selection. However, the discovery of distinct proteolytic pathways in cTECs has revived interest in the possibility of 'private' pMHC ligands in cTECs having crucial physiological relevance. The 'peptide-switch' model proposes that minimizing the overlap between the pMHC ligandomes of cTECs and those of negatively selecting APCs prevents positive and negative selection from canceling each other out[15,43]. Although we cannot formally exclude contributions from negative selection, our findings demonstrate that the diminished CD4SP compartment in $Ctsl^{\Delta TEC}$ mice primarily reflects a genuine bottleneck in positive selection.

How 'private' peptides generated by CTSL contribute to optimal CD4[+] T cell selection remains speculative. Their unique role may be specified through conserved TCR contact and/or MHC anchor residues or via an allosteric influence on the 'nonpeptide' MHC–TCR interface. Methodological advances will be required to comprehensively characterize the cTEC pMHC ligandome. Another intriguing question is how the requirement for 'private' peptides in positive selection can be reconciled with the hypothesis that the very same self-peptides mediating positive selection also support naive T cell homeostasis in the periphery and act as coagonists when T cells respond to foreign antigens[44,45].

## Online content

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

[1]Institute for Immunology, Biomedical Center, Faculty of Medicine, LMU Munich, Planegg-Martinsried, Germany. [2]Institute of Immunology, Hannover Medical School, Hannover, Germany. [3]Division of Animal Physiology and Immunology, School of Life Sciences Weihenstephan and TUM Center for Infection Prevention, Technical University of Munich, Freising, Germany. [4]Institute for Experimental Hematology, Center for Translational Cancer Research (TranslaTUM), School of Medicine and Health, Technical University of Munich, Munich, Germany. [5]Institute of Systems Immunology, Hamburg Center for Translational Immunology, University Medical Center Hamburg-Eppendorf, Hamburg, Germany. [6]Department of Tumor Immunology, Radboud Institute for Molecular Life Sciences, Radboud University Medical Center, Nijmegen, the Netherlands. [7]Institute for Molecular Health Sciences, ETH Zürich, Zürich, Switzerland. [8]Institute of Molecular Medicine and Cell Research, Faculty of Medicine, University of Freiburg, Freiburg, Germany. [9]Bioinformatics Unit, Biomedical Center, Faculty of Medicine, LMU Munich, Planegg-Martinsried, Germany. [10]Centre for Immunology and Infection Control, School of Biomedical Sciences, Faculty of Health, Queensland University of Technology, Brisbane, Queensland, Australia. [11]Present address: Onkologisches Zentrum Freising MVZ, Freising, Germany. [12]Present address: Repertoire Immune Medicines, Schlieren, Switzerland. [13]These authors contributed equally: Stephen R. Daley, Ludger Klein. ✉e-mail: ludger.klein@med.uni-muenchen.de

## Methods

### Mice

$Ctsl^{fl/fl[46]}$, $Ctsl^{-/-[47]}$, $Tcr\text{-}Dep^{48}$, $Ciita^{[kd28]}$, $Tcr\text{-}Plp1$ (ref. [49]), $Tcr\text{-}LLO56$ and $Tcr\text{-}LLO118$ (ref. [50]), $Tcr\text{-}AND$ and $Tcr\text{-}AD10$ (ref. [51]), $Tcr\text{-}OT\text{-}II^{52}$, $Foxn1\text{-}cre^{53}$, $MHCI^{-/-}$ $(B2m^{-/-})^{54}$, $MHCII^{-/-}$ $(H2\text{-}Ab1^{-/-})^{55}$, $Rag1^{-/-[56]}$, $Plp1^{-/-[57]}$, $Tcra^{-/-[58]}$ and $Foxp3^{GFP}$ reporter mice (DEREG)[59] have been described previously. For the generation of $Lm54^{Tg}$ and $Lm6^{Tg}$ mice, pTα and pTβ cassette vectors[60] were modified to contain the V(D)J regions of the respective TCRs identified by single-cell TCR sequencing. Transgenic mice were generated by injection of linearized DNA into the pronuclei of C57BL/6 zygotes. $Tcrb^{Fixed}$ transgenic mice only express the TCRβ chain of the $Lm54$ clone. All mice used were on a C57BL/6J background. Mice were maintained under specific pathogen-free conditions in individually ventilated cages at an ambient temperature of 22 °C and 55% humidity with standard light cycle conditions. All phenotypic analyses were performed in mice of 8–12 weeks of age, unless otherwise indicated, and animals of both sexes were included, as we did not find any evidence for sex differences in the parameters addressed. Animal studies and procedures were approved by local authorities (Regierung von Oberbayern; Az Vet_02-22-66).

### CTSL active site labeling and western blotting

Sorted TECs were incubated for 1 h at 37 °C in conditioned culture medium containing 1 µM BMV109 and processed as previously described[61]. Briefly, cell pellets obtained from $3 \times 10^4$ TECs were lysed in hypotonic buffer in the presence of 4 mM DTT for 15 min on ice and centrifuged for 30 min at 10,000g. Supernatants were denatured by the addition of 3× SDS sample buffer (containing 10% β-mercaptoethanol) and resolved by SDS–PAGE (15%) at 120 V for 60 min, together with the protein marker ROTI Mark Tricolor (Roth). The gel was scanned with a Typhoon FLA9500 imager in the Cy5 channel (GE Healthcare). Proteins were either stained with Coomassie or transferred in an exact replica of the gel onto a nitrocellulose membrane via semidry western blotting. Transfer was performed with transfer buffer containing 8% methanol and at a maximum of 50 V and 160 mA for 90 min. Following protein transfer, the membrane was incubated with goat anti-mouse CTSL polyclonal IgG (AF1515, R&D; 1 µg ml$^{-1}$), followed by secondary mouse anti-goat horseradish peroxidase-conjugated polyclonal IgG (205-035-108, Jackson ImmunoResearch; 40 ng ml$^{-1}$) and SuperSignal West Pico Chemiluminescent Substrate (Thermo Scientific). Membranes were imaged with an iBright1500 scanner (Invitrogen). For the housekeeping control, mouse mAb to β-actin (clone AC-15, Sigma; 200 ng ml$^{-1}$) was used as primary antibody, followed by secondary rabbit anti-mouse horseradish peroxidase-conjugated polyclonal IgG (P0260, Agilent Dako).

### Histology

Five-micron sections of paraffin-embedded material were stained with hematoxylin (Mayer) and eosin (Morphisto). Microscopy was performed with a Leica DM2500 brightfield microscope, equipped with a DMC2900 CMOS camera and HC PL FL ×10/0.30-NA PH1 or HC PL FLUOTAR ×5/0.15-NA objective. The resulting image pixel sizes were 581 nm (×10) and 1.162 nm (×5). Microscopy images were acquired with LAS X Office v1.4.6 (Leica Microsystems).

### Determination of cell size

Cell size measurements were conducted on a Countess 3 Automated Cell Counter (Thermo Fisher).

### Preparation of TECs

Thymi from 3- to 5-week-old animals were cut into pieces, and thymocytes were mechanically released by pipetting up and down. The supernatant containing thymocytes was discarded. The thymus fragments were digested with liberase (0.5 U ml$^{-1}$; Roche) and DNase I (10 mg ml$^{-1}$; Roche) at 37 °C in three consecutive rounds of 15 min.

Cells were washed and resuspended in 1 ml of high-density Percoll ($\rho = 1.115$; GE Healthcare) and overlaid with 1 ml of low-density Percoll ($\rho = 1.055$), followed by a layer of 1 ml of RPMI (Gibco). The gradient was centrifuged at 1,350g for 30 min at 4 °C (without brake). The top interphase, containing the low-density cell fraction, was collected, washed and subjected to CD45 magnetic-activated cell sorting depletion, using CD45 MicroBeads (Miltenyi Biotech). The CD45$^-$ fraction was stained with DAPI and surface antibodies. TECs were analyzed or sorted according to the expression of CD45, Ly51, EpCAM, CD80 and MHCII as follows: cTECs (CD45$^-$EpCAM$^+$Ly51$^+$), total mTECs (CD45$^-$EpCAM$^+$Ly51$^-$) and mTEC$^{hi}$ (CD45$^-$EpCAM$^+$Ly51$^-$ MHCII$^{hi}$, CD80$^{hi}$).

### Flow cytometry

Antibodies were purchased from Biolegend, unless otherwise specified, and used as conjugates with various fluorochromes: anti-CD4 (clone RM4-5), anti-CD8α (53-7.3), anti-CD326/EpCAM (G8.8), anti-Ly51 (6C3), anti-CD80 (16-10A1), anti-CD5 (53-7.3), anti-TCRβ (H57-597), anti-CD69 (H1.2F3), anti-H-2K$^b$ (AF6-88.5), anti-CD45.1 (A20), anti-CD45.2 (104), anti-CD44 (IM7), anti-CD62L (MEL-14), anti-TCRvα2 (B20.1), anti-CD127/IL-7Rα (A7R34), anti-PD-1 (RMP1-30), anti-BTLA (6A6) and anti-I-A/I-E (M5/114.15.2). The following antibodies were used to distinguish MHCII structural epitopes: anti-CLIP:I-A$^b$ (15G4)[62], anti-non-CLIP:I-A$^b$ (BP107.2.2, a kind gift from A. Rudensky, Memorial Sloan Kettering Cancer Center)[63] and anti-Eα$_{52-68}$:I-A$^b$ (Y-Ae)[64]. To determine translation activity, thymocytes were pulsed for 5 min with 5 µg ml$^{-1}$ puromycin (Merck) immediately before intracellular staining with phycoerythrin (PE)-conjugated mAb to puromycin (2A4)[65]. Control cells were pretreated for 15 min with 5 µg ml$^{-1}$ harringtonine (MedChem Express) and, during the puromycin pulse, treated with 100 µg ml$^{-1}$ cycloheximide (Roth) and 150 µg ml$^{-1}$ chloramphenicol (Merck). For intracellular staining, cells were fixed and permeabilized using reagents from a Foxp3 staining kit (eBioscience) and stained with PE-conjugated anti-Nur77 (12.14, eBioscience) or PE-conjugated anti-BCL-2 (BCL/10C4). Cells were analyzed using a BD FACSCANTOII or LSRFortessa flow cytometer or sorted using a BD FACSAriaFusion sorter. Flow cytometry data were acquired using FACSDiva v6.2 software (BD Bioscience). Flow cytometry data were analyzed with FlowJo v10.9.0 software.

### Tet staining

Tets conjugated to APC or PE were kindly provided by M. Jenkins (University of Minnesota)[66]. Single-cell suspensions from thymi or pooled lymph nodes and spleens were incubated for 1 h at 25 °C with 10 nM of both APC- and PE-conjugated Tets, as previously described[67]. Cells were next washed and enriched with anti-APC and anti-PE MicroBeads (Miltenyi Biotech). AccuCheck Counting beads (Thermo Fisher) were used to determine the number of Tet$^+$ cells in the column-bound fraction. Tet staining on samples from LLO-immunized mice was followed by enrichment with anti-CD4 MicroBeads (Miltenyi Biotech), instead of anti-APC/anti-PE MicroBeads. A dump cocktail of antibodies, containing anti-CD11b (M1/70), anti-CD11c (N418), anti-B220 (RA3-6B2) and anti-F4/80 (BM8), was used to exclude non-T cells and autofluorescent cells in flow cytometric analyses.

### BM chimeras

Recipient mice were irradiated with two split doses of 4.5 Gy, at least 2 h apart, the day before reconstitution. BM was depleted of differentiated cells using biotinylated mAbs to CD4 (clone RM4-5), CD8α (53-7.3), B220 (RA3-6B2), CD11b (M1/70), CD11c (N418), Gr1 (RB6-8C5) and F4/80 (BM8; Biolegend) together with streptavidin magnetic-activated cell sorting MicroBeads (Miltenyi Biotech). Recipient mice were injected i.v. with $5 \times 10^6$–$10 \times 10^6$ BM cells. Neomycin (Belapharm) was supplemented in the drinking water for the first 4 weeks, and mice were killed 6 weeks after reconstitution.

## Large-scale *Tcra* sequencing

For experiments depicted in Fig. 3 (global M2 repertoire comparison in *Ctsl*$^{\Delta TEC}$ versus *Ctsl*$^{+/+}$ *Tcrb*$^{Fixed}$ mice), $3 \times 10^5$–$4 \times 10^5$ mature M2 CD4SP (CD4$^+$CD8$\alpha^-$CD69$^-$MHCI$^+$CD25$^-$FoxP3$^-$) cells from 5- to 8-week-old *Ctsl*$^{\Delta TEC}$*Tcra*$^{+/-}$*Tcrb*$^{Fixed}$Foxp3$^{GFP}$ mice or corresponding controls were bulk sorted. Material from two to three mice was pooled to obtain comparable total cell counts in each sample. For experiments depicted in Fig. 5a–c (global 'natural' CD5$^{lo}$ versus 'natural' CD5$^{hi}$ M2 repertoire comparison in *Ctsl*$^{+/+}$*Tcrb*$^{Fixed}$ mice), $1 \times 10^5$–$2 \times 10^5$ mature M2 CD4SP cells (CD4$^+$CD8$\alpha^-$CD69$^-$MHCI$^+$CD25$^-$FoxP3$^-$), corresponding to the 15% lowest or 15% highest levels of CD5 expression on total CD4SP cells, were bulk sorted from individual 5- to 8-week-old *Ctsl*$^{+/+}$*Tcra*$^{+/-}$ *Tcrb*$^{Fixed}$Foxp3$^{GFP}$ mice. For experiments depicted in Fig. 5d,e (TCR analysis of expanded LLO-Tet$^+$ cells), $1 \times 10^5$–$2 \times 10^5$ LLO-Tet$^+$ cells were bulk sorted from pooled spleen and lymph node cells of individual 6- to 8-week-old *Ctsl*$^{+/+}$*Tcra*$^{+/-}$ *Tcrb*$^{Fixed}$Foxp3$^{GFP}$ mice 7 days after systemic immunization with LLO$_{190-201}$. Samples were stored at −80 °C in RNAprotect (Qiagen). All further sample preparation steps were performed by Qiagen. RNA was isolated using an RNeasy Mini/Micro kit (Qiagen), and library preparation was performed using a QIAseq Immune Repertoire RNA Library kit (Qiagen), modified to include only *Tcra*-specific primers for both the reverse transcription and target enrichment steps. Library preparation quality control was performed using an Agilent DNA 7500 Chip and Qubit dsDNA HS. Libraries that passed quality control were finally sequenced on a NovaSeq 6000 (Illumina) sequencing instrument according to the manufacturer's instructions, with a read length of $2 \times 250$ bp and an SP flow cell. Raw data were demultiplexed, and FASTQ files for each sample were generated using bcl2fastq software (Illumina).

## Single-cell TCR sequencing

To increase the likelihood of finding shared TCRs and to ensure subsequent detection of transgenically re-expressed TCRs by fluorescence-activated cell sorting, we restricted this analysis to cells stainable with the mAbs MR9-4 (anti-TCR-V$\beta$5; genes *Trbv5.n*) and B20.1 (anti-TCR-v$\alpha$2; genes *Trav14.n*). Dump cocktail-negative MR9-4$^+$B20.1$^+$Tet$^+$ cells were single-cell sorted into 96-well plates and immediately frozen at −80 °C. All subsequent steps were adapted from Dössinger et al.[68]. Reverse transcription was performed using an iScript Select cDNA Synthesis kit (Bio-Rad) with a mix of primers specific for the TCR$\alpha$ (*Trac*) and TCR$\beta$ (*Trbc*) constant regions. Reverse transcription was followed by a digestion step with exonuclease I (Thermo Scientific) to remove single-stranded primers. The exonuclease digestion product was further subjected to a dGTP tailing step and split into two 96-well plates to continue with separate PCRs for *Trac* and *Trbc* sequencing. The first PCR was performed with an anchor primer complementary to the introduced 3′-guanosine overhang together with primers specific for *Trac* or *Trbc*, respectively. The following two rounds of nested PCR were performed with primers binding *Trav14* and *Trac* or *Trbv5* and *Trbc*, respectively. PCR products were Sanger sequenced by Eurofins Genomics by using either a *Trav14*-specific primer (for TCR$\alpha$) or a *Trbv5*-specific primer (for TCR$\beta$). Both *Trac* and *Trbc* sequences were annotated using IMGT/V-Quest[69] with the C57BL/6-specific library. TCR clonotypes were defined at the level of amino acid sequence (V-region, CDR3 and J-region), and only those consisting of both an in-frame *Trac* and an in-frame *Trbc* were retained. As each TCR clonotype found in a given mouse was counted only once, counts in the Source Data related to Fig. 6a refer to the number of individual mice where the respective clonotype was detected.

## TCRα repertoire analysis

The CLC Genomics Workbench software (v23.0.3) provided by Qiagen was used to generate clonotype reads. Briefly, sequences with the same unique molecular index were merged. Further quality control steps included merging and trimming of overlapping paired-end sequences. Results were annotated using the C57BL/6J-specific functional *Tcra* genes extracted from the IMGT mouse TCR database. Analyses were performed using RStudio (version 2024.04.2+764). A clonotype was defined as a unique combination of *Trav* gene and *Traj* gene and an in-frame CDR3 amino acid sequence. Clonotype abundance was defined as the number of clonotype-encoding unique cDNA molecules (distinguished by unique molecular index) detected in a sample. TCR diversity was assessed using the approach of Chao et al.[70], wherein one unique cDNA molecule was defined as one 'individual' and one clonotype as one 'species'. Rarefaction with Hill numbers was performed using 'iNEXT' (versions 2.0.20 and 3.0.1). Shannon diversity is the exponential of Shannon entropy and is calculated in 'iNEXT' using $q = 1$. Shannon diversity (or effective number of TCR clonotypes) is the number of equally abundant clonotypes that would be needed to give the same Shannon diversity as the sample. As sampling completeness approached the maximum (coverage = 1) for each sample and subrepertoire examined, the observed Shannon diversities were used to calculate 'relative diversity'. As adding 95% confidence intervals constructed from five bootstrap replications made a negligible difference to the appearance of the rarefaction curves and were smaller than the symbols on the 'relative diversity' summaries, these details were omitted. To assess TCR overlap between pairs of samples, we used 'abdiv' (version 0.2.0) to calculate the Morisita–Horn index. Quantification of nucleotide deletion or addition at *Trav*–*Traj* junctions was performed using results from the IMGT Junction Analysis tool. The 'no. nucleotides deleted or added at the *Trav*–*Traj* junction' was defined as the sum of the absolute values of the number of nucleotides deleted from the germline *Trav* and/or *Traj* gene(s) plus the number of P and/or N nucleotides added between the remaining germline *Trav* and/or *Traj* nucleotides. Due to duplication events at the *Tcra* locus in C57BL/6 mice, 23–30% of unique cDNA molecules per sample aligned with greater than one *Trav* paralog; these sequences were excluded from the analyses of chromosomal location of *Trav* gene usage and nucleotide deletion or addition at *Trav*–*Traj* junctions.

## Immunization

For hock immunization, isoflurane-anesthetized mice were subcutaneously injected with an emulsion of LLO$_{190-201}$ peptide (NEKYAQAYPNVS, GenScript) in PBS and Freund's adjuvant (Sigma). Systemic immunization was performed by i.v. injection of 50 µg of LLO$_{190-201}$ peptide and 50 µg of high-molecular-weight poly(I:C) (InvivoGen) in PBS. When combined with adoptive cell transfers, mice were immunized 6 h before the injection of donor cells.

## Adoptive T cell transfer

For experiments depicted in Fig. 4d, mature CD4SP conventional thymocytes from *Ctsl*$^{\Delta TEC}$ Foxp3$^{GFP}$ (CD45.2) and Foxp3$^{GFP}$ littermate control (CD45.1) mice were sorted (CD4$^+$CD8$\alpha^-$CD69$^-$MHCI$^+$CD25$^-$FoxP3$^-$) and mixed at a 1:1 ratio. Cells were injected i.v. into CD45.1/CD45.2 recipients ($10^6$ total cells per mouse), previously immunized with LLO plus poly(I:C). For transfer experiments depicted in Fig. 7, M2 CD4SP thymocytes from *Ctsl*$^{\Delta TEC}$*Lm54*$^{Tg}$*Rag1*$^{-/-}$ and *Ctsl*$^{+/+}$*Lm54*$^{Tg}$*Rag1*$^{-/-}$ mice were sorted, mixed at a 1:1 ratio and i.v. injected into CD45.1/CD45.2 recipients ($1 \times 10^6$ total cells per mouse). Where indicated, the cells were labeled with CellTrace Violet (Invitrogen; 5 µM for 5 min at 37 °C) before adoptive transfer. All results were confirmed with inverted combinations of congenic markers.

## In vitro stimulation of *Lm54* CD4SP thymocytes

For antigen-specific in vitro stimulation, M2 CD4SP thymocytes were stimulated for 18 h at 37 °C in the presence of soluble anti-CD28 (clone 37.51, Bio X Cell; 250 ng ml$^{-1}$) and titrated amounts of plate-bound LLO$_{190-201}$:I-A$^b$ monomer. All stimulations were performed with $1.5 \times 10^5$ cells per well in 96-well, U-bottom plates and HL-1 serum-free

medium (Lonza). The viability dyes Zombie NIR and Apotracker Green (Biolegend) were used to exclude dead and apoptotic cells. Short-term PMA stimulation was performed with 50 ng ml$^{-1}$ PMA for 5 min at 37 °C.

### RNA sequencing and analysis

Cells were collected and stored at −80 °C in RNAprotect (Qiagen). All further sample preparation steps (RNA isolation, rRNA depletion and library preparation) were performed by Eurofins Genomics, using the INVIEW transcriptome discovery package. Libraries that passed quality control were sequenced on a NovaSeq 6000 (Illumina) sequencing instrument, with 2 × 150 bp read length and S4 flow cells. All data processing methods were applied using default parameters unless specified. Expression quantification was performed using kallisto (version 0.48) with Ensembl release version 106 for *M. musculus*. In R/Bioconductor, expression data were collapsed from the isoform level to the gene level for downstream processing. Differential expression was assessed using DESeq2 (version 1.36). GSEAs were conducted using fgsea (version 1.22).

### Statistical analysis

Statistical analyses were performed using GraphPad Prism (v9), except for calculations of Shannon diversity and Morisita–Horn indexes. The specific statistical tests used are indicated in the corresponding figure legends. Data distribution was assumed to be normal, but this was not formally tested. No a priori sample size calculations were performed; sample sizes were based on prior experience with similar experiments and were deemed sufficient to detect biologically meaningful differences. Measurements were not conducted blind to the conditions of the experiment, as flow cytometric outputs were analyzed using standardized, objective gating strategies. No data points were excluded from analysis.

### Reporting summary

Further information on research design is available in the Nature Portfolio Reporting Summary linked to this article.

## Data availability

Sequencing data from this study have been deposited at the Gene Expression Omnibus and will be publicly available from the date of publication. The accession numbers are GSE269202 for the bulk TCR-sequencing data (relating to Figs. 3c–g and 5b–e) and GSE269197 for the RNA-sequencing data (relating to Fig. 7a,b,f). Source data are provided with this paper.

## Code availability

An executable capsule containing code and bulk TCR-sequencing data is available at https://doi.org/10.24433/CO.5359169.v1.

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

## Acknowledgements

E.P. was supported by the Boehringer Ingelheim Fonds. E.P., A.S. and L.K. received funding from the Deutsche Forschungsgemeinschaft (DFG; German Research Goundation) under project number

456882036. A.S. and L.K. received funding from the DFG under SFB-TRR 355/1 Project B01 (490846870). J.A.F. and L.K. were supported by the European Union Horizon 2020 Research and Innovation Program under the Marie Skłodowska-Curie grant, agreement number 675395 (ENLIGHT-TEN+). T.R. received funding from the DFG GRK 2606 (423813989). S.R.D. received funding from the Australian National Health and Medical Research Council (grant 1188589) and Queensland University of Technology. A.M.S. was supported by European Union's Horizon 2020 research and innovation program under the Marie Skłodowska-Curie grant agreement number 754462 and by the DFG (419162346 and SFB1371). We thank P. Allen (Washington University) for LLO118 and LLO56 mice, M. Jenkins (University of Minnesota) for MHCII Tets and A. Rudensky (Memorial Sloan Kettering Cancer Center) for the BP107.2.2 hybridoma. We thank B. Tast and the staff of the Flow Cytometry Core Facility at the Biomedical Center for flow cytometry support.

## Author contributions

E.P., J.N., S.R.D. and L.K. designed the experimental strategies. E.P., A.S., C.F., S.R., A.M.S. and B.A. performed the experiments. J.A.F., I.P., S.R. and S.R.D. analyzed TCR repertoires. M.V. supported active site labeling experiments. J.K. contributed to the characterization of TCRs. M.S.-S. designed transgenic constructs. T.S. performed gene expression analyses. D.Z. designed immunization strategies. T.R. generated the conditional knockout model. M.V., R.O. and T.R. contributed essential materials. L.K., S.R.D. and E.P. wrote the manuscript.

## Funding

## Competing interests

The authors declare no competing interests.

## Additional information

**Extended data** is available for this paper at https://doi.org/10.1038/s41590-025-02182-y.

**Correspondence and requests for materials** should be addressed to Ludger Klein.

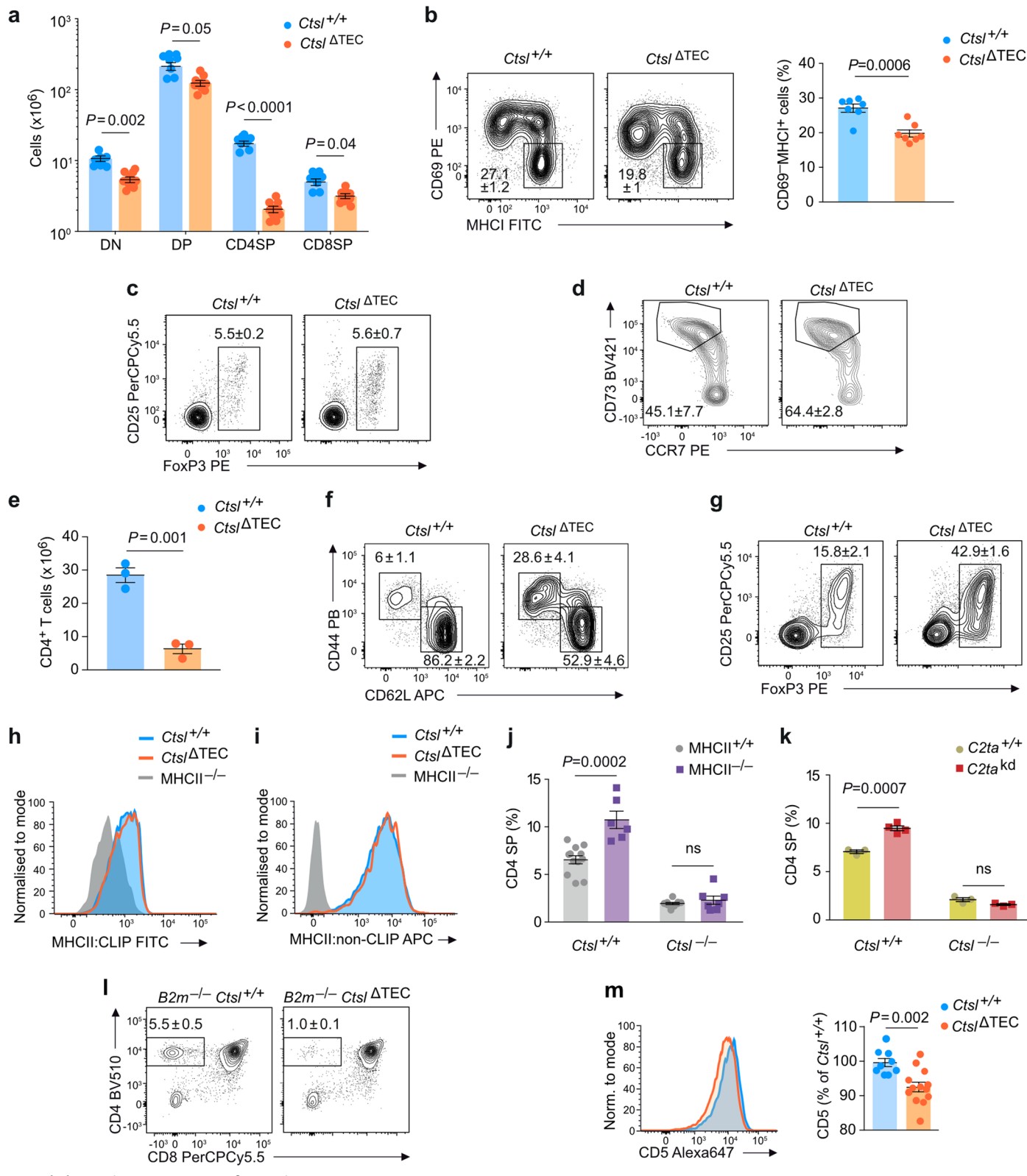

**Extended Data Fig. 1 | See next page for caption.**

**Extended Data Fig. 1 | Ctsl-deficiency alters the cTEC pMHCII ligandome and impairs CD4⁺ T cell positive selection. a**, Thymocyte subset cell numbers (Mean ± SEM) in *Ctsl*^+/+ (*n* = 8) and *Ctsl*^ΔTEC mice (*n* = 10). **b**, Representative flow cytometry plots of MHCI and CD69 on CD4SPs and percentage ± SEM of M2 cells in *Ctsl*^+/+ (*n* = 7) and *Ctsl*^ΔTEC mice (*n* = 7). **c**, Representative flow cytometry plots of CD25 versus Foxp3 and percentage ± SEM of Foxp3⁺CD25⁺ T_reg cells in CD4SPs in *Ctsl*^+/+ (*n* = 7) and *Ctsl*^ΔTEC mice (*n* = 7); difference not significant. **d**, Representative flow cytometry plots of CD73 versus CCR7 on Foxp3⁺CD25⁺ CD4SPs in *Ctsl*^+/+ (*n* = 3) and *Ctsl*^ΔTEC mice (*n* = 4). Percentage ± SEM of CD73⁺CCR7⁻ re-immigrants is indicated; P = 0.1. **e**, Mean ± SEM of lymph node CD4⁺ T cells in *Ctsl*^+/+ (*n* = 3) and *Ctsl*^ΔTEC mice (*n* = 3). **f**, Representative flow cytometry plots of CD44 versus CD62L on CD4⁺ T cells from *Ctsl*^+/+ (*n* = 3) and *Ctsl*^ΔTEC mice (*n* = 3). **g**, Representative flow cytometry plots of CD25 versus Foxp3 and percentage ± SEM of Foxp3⁺CD25⁺ T_reg cells in CD4⁺ T cells from *Ctsl*^+/+ (*n* = 3) and *Ctsl*^ΔTEC mice (*n* = 3); P = 0.016. **h**, Representative flow cytometry plots of MHCII:CLIP on CD80⁻ (immature) mTECs from *Ctsl*^+/+ (*n* = 13) and *Ctsl*^ΔTEC mice (*n* = 13); MHCII^−/− defines background. **i**, Representative flow cytometry plots of MHCII:non-CLIP on immature mTECs from *Ctsl*^+/+ (*n* = 13) and *Ctsl*^ΔTEC mice (*n* = 13); MHCII^−/− defines background. **j**, CD4SP percentages ± SEM in *Ctsl*^+/+ mice (*n* = 10 or 6) or *Ctsl*^−/− mice (*n* = 9 or 7) reconstituted with bone marrow from MHCII^+/+ or MHCII^−/− mice. **k**, CD4SP percentages ± SEM in *Ctsl*^+/+ mice (*n* = 3 or 4) or *Ctsl*^−/− mice (*n* = 3 each) on wild-type or C2ta^kd transgenic background. **l**, Representative flow cytometry plots of thymocyte subsets and frequency ± SEM of CD4SPs in *Ctsl*^+/+ B2m^−/− (*n* = 9) and *Ctsl*^ΔTEC B2m^−/− mice (*n* = 13); P < 0.0001. **m**, Representative flow cytometry analysis and MFI ± SEM relative to *Ctsl*^+/+ cells of CD5 expression on TCRβ^int CD69⁺ DPs from *Ctsl*^+/+ B2m^−/− (*n* = 9) and *Ctsl*^ΔTEC B2m^−/− mice (*n* = 13). P-values: (a) by 2-way ANOVA and Sidak's test for multiple comparisons; (b-g) and (j-m) by Student's two-tailed t-test.

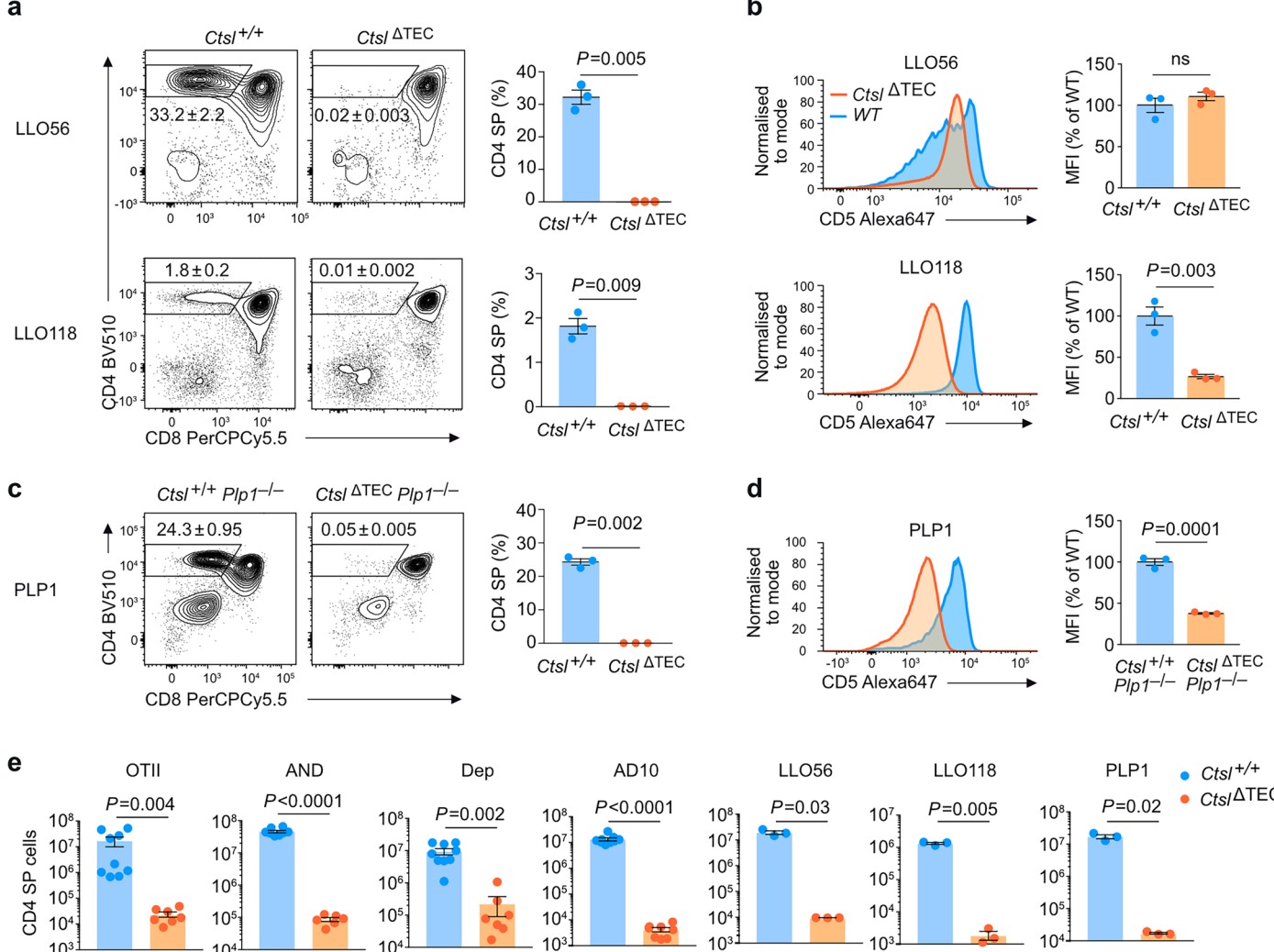

**Extended Data Fig. 2 | Ctsl is essential for positive selection of multiple MHCII-restricted transgenic TCRs. a**, Representative flow cytometry plots of thymocyte subsets in *Ctsl*[+/+] and *Ctsl*[ΔTEC] mice reconstituted with bone marrow (BM) from the indicated *Rag1*[−/−] TCR transgenic donors and corresponding percentages ± SEM of CD4SPs. (*n* = 3 *Ctsl*[+/+] and 3 *Ctsl*[ΔTEC] for *LLO56*; *n* = 3 and 3 for *LLO118*). **b**, Representative flow cytometry plots and MFI ± SEM of CD5 expression in DP thymocytes from BM chimeras as in (**a**), relative to cells selected in *Ctsl*[+/+] chimeras. **c**, Representative flow cytometry plots of thymocyte subsets in *Ctsl*[+/+] *Plp1*[−/−] (*n* = 3) and *Ctsl*[ΔTEC] *Plp1*[−/−] mice (*n* = 3) reconstituted with BM from the *PLP1* TCR transgenic donors and corresponding percentages ± SEM of CD4SPs. **d**, Representative flow cytometry plots and MFI ± SEM of CD5 expression in DP thymocytes from BM chimeras as in (**c**), relative to cells selected in *Ctsl*[+/+] *Plp1*[−/−] chimeras. **e**, CD4SP numbers ± SEM in *Ctsl*[+/+] and *Ctsl*[ΔTEC] mice (or *Ctsl*[+/+] *Plp1*[−/−] and *Ctsl*[ΔTEC] *Plp1*[−/−] mice in case of *PLP1*) reconstituted with BM from the indicated *Rag1*[−/−] TCR transgenic donors as in main Fig. 2a or **a** or **c**, respectively). All P-values by Student's two-tailed t-test.

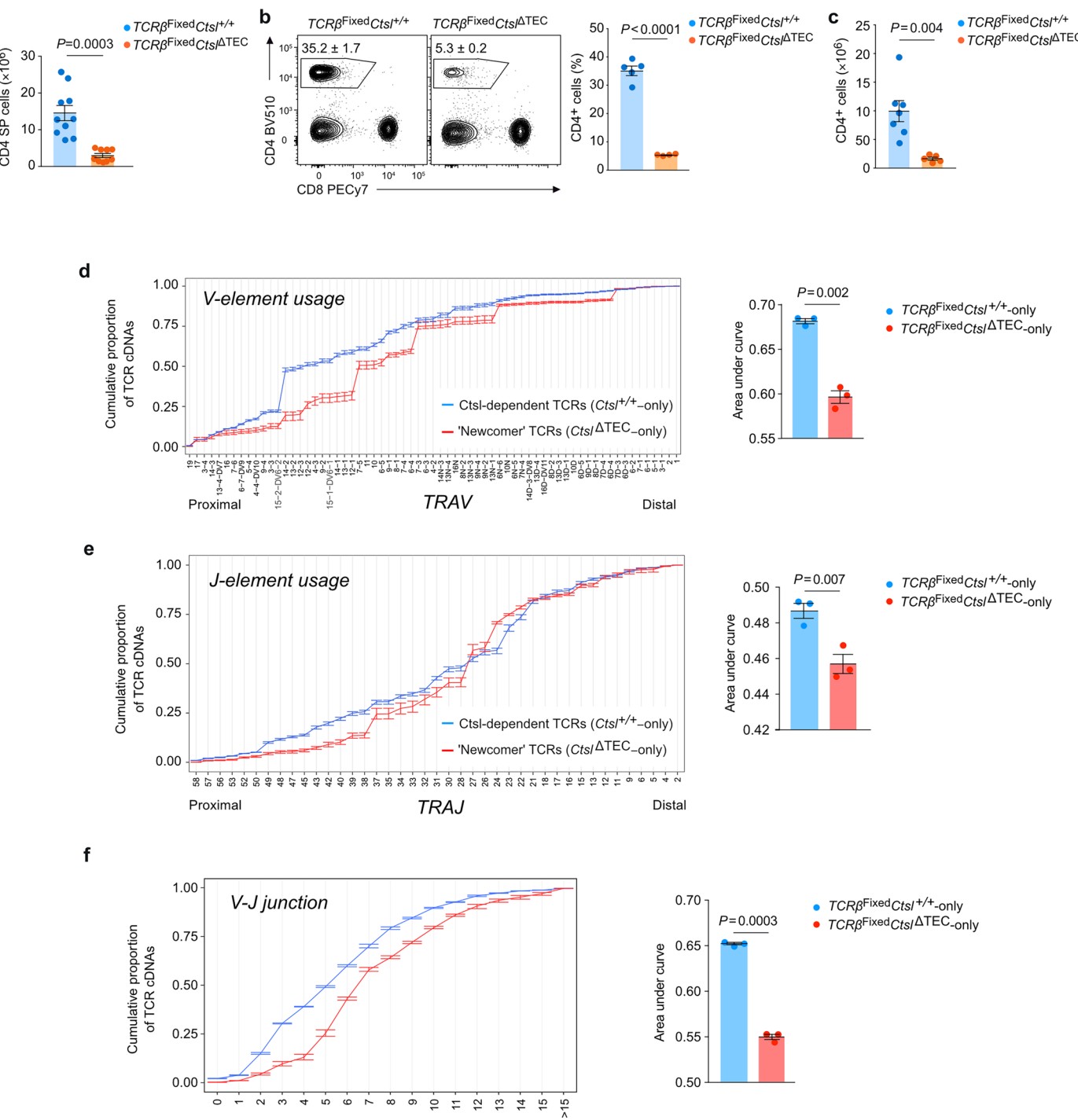

**Extended Data Fig. 3 | Ctsl-deficiency results in an altered repertoire in**
**_TCRβ_<sup>Fixed</sup> mice. a**, CD4SP cell number ± SEM in _TCRβ_<sup>Fixed</sup> _Ctsl_<sup>+/+</sup> (_n_ = 10) and
_TCRβ_<sup>Fixed</sup> _Ctsl_<sup>ΔTEC</sup> mice (_n_ = 9). **b**, Representative flow cytometry plots of CD4
versus CD8 and percentage ± SEM of CD4<sup>+</sup> T cells in lymph nodes from _TCRβ_<sup>Fixed</sup>
_Ctsl_<sup>+/+</sup> (_n_ = 5) and _TCRβ_<sup>Fixed</sup> _Ctsl_<sup>ΔTEC</sup> mice (_n_ = 4). **c**, Cell number ± SEM of CD4<sup>+</sup>
T cells in lymph nodes from _TCRβ_<sup>Fixed</sup> _Ctsl_<sup>+/+</sup> (_n_ = 7) and _TCRβ_<sup>Fixed</sup> _Ctsl_<sup>ΔTEC</sup> mice
(_n_ = 5). **d**, (left) V-element (_Trav_ gene) usage in recurrent TCRα clones detected
exclusively in _Fixed-β Ctsl_<sup>+/+</sup> TCRα inventories (n = 3) (blue; 'Ctsl<sup>+/+</sup>-only' Ctsl-
dependent TCRs), or exclusively in _Fixed-β Ctsl_<sup>ΔTEC</sup> TCRα inventories (n = 3) (red;
'_Ctsl_<sup>ΔTEC</sup>-only' newcomer TCRs). Curves show the cumulative proportion of
TCR cDNAs using the respective _Trav_ segment ordered from proximal to distal.
(right) Area under the curve (Mean ± SEM), determined by dividing the sum of

the cumulative proportions by the number of distinct Trav segments detected.
**e**, (left) J-element (_Traj_ gene) usage in 'Ctsl<sup>+/+</sup>-only' TCRα clones or '_Ctsl_<sup>ΔTEC</sup>-only'
newcomer TCRα clones as in (**d**). (right) Area under the curve (Mean ± SEM),
determined by dividing the sum of the cumulative proportions by the number
of distinct Traj segments detected. **f**, (left) Cumulative proportion of TCRα
cDNAs whose V-J junction was modified by deletion or addition of the number of
nucleotides specified on the x-axis. (right) Area under the curve (Mean ± SEM),
determined by dividing the sum of the cumulative proportions by the number
of nucleotide deletion/addition categories (17 possibilities in total). Data in d-f are
from n = 3 biological replicates per genotype; each replicate represents pooled
cells from 2 or 3 mice. All P-values by Student's two-tailed t-test.

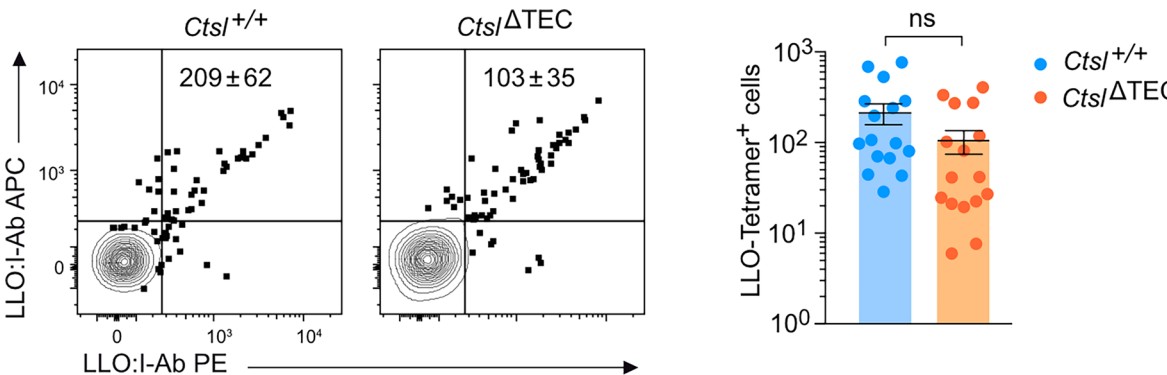

**Extended Data Fig. 4 | Ctsl-deficiency does not affect the number of LLO-Tet+ CD4SPs.** (left) Representative flow cytometry plots and total number ± SEM of LLO$_{190-201}$-specific CD4SPs in the thymus of $Ctsl^{+/+}$ ($n = 16$) and $Ctsl^{\Delta TEC}$ mice ($n = 16$) Dot plots are gated on dump-negative (CD11b⁻CD11c⁻B220⁻F4/80⁻) CD4⁺CD8⁻ cells after magnetic enrichment of Tet⁺ cells. P-value by Student's two-tailed t-test.

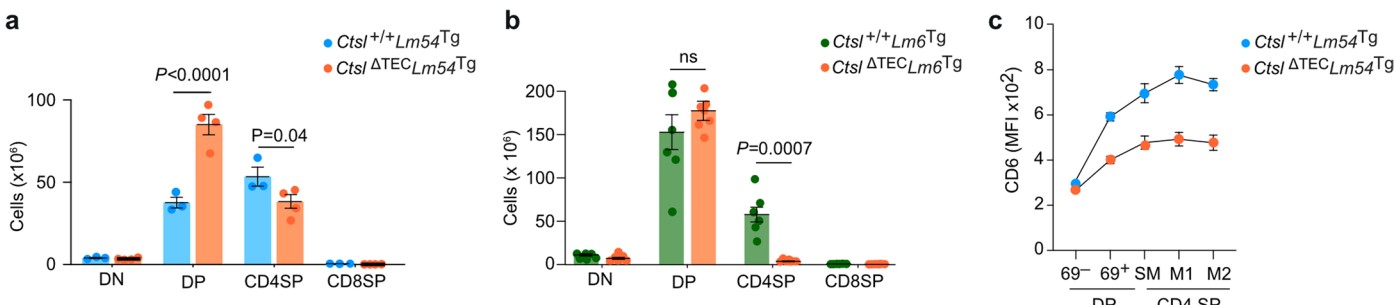

**Extended Data Fig. 5 | Ctsl calibrates the positively selecting signal strength.** **a**, Thymocyte subset cell numbers (Mean ± SEM) in *Ctsl*$^{+/+}$*Rag*1$^{-/-}$*Lm54*$^{Tg}$ (n = 3) and *Ctsl*$^{ΔTEC}$*Rag*1$^{-/-}$*Lm54*$^{Tg}$ mice (n = 4). **b**, Thymocyte subset cell numbers (Mean ± SEM) in *Ctsl*$^{+/+}$*Rag*1$^{-/-}$*Lm6*$^{Tg}$ (n = 6) and *Ctsl*$^{ΔTEC}$*Rag*1$^{-/-}$*Lm6*$^{Tg}$ mice

(n = 7). **c**, CD6 surface expression (MFI ± SEM) at consecutive DP and CD4SP differentiation stages in *Ctsl*$^{+/+}$*Rag*1$^{-/-}$*Lm54*$^{Tg}$ (n = 4) and *Ctsl*$^{ΔTEC}$*Rag*1$^{-/-}$*Lm54*$^{Tg}$ mice (n = 5), assessed by flow cytometry. P-value by Student's two-tailed t-test.

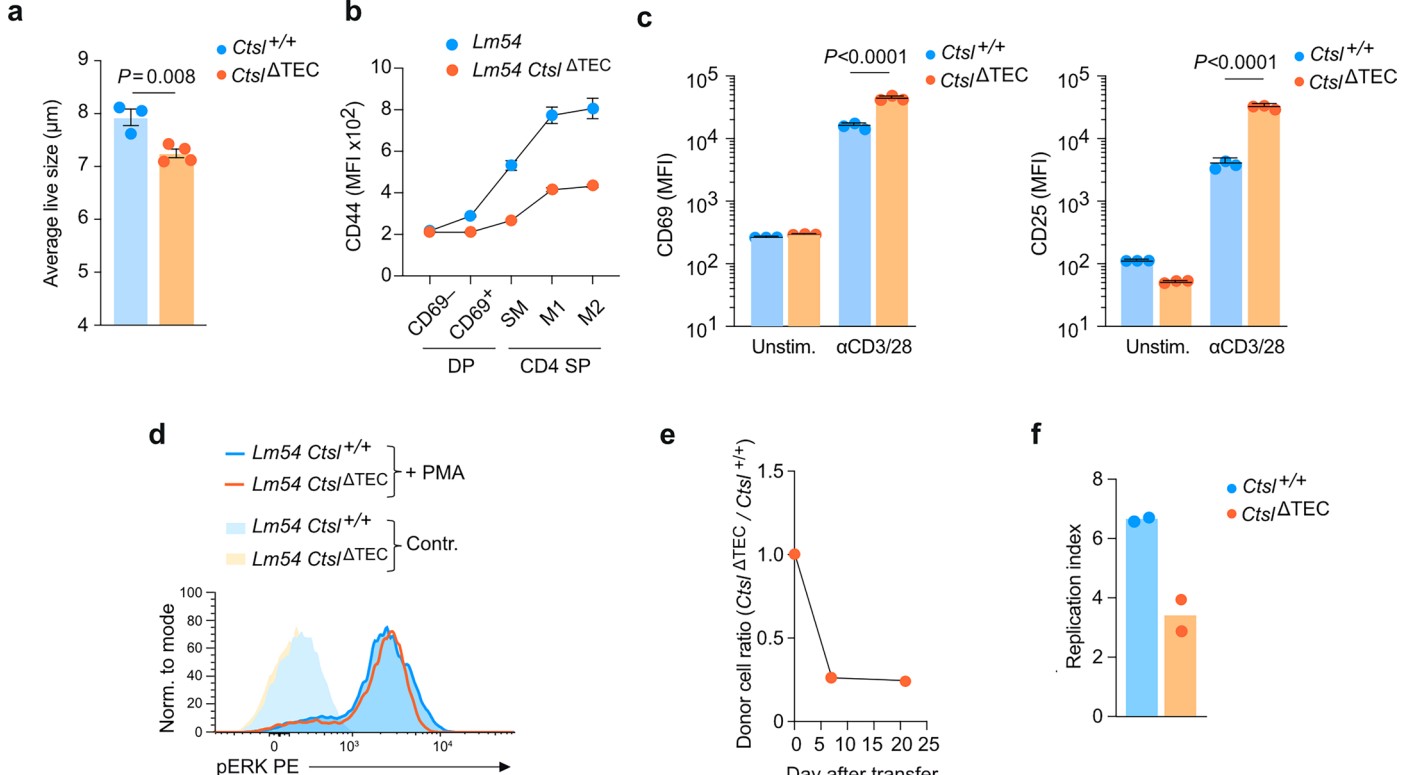

**Extended Data Fig. 6 | Selection in the absence of Ctsl causes aberrant functional tuning and impaired homeostatic fitness of CD4⁺ T cells. a**, Cell size (Mean ± SEM) of M2 thymocytes in $Ctsl^{+/+}Rag1^{-/-}Lm54^{Tg}$ ($n = 3$) and $Ctsl^{\Delta TEC}Rag1^{-/-}Lm54^{Tg}$ mice ($n = 4$). **b**, CD44 surface expression (MFI ± SEM) at consecutive DP and CD4SP differentiation stages in $Ctsl^{+/+}Rag1^{-/-}Lm54^{Tg}$ ($n = 4$) and $Ctsl^{\Delta TEC}Rag1^{-/-}Lm54^{Tg}$ mice ($n = 5$); assessed by flow cytometry. **c**, CD69 and CD25 expression (MFI ± SEM) in polyclonal M2 thymocytes from $Ctsl^{\Delta TEC}$ (n = 3) or $Ctsl^{+/+}$ mice (n = 3) after 40 h culture with or without (unstim.) plate-bound antibodies to CD3 and CD28; assessed by flow cytometry. **d**, Representative

flow cytometry plots of intracellular phospho-ERK staining of M2 thymocytes from $Ctsl^{+/+}Rag1^{-/-}Lm54^{Tg}$ ($n = 3$) and $Ctsl^{\Delta TEC}Rag1^{-/-}Lm54^{Tg}$ mice (n = 3) after 5 min stimulation with PMA. Controls without stimulation define background. **e**, Donor cell ratio (Mean) in the spleen of 4.5 Gy-irradiated recipients ($n = 2$) after transfer of a 1:1 mixture of CTV-labelled $Ctsl^{+/+}$ and $Ctsl^{\Delta TEC}$ M2 CD4SPs; assessed by flow cytometry. **f**, Replication index of donor cells on day 21 after transfer as in (e) assessed with FlowJo_v10.9.0 software, assessed by flow cytometry on the basis of CTV-dilution profiles. P-values by Student's two-tailed t-test.

# Reporting Summary

## Statistics

For all statistical analyses, confirm that the following items are present in the figure legend, table legend, main text, or Methods section.

| n/a | Confirmed | |
|---|---|---|
| ☐ | ☒ | The exact sample size (*n*) for each experimental group/condition, given as a discrete number and unit of measurement |
| ☐ | ☒ | A statement on whether measurements were taken from distinct samples or whether the same sample was measured repeatedly |
| ☐ | ☒ | The statistical test(s) used AND whether they are one- or two-sided *Only common tests should be described solely by name; describe more complex techniques in the Methods section.* |
| ☒ | ☐ | A description of all covariates tested |
| ☐ | ☒ | A description of any assumptions or corrections, such as tests of normality and adjustment for multiple comparisons |
| ☐ | ☒ | A full description of the statistical parameters including central tendency (e.g. means) or other basic estimates (e.g. regression coefficient) AND variation (e.g. standard deviation) or associated estimates of uncertainty (e.g. confidence intervals) |
| ☐ | ☒ | For null hypothesis testing, the test statistic (e.g. *F*, *t*, *r*) with confidence intervals, effect sizes, degrees of freedom and *P* value noted *Give P values as exact values whenever suitable.* |
| ☒ | ☐ | For Bayesian analysis, information on the choice of priors and Markov chain Monte Carlo settings |
| ☒ | ☐ | For hierarchical and complex designs, identification of the appropriate level for tests and full reporting of outcomes |
| ☒ | ☐ | Estimates of effect sizes (e.g. Cohen's *d*, Pearson's *r*), indicating how they were calculated |

*Our web collection on statistics for biologists contains articles on many of the points above.*

## Software and code

Policy information about availability of computer code

| Data collection | Flow cytometry data was acquired using FACSDiva v6.2 (BD Bioscience). Microscopy images were acquired with LAS X Office v1.4.6 (Leica microsystems). |
|---|---|
| Data analysis | All analyses are described in the relevant section of the Methods. Flow cytometry data were analysed with FlowJo v10.9.0. TCR analyses: Raw data were de-multiplexed and FASTQ files for each sample were generated using the bcl2fastq software (Illumina).The CLC Genomics Workbench software (v23.0.3) provided by Qiagen was used to generate clonotype reads. TCRα sequencing data were analysed with RStudio (2024.04.2+764), using the R packages iNEXT (Versions 2.0.20 and 3.0.1) and abdiv (Version 0.2.0). An executable capsule containing code and bulk TCRseq data is available at https://doi.org/10.24433/CO.5359169.v1. RNAseq: Expression quantification was performed using kallisto (version 0.48) with Ensembl release version 106 for M. musculus. In R/Bioconductor, expression data were collapsed from isoform to gene level for downstream processing. Differential expression was assessed using DESeq2 (version 1.36). Gene Set Enrichment analyses were conducted using fgsea (version 1.22). Statistical analyses were performed with GraphPad Prism v9. |

For manuscripts utilizing custom algorithms or software that are central to the research but not yet described in published literature, software must be made available to editors and reviewers. We strongly encourage code deposition in a community repository (e.g. GitHub). See the Nature Portfolio guidelines for submitting code & software for further information.

## Data

Policy information about availability of data

All manuscripts must include a data availability statement. This statement should provide the following information, where applicable:

- Accession codes, unique identifiers, or web links for publicly available datasets
- A description of any restrictions on data availability
- For clinical datasets or third party data, please ensure that the statement adheres to our policy

> Sequencing data from this study have been deposited at the GEO and will be publicly available from the date of publication. The accession numbers are GSE269202 for bulk TCRseq data (relating to Figs. 3c-g and 5b-e) and GSE269197 for RNAseq data (relating to Figs. 7 a,b,f). Source data referring to bulk and single-cell TCRseq datasets and bulk RNAseq are provided with this paper.

## Research involving human participants, their data, or biological material

Policy information about studies with human participants or human data. See also policy information about sex, gender (identity/presentation), and sexual orientation and race, ethnicity and racism.

| | |
|---|---|
| Reporting on sex and gender | n.a. |
| Reporting on race, ethnicity, or other socially relevant groupings | n.a. |
| Population characteristics | n.a. |
| Recruitment | n.a. |
| Ethics oversight | n.a. |

Note that full information on the approval of the study protocol must also be provided in the manuscript.

# Field-specific reporting

Please select the one below that is the best fit for your research. If you are not sure, read the appropriate sections before making your selection.

☒ Life sciences ☐ Behavioural & social sciences ☐ Ecological, evolutionary & environmental sciences

For a reference copy of the document with all sections, see nature.com/documents/nr-reporting-summary-flat.pdf

# Life sciences study design

All studies must disclose on these points even when the disclosure is negative.

| | |
|---|---|
| Sample size | In a single experiment, at least three biological replicates were used, with the exception of stimulated samples in Fig.6f. As the present study was of exploratory character, no sample size calculation was performed. In general, for these types of experiments, we aim for a group size of 7 each (Allgoewer and Mayer, 2007). When smaller sample sizes were used, the sample size was determined based on previous work or the experience from the first experimental replicate. Moreover, data was collected in repeated independent experiments. |
| Data exclusions | No data were excluded. |
| Replication | All major experiments were repeated at least once (as detailed in the figure legends) and replications were successful. The replication number of each experiment is included in the legends. |
| Randomization | Randomization is not relevant to this study as samples in each experiment were treated uniformly and the same data analysis procedure was applied to all samples of the same experiment. |
| Blinding | Investigators were not blinded in this study because all results presented are based on quantitative analysis, which is not subject to human biases. |

# Reporting for specific materials, systems and methods

We require information from authors about some types of materials, experimental systems and methods used in many studies. Here, indicate whether each material, system or method listed is relevant to your study. If you are not sure if a list item applies to your research, read the appropriate section before selecting a response.

## Materials & experimental systems

| n/a | Involved in the study |
|-----|----------------------|
| ☐ | ☒ Antibodies |
| ☒ | ☐ Eukaryotic cell lines |
| ☒ | ☐ Palaeontology and archaeology |
| ☐ | ☒ Animals and other organisms |
| ☒ | ☐ Clinical data |
| ☒ | ☐ Dual use research of concern |
| ☒ | ☐ Plants |

## Methods

| n/a | Involved in the study |
|-----|----------------------|
| ☒ | ☐ ChIP-seq |
| ☐ | ☒ Flow cytometry |
| ☒ | ☐ MRI-based neuroimaging |

## Antibodies

| Antibodies used | Antibody, catalogue #, clone #, supplier |
|-----------------|------------------------------------------|
| | anti-CD28, #BE0015-1, 37.51, Bio X Cell |
| | anti-CD4 BV510, 100559, RM4-5, Biolegend |
| | anti-CD4 biotinylated, 100508, RM4-5, Biolegend |
| | anti-CD8α PeCy7, 100722 , 53-6.7, Biolegend |
| | anti-CD8α PerCPCy5.5, 100734, 53-6.7,  Biolegend |
| | anti-CD8α biotinylated, 100704, 53-6.7, Biolegend |
| | anti-CD326/Ep-CAM PeCy7, 118216, G8.8, Biolegend |
| | anti-Ly51 Alexa647, 108312, 6C3, Biolegend |
| | anti-Ly51 PE, 108308, 6C3, Biolegend |
| | anti-CD80 PE, 104708 , 16-10A1, Biolegend |
| | anti-CD5 Alexa647, 100614, 53-7.3, Biolegend |
| | anti-CD5 PE, 100608, 53-7.3, Biolegend |
| | anti-CD5 BV421, 100617, 53-7.3, Biolegend |
| | anti-TCRβ APC, 109212, H57-597, Biolegend |
| | anti-CD69 PE, 104508, H1.2F3, Biolegend |
| | anti-CD69 BV421, 104527, H1.2F3, Biolegend |
| | anti-CD69 PeCy7, 104512, H1.2F3, Biolegend |
| | anti-CD69 BV711, 104537, H1.2F3, Biolegend |
| | anti-H-2Kb FITC, 116505, AF6-88.5, Biolegend |
| | anti-H-2Kb BV786, 742863, AF6-88.5, BD |
| | anti-CD45.1 Alexa647, 110720, A20, Biolegend |
| | anti-CD45.1 FITC, 110706, A20, Biolegend |
| | anti-CD45.1 BV421, 110731, A20, Biolegend |
| | anti-CD45.2 Alexa647, 109818, 104, Biolegend |
| | anti-CD45.2 FITC, 109806, 104, Biolegend |
| | anti-CD45.2 BV421, 109832, 104, Biolegend |
| | anti-CD44 BV421, 103039, IM7, Biolegend |
| | anti-CD44 APC, 103012, IM7, Biolegend |
| | anti-CD25 PeCy7, 102016, PC61, Biolegend |
| | anti-CD62L FITC, 104406, MEL-14, Biolegend |
| | anti-CD62L APCCy7, 104428, MEL-14, Biolegend |
| | anti-TCRvα2 APCCy7, 127818, B20.1, Biolegend |
| | anti-TCRvα2 BV711, 743832, B20.1, BD |
| | anti-TCRvβ5 FITC, 139513, MR9-4, Biolegend |
| | anti-CD127/IL-7Rα PE, 135010, A7R34, Biolegend |
| | anti-CD127/IL-7Rα Alexa488, 135018, A7R34, Biolegend |
| | anti-CCR7 PE, 120106, 4B12, Biolegend |
| | anti-I-A/I-E APCCy7, 107628, M5/114.15.2, Biolegend |
| | anti-CLIP:I-Ab FITC, sc-53946 FITC, 15G4, Santa Cruz Biotechnology |
| | anti-nonCLIP:I-Ab APC, no cat#, BP107.2.2, hybridoma gift from A. Rudensky (Memorial Sloan Kettering Cancer Center) |
| | anti-Ea52-68:I-Ab FITC, Y-Ae, hybridoma gift from B. Kyeswki (German Cancer Research Center) |
| | anti-Nur77 PE, 12-5965-82, 12.14,  eBioscience |
| | anti-Bcl2 PE, 633508, BCL/10C4, Biolegend |
| | anti-CD11b PeCy7, 101216, M1/70, Biolegend |
| | anti-CD11b biotinylated, 101204, M1/70, Biolegend |
| | anti-CD11c PeCy7, 117318, N418, Biolegend |
| | anti-CD11c biotinylated, 117304, N418, Biolegend |
| | anti-B220 PeCy7, 103222, RA3-6B2, Biolegend |
| | anti-B220 biotinylated, 103204, RA3-6B2, Biolegend |
| | anti-F4/80 PeCy7, 123114, BM8, Biolegend |
| | anti-F4/80 biotinylated, 123106, BM8, Biolegend |
| | anti-Gr1 biotinylated, 108404, RB6-8C5, Biolegend |
| | goat anti-mouse Cathepsin L polyclonal IgG, AF1515, R&D |
| | mouse anti-goat HRP-conjugated polyclonal IgG, RRID: AB_2339057, 205-035-108, Jackson ImmunoResearch |
| | mouse anti-mouse β-actin,A2228 , AC-15, Sigma |

rabbit anti-mouse HRP-conjugated polyclonal IgG, P026002-2, P0260, Agilent Dako

Validation

The antibodies used in this study were used according to the manufacturere's recommendation. Validation was based on the description provided on the manufacturers' homepage Prior to use in experiments, all fluorochrome-labelled antibodies were titrated using a dilutio series on respective antigen positive cells to determine optimal working dilution and performance.
Antibody (clone) Validation:
anti-CD28,https://bioxcell.com/invivomab-anti-mouse-cd28-be0015-1
anti-CD4 (RM4-5) https://www.biolegend.com/de-de/products/brilliant-violet-510-anti-mouse-cd4-antibody-7991
https://www.biolegend.com/de-de/products/biotin-anti-mouse-cd4-antibody-479
anti-CD8α (53-7.3) https://www.biolegend.com/de-de/products/pe-cyanine7-anti-mouse-cd8a-antibody-1906
https://www.biolegend.com/de-de/search-results/percp-cyanine5-5-anti-mouse-cd8a-antibody-4255
https://www.biolegend.com/de-de/products/biotin-anti-mouse-cd8a-antibody-152?GroupID=BLG6765
anti-CD326/Ep-CAM (G8.8) https://www.biolegend.com/de-at/products/pe-cyanine7-anti-mouse-cd326-ep-cam-antibody-5303
anti-Ly51 (6C3) https://www.biolegend.com/de-de/products/alexa-fluor-647-anti-mouse-ly-51-antibody-3310
https://www.biolegend.com/de-de/products/pe-anti-mouse-ly-51-antibody-178
anti-CD80 (16-10A1) https://www.biolegend.com/de-de/products/pe-anti-mouse-cd80-antibody-43?GroupID=BLG1851
anti-CD5 (53-7.3) https://www.biolegend.com/de-de/products/alexa-fluor-647-anti-mouse-cd5-antibody-3199
https://www.biolegend.com/de-de/products/pe-anti-mouse-cd5-antibody-160
https://www.biolegend.com/de-de/products/brilliant-violet-421-anti-mouse-cd5-antibody-8585
anti-TCRβ (H57-597) https://www.biolegend.com/de-de/products/apc-anti-mouse-tcr-beta-chain-antibody-268
anti-CD69 (H1.2F3)https://www.biolegend.com/de-de/products/pe-anti-mouse-cd69-antibody-265
https://www.biolegend.com/de-de/products/brilliant-violet-421-anti-mouse-cd69-antibody-7358
https://www.biolegend.com/de-de/products/pe-cyanine7-anti-mouse-cd69-antibody-3168
https://www.biolegend.com/de-de/products/brilliant-violet-711-anti-mouse-cd69-antibody-12139
anti-H-2Kb (F6-88.5) https://www.biolegend.com/de-de/products/fitc-anti-mouse-h-2kb-antibody-1748
anti-H-2Kb (AF6-88.5)  https://www.bdbiosciences.com/en-au/products/reagents/flow-cytometry-reagents/research-reagents/single-color-antibodies-ruo/bv786-mouse-anti-mouse-h-2kb.742863
anti-CD45.1 (A20) https://www.biolegend.com/de-de/products/alexa-fluor-647-anti-mouse-cd45-1-antibody-3104
https://www.biolegend.com/de-de/products/fitc-anti-mouse-cd45-1-antibody-198
https://www.biolegend.com/de-de/products/brilliant-violet-421-anti-mouse-cd45-1-antibody-7255
anti-CD45.2 (104) https://www.biolegend.com/de-de/products/alexa-fluor-647-anti-mouse-cd45-2-antibody-3107
https://www.biolegend.com/de-de/products/fitc-anti-mouse-cd45-2-antibody-6
https://www.biolegend.com/de-de/products/brilliant-violet-421-anti-mouse-cd45-2-antibody-7328
anti-CD44 (IM7) https://www.biolegend.com/de-de/products/brilliant-violet-421-anti-mouse-human-cd44-antibody-7225
https://www.biolegend.com/de-de/products/apc-anti-mouse-human-cd44-antibody-312
anti-CD25 (PC61) https://www.biolegend.com/de-de/products/pe-cyanine7-anti-mouse-cd25-antibody-1929
anti-CD62L (MEL-14) https://www.biolegend.com/de-de/products/fitc-anti-mouse-cd62l-antibody-384
https://www.biolegend.com/de-de/products/apc-cyanine7-anti-mouse-cd62l-antibody-3934
anti-TCRvα2 (B20.1) https://www.biolegend.com/de-de/products/apc-cyanine7-anti-mouse-tcr-valpha2-antibody-7016
https://www.bdbiosciences.com/en-de/products/reagents/flow-cytometry-reagents/research-reagents/single-color-antibodies-ruo/bv711-rat-anti-mouse-v-2-tcr.743832
anti-CD127/IL-7Rα (A7R34) https://www.biolegend.com/de-de/products/pe-anti-mouse-cd127-il-7ralpha-antibody-6190
https://www.biolegend.com/de-de/products/alexa-fluor-488-anti-mouse-cd127-il-7ralpha-antibody-6194
anti-CCR7 (4B12) https://www.biolegend.com/de-de/products/pe-anti-mouse-cd197-ccr7-antibody-2799
anti-I-A/I-E (M5/114.15.2) https://www.biolegend.com/de-de/products/apc-cyanine7-anti-mouse-i-a-i-e-antibody-5966
anti-CLIP:I-Ab (15G4) https://www.scbt.com/p/mhc-class-ii-antibody-15g4
anti-Nur77  (12.14) https://www.thermofisher.com/antibody/product/Nur77-Antibody-clone-12-14-Monoclonal/12-5965-82
anti-Bcl2 (BCL/10C4) https://www.biolegend.com/de-de/products/pe-anti-bcl-2-antibody-6466
anti-CD11b (M1/70) https://www.biolegend.com/de-de/products/pe-cyanine7-anti-mouse-human-cd11b-antibody-1921
https://www.biolegend.com/de-de/products/biotin-anti-mouse-human-cd11b-antibody-346
anti-CD11c (N418) https://www.biolegend.com/de-de/products/pe-cyanine7-anti-mouse-cd11c-antibody-3086
https://www.biolegend.com/de-de/products/biotin-anti-mouse-cd11c-antibody-1814
anti-B220 (RA3-6B2)https://www.biolegend.com/de-de/products/pe-cyanine7-anti-mouse-human-cd45r-b220-antibody-1930
https://www.biolegend.com/de-at/products/biotin-anti-mouse-human-cd45r-b220-antibody-444
anti-F4/80 (BM8) https://www.biolegend.com/de-at/products/pe-cyanine7-anti-mouse-f4-80-antibody-4070
https://www.biolegend.com/de-at/products/biotin-anti-mouse-f4-80-antibody-4066
anti-Gr1 (RB6-8C5)
goat anti-mouse Cathepsin L polyclonal IgG (AF1515)https://www.rndsystems.com/products/mouse-rat-cathepsin-l-antibody_af1515
mouse anti-goat HRP-conjugated polyclonal IgG (205-035-108) https://www.jacksonimmuno.com/catalog/products/205-035-108
mouse anti-mouse β-actin (AC-15) https://www.sigmaaldrich.com/EE/en/product/sigma/a2228
rabbit anti-mouse HRP-conjugated polyclonal IgG (P0260) https://www.agilent.com/en/product/specific-proteins/elisa-kits-accessories/rabbit-anti-mouse-immunoglobulins-hrp-solid-phase-absorbed-2717115

# Animals and other research organisms

Policy information about studies involving animals; ARRIVE guidelines recommended for reporting animal research, and Sex and Gender in Research

Laboratory animals

All mice used were on a C57BL/6J background except for F1 B6 x BALB/c in Fig. 1g and were maintained under specific pathogen-free conditions in individually ventilated cages at an ambient temperature of 22°C and 55% humidity with standard light cycle conditions. Ctslfl/fl, Ctsl–/–, TCR-Dep, C2TAkd, TCR-PLP1, TCR-LLO56 and TCR-LLO118, TCR-AND and TCR-AD10, TCR-OT-II, Foxn1-Cre, MHCI–/– (B2m–/–), MHCII–/– (H2-Ab1–/–), Rag1–/–, , Plp1–/–, TCRα–/– and Foxp3GFP reporter (DEREG) mice were previously reported. Lm6, Lm54 and Fixed-β transgenic mice were generated by injection of linearized DNA (encoding the V(D)J regions of the respective TCRs

identified by single cell TCR sequencing) into pronuclei of C57BL/6 zygotes. All phenotypic analyses were performed in mice of 8–12 weeks of age, unless otherwise indicated.

| | |
|---|---|
| Wild animals | No wild animals were used in this study |
| Reporting on sex | Sex-matched controls were used whenever possible. Results presented include both female and male animals, as no sex-specific effects were observed in the tissues analysed. |
| Field-collected samples | no field collected samples were used in the study. |
| Ethics oversight | Animal studies and procedures were approved by local authorities (Regierung von Oberbayern) and performed under protocol #Vet_02-22-66. |

Note that full information on the approval of the study protocol must also be provided in the manuscript.

# Plants

| | |
|---|---|
| Seed stocks | n.a. |
| Novel plant genotypes | n.a. |
| Authentication | n.a. |

# Flow Cytometry

## Plots

Confirm that:

☒ The axis labels state the marker and fluorochrome used (e.g. CD4-FITC).

☒ The axis scales are clearly visible. Include numbers along axes only for bottom left plot of group (a 'group' is an analysis of identical markers).

☒ All plots are contour plots with outliers or pseudocolor plots.

☒ A numerical value for number of cells or percentage (with statistics) is provided.

## Methodology

| | |
|---|---|
| Sample preparation | To obtain single-cell suspensions from thymi, lymph nodes or spleens, organs were dissected, smashed and filtered through a 150μm cell strainer. Splenocytes were additionally subjected to red blood cell lysis by incubation in BD Pharm Lyse™ lysing solution for 5 min at RT. <br> To obtain single-cell suspensions from the bone marrow, femur and tibia bones were dissected, smashed with the use of mortar and pestle and filtered through a 150μm cell strainer. Cells were then subjected to red blood cell lysis by incubation in BD Pharm Lyse™ lysing solution for 1 min at RT. <br> For isolation of TECs, thymi were dissected and cut into pieces. Thymocytes were mechanically released by pipetting up and down and the supernatant containing thymocytes was discarded. The thymus fragments were digested with liberase™ (0.5 U/ml; Roche) and DNase I (10 mg/ml; Roche) at 37°C in three consecutive rounds of 15 min. Cells were filtered though a 80μm cell strainer and washed. Cell pellets were re-suspended in 1 ml of high-density Percoll™ (ρ=1.115; GE Healthcare) and overlaid with 1 ml of low-density Percoll (ρ=1.055), followed by a layer of 1 ml RPMI (Gibco). The gradient was centrifuged at 1350 g for 30 min at 4 °C (w/o brake). The upper interphase, containing the low-density cell fraction, was harvested, washed and subjected to CD45 MACS depletion, using CD45 MicroBeads (Miltenyi Biotech). |
| Instrument | Sorts were performed on a FACSAriaFusion sorter (BD). Samples were acquired on a FACSCantoII or LSRFortessa (BD). |
| Software | Analysis of Flow Cytometry data was performed with FlowJo v10.9.0. |
| Cell population abundance | Post-sort purity was determined by re-acquiring and recording aliquots of the sorted populations. Sorted populations were typically 90-99% pure. |

Gating strategy

For thymus/LN samples: 1. FSC-A/SSC-A to exclude cell debris; 2. FSC-A/FSC-H to retain only singlets; 3. CD4/CD8
For tetramer-stained cells: 1. FSC-A/SSC-A to exclude cell debris; 2. FSC-A/FSC-H to retain only singlets; 3. dump/CD3 to exclude tetramer non specific staining; 4. CD4/CD8; 5. TetAPC/TetPE
For TECs: 1. DAPI/CD45 to exclude dead and hematopoietic cells; 2. EpCAM/FSC-A to retain only big cells of the epithelial lineage; 3. Ly51/MHCII to distinguish cTECs and mTECs; 4. CD80/MHCII to distinguish mTEClo and mTEChi
Supplementary Fig S2 shows example gating strategies.

☒ Tick this box to confirm that a figure exemplifying the gating strategy is provided in the Supplementary Information.

