## [Peer Review File · Nature Immunology]

Cathepsin L-dependent positive selection shapes clonal composition and functional fitness of CD4⁺ T cells

Corresponding Author: Professor Ludger Klein

Version 0:

Decision Letter:

22nd Jun 2024

Dear Dr. Klein,

Your Article, "Cathepsin L-Dependent Positive Selection Shapes Clonal Composition and Functional Fitness of CD4 T Cells" has now been seen by 3 referees. While we find your work of considerable potential interest, the reviewers have raised substantial concerns that must be addressed. As such, we cannot accept the current version of the manuscript for publication, but would be happy to consider a revised version that addresses these concerns, as long as novelty is not compromised in the interim.

Please revise the manuscript to address all issues raised by the referees. We believe it is important to better define the CTSL-dependent peptidome and its role in the regulation of the CD4 T cell repertoire and address the issues raised regarding the role of other thymic cell subsets and negative selection, as outlined by the reviewers. At resubmission, please include a point-by-point "Response to referees" detailing how you have addressed each referee comment (please specify the page and figure number where the new data can be found in the revised manuscript and please highlight the changes in the manuscript as well). This response will be sent back to the referees along with the revised manuscript.

In addition, please include a revised version of any required reporting checklist. It will be available to referees (and, potentially, statisticians) to aid in their evaluation if the manuscript goes back for peer review. A revised checklist is essential for re-review of the paper. The Reporting Summary can be found here:

Extended Data figures and tables are online-only (appearing in the online PDF and full-text HTML version of the paper), peer-reviewed display items that provide essential background to the Article but are not included in the printed version of the paper due to space constraints or being of interest only to a few specialists. A maximum of ten Extended Data display items (figures and tables) is typically permitted. When re-submitting your manuscript, please ensure that any supplementary figures and tables that are more critical to the manuscript's conclusions are converted to Extended data to increase these data's visibility.

Link Redacted

Note: This URL links to your confidential home page and associated information about manuscripts you may have submitted, or that you are reviewing for us. If you wish to forward this email to co-authors, please delete the link to your

homepage.

We hope to receive a suitably revised manuscript within 6 months. If you cannot send it within this time, please let us know. We will be happy to consider your revision so long as nothing similar has been accepted for publication at Nature Immunology or published elsewhere.

Nature Immunology is committed to improving transparency in authorship. As part of our efforts in this direction, we are now requesting that all authors identified as 'corresponding author' on published papers create and link their Open Researcher and Contributor Identifier (ORCID) with their account on the Manuscript Tracking System (MTS), prior to acceptance. ORCID helps the scientific community achieve unambiguous attribution of all scholarly contributions. You can create and link your ORCID from the home page of the MTS by clicking on 'Modify my Springer Nature account'. For more information please visit please visit www.springernature.com/orcid.

Thank you for the opportunity to review your work.

Sincerely,

Ioana Staicu, Ph.D.
Senior Editor
Nature Immunology

Tel: 212-726-9207
Fax: 212-696-9752
www.nature.com/ni

Reviewers' Comments:

Reviewer #1:

Remarks to the Author:

The understanding of the molecular mechanisms by which cortical thymic epithelial cells (cTECs) generate specialized self-peptides II complexes capable of positively selecting CD4 and CD8 T cells remains a topic of intense scrutiny. While seminal studies have underscored the involvement of lysosomal proteins, such as cathepsin L (Ctsl) and TTSP, in the selection of CD4 T cells, and the β 5t-containing thymoproteasome in the selection of CD8 T cells, the underlying mechanisms and the nature of the peptide-MHC ligandome expressed by these cells remain largely unknown. Although the functional consequences of β 5t-thymoproteasome-dependent selection on peripheral CD8 T cells have been investigated, the impact of Ctsl-dependent pathways on the TCR diversification and response of CD4 T cells remains elusive.

In this study, Petrozziello et al. investigated the TEC-intrinsic role of Ctsl in shaping T cell selection, TCR repertoire diversity, and T cell function. Previous studies have shown that Ctsl is highly expressed in cTECs, affects CD4 T cell selection, and suggests that it may regulate both the processing of the MHC-II invariant chain (Ii) and the generation of selecting MHC-II-associated self-peptides for CD4+ T cell development. However, the impact on TCR repertoire formation and T cell functionality remains unknown.

Using a novel Ctsl cKO mouse model, the authors demonstrated that TEC-specific deficiency in Ctsl affects the selection of CD4 T cells (FIG1). Through an extensive collection of BM-transplantations, they further showed that Ctsl-deficiency influences the selection of several MHC II-restricted TCR transgenics (FIG2). Using robust high-content TCR sequencing, they demonstrated that Ctsl-deficiency markedly restrains TCR diversity (FIG3). By analyzing the selection and response of endogenous antigen-specific CD4 T cells, they demonstrated a functional impairment in the response of antigen-specific selected in a Ctsl-deficient thymic niche (FIG4). Moreover, generating and studying new TCR transgenic mice from an inventory of Ctsl-dependent and independent TCR repertoires, they determined the impact of the Ctsl-dependent pathway on the selection and function of these TCR transgenic T cells (FIG5). Interestingly, even Ctsl-independent TCRs exhibited defects in their developmental program and function (FIG6), indicating that the absence of Ctsl affects not only the selection of a broad TCR repertoire but it has also a pervasive effect in the response of Ctsl-independent TCR clones.

This study has a significant merit for its detailed analysis of the consequences of Ctsl deficiency in T cell development, employing several advanced and complementary methodologies to demonstrate its impact. Apart of some comments on T cell analysis, the manuscript does not fully address, as discussed by the authors, the molecular analysis and mechanisms controlled by Ctsl in TECs. Although these experiments (e.g. identification of the peptidome controlled by Ctsl) are challenging, the study would benefit from a more in-depth examination of the molecular pathways to fully elucidate the role of Ctsl in T cell development. My specific comments are found below.

1. Impact of Cathepsin L on MHC II Processing and Peptidome Generation: Previous studies have shown that the thymus of mice with combined deficiencies in Ctsl and Ii generated fewer CD4 T cells compared to Ctsl and Ii single-deficient mice. This suggests that Ctsl may also contribute to the generation of specific selecting peptides for CD4 T cells. However, the

current results do not formally address whether Ctsl affects MHC II processing and/or peptidome generation. These are challenging points to address experimentally due to the complexity of proteomics and peptide-bound MHC II identification in rare cells such as cTECs. Nevertheless, an analysis of MHC II processing in sorted cTEC and mTEC populations may be provided in for example sorted cTEC and mTEC. Is there any possibility that the authors extended the analysis of peptidome, apart of the analysis with MHCII:non-CLIP Ab shown in Fig1F. These points and limitations should be attempted to be further enhanced and discussed.

2. The cellularity of thymic and splenic T cell subsets in control and Ctsl cKO mice is not shown in Figure 1. Including these data would provide a more comprehensive understanding of the effects of Ctsl deficiency on T cell populations.

3. The manuscript does not address the development of regulatory T cells (Tregs) and the maturation of SP4 cells in Ctsl cKO mice. Although Treg development is primarily dependent on mTECs, it would be interesting to examine how this process is influenced by purported altered positively selected CD4 T cells. Additionally, the analysis of peripheral T cell composition (naïve, effector, and memory subsets) in Ctsl cKO mice is limited and can be expanded. Moreover, do the mice develop any signs of disturbed immunotolerance with age? or the hyporesponsive state and imprinted abnormal selection of T cells in Ctsl cKO impairs the development of autoimmune response.

4. The conclusion on page 6 states, "These findings rendered a reduced complexity of the cTEC ligandome per se or 'excessive' clonal deletion unlikely explanations for the diminished CD4SP cell compartment in Ctsl Δ TEC mice, corroborating the idea that 'normal' positive selection of CD4 T cells requires 'private' peptides generated through processing by Ctsl." This statement needs clarification. The results might favor the hypothesis of a less complex and/or available cTEC ligandome. The authors should elucidate how their findings support or contradict this hypothesis.

5. The results presented in Figure 2 involving several MHC II-restricted TCR transgenics are thorough. However, similar to Figure 1, the cellularity of distinct thymic populations should be included. Additionally, it would be valuable to know if the authors performed similar control experiments with some MHC I-restricted TCR transgenics.

6- The hyporesponsive state of LLO-specific T cells and Cts-independent TCRs, as suggested later in the manuscript, should perhaps be expected to exist in the context of polyclonal CD4 T cells or in a TCR transgenic Ctsl-deficient setting (Fig2). Did the authors test the peripheral antigen-specific responses in TCR transgenics (Fig2)? Moreover, the homeostatic capacity of polyclonal peripheral CD4 T cells derived from Ctsl cKO and control mice can be examined in competitive homeostatic proliferation experiments, addressing these aspects in a more polyclonal setting.

Other points:

1. Can the authors comment on the high level of MHC II:clip in mTECs relatively to cTEC?

Reviewer #2:

Remarks to the Author:

This manuscript by Petrozziello et al makes use of a cTEC-restricted cathepsin-L (Ctsl) KO to investigate whether the unique set of peptides generated by cathepsin-L impacts positive selection and thus whether its absence alters the composition of the naïve CD4 T cell repertoire. Using an elegant set of approaches and clever mouse models, the authors show that Ctsl-TEC mice have a large reduction in maturing CD4 T cells, and provide compelling evidence that this reduction is not a result of increased negative selection, but reflects decreases in the fraction of CD4 T cells that are positively selected as a result of an altered self-immunopeptidome. They show that defects in positive selection hold for individual CD4+ T cell receptor (TCR) transgenic T cells, and that these defects alter the polyclonal CD4 T cell repertoire using TCR sequencing in a TCR β -chain fixed system. Finally, the authors show that CD4 T cells specific for the listeria LLO epitope are not altered in precursor frequency in absence of Ctsl expression, and hone in on two specific TCRs that are Ctsl-independent to ask whether loss of Ctsl impacts their function despite being successfully selected. Indeed, their work indicates that even TCR sequences that are Ctsl-'independent' ultimately have severe defects in their response to foreign antigen, presumably as a function of changes in thymic positive selection.

This is a very well-written and interesting article that provides a novel perspective on the role of the self-immunopeptidome on generating an effective T cell repertoire, a topic of key importance to better understanding how T cells are trained to be poised to respond effectively to non-self. The discussion relates the findings well to prior work, data figures are well-put together and analyses performed for the most part clear. While the data is already quite strong, some additional experiments and/or analyses of the data already in hand would strengthen the conclusions and perhaps better relate what was done at the polyclonal versus individual clonotype level with regard to the impact of loss of Ctsl expression in cTECs.

(1) The introduction is very well written and clear. However, it would be helpful to add a bit more detail on the role of cathepsin L in antigen presentation. This part in the introduction is very brief and yet central to the paper. What types of peptides are lost and what remains? Is anything known about the reduction in the complexity of the self-peptidome without it? It would be great to make the introduction relating to Ctsl more generally accessible for a broader immunology audience, and would also be an opportunity to introduce CLIP/MHCII antigen presentation protein processing, especially as anti-CLIP T cells are measured later on without much explanation as it relates to cathepsin L.

(2) From the analyses done as shown it is not addressed whether there is an affinity bias with regard to which T cells are positively selected or whether all T cells are equally impaired in absence of cTEC Ctsl expression. The authors interpret the

CD5 decreases in the TCR transgenic CD4 T cells as interactions with self-pMHC being weaker across the board (and make the intriguing point in the discussion that technical biases in how monoclonal TCRs were selected/identified might have led to them all being Ctsl-dependent), but it is unclear from the histograms shown in Fig 2b whether the CD5 histograms can be compared between the TCR transgenics. In other words, the fold reduction in CD5 between the TCR transgenics could be informative (e.g. the fold reduction in CD5 expression for AND looks less than for OTII). Showing all the CD5 levels relative to the polyclonal CD4 T cell distribution, and summarizing the mean fold reduction in CD5 in absence of Ctsl might reveal if there is a bias among the repertoire with regard to which T cell clones are more impacted than others. That the loss of Ctsl does not lead to lower self-pMHC interaction strength across all TCRs is suggested as a possibility by the TCRseq analyses, since there are Ctsl-independent sequences identified, but since CD5 expression is not shown for the polyclonal population analysed in Figure 3 from the fixed beta-chain mice, it is not clear whether selection of Ctsl-independent sequences is therefore normal, or whether those are also obtaining reduced TCR signals during thymic selection.

In summary, to distinguish between population-level effects and individual clonotype effects, it would be important to analyse the data to address whether (a) among the Ctsl-dependent TCR sequences (of which the TCR transgenics were presumably a part, but this is not explicitly commented on) all TCRs are equally impacted, regardless of pMHC affinity, and (b) among Ctsl-independent TCR sequences positive selection is indeed normal (CD5 distribution is comparable to polyclonal population that is selected in the presence of Ctsl). Some of this is addressed for the specific TCRs analysed in the later part of the paper but could already be introduced for the polyclonal repertoire here.

(3) Given that the authors show that there is an enrichment in CLIP-specific T cells in the ctsl- TEC CD4 T cell repertoire, are the 'newcomer' TCRs enriched for CLIP-specificity/cross-reactivity? In addition, are there features that the 'newcomer' TCRs share with regard to their amino acid usage that would suggest that they differ in pMHC binding properties (eg. Stadinski BD et al, Nat Immunol, 2016; Textor J et al, Cell Systems, 2023; Lagattuta KA et al, Nat Immunol, 2022)? A skew in regulatory T cell development among Ctsl-independent TCRs, or a greater presence of CD44hi cells in the periphery (due to lymphopenia-induced proliferation) might also suggest affinity differences.

(4) It is surprising that the frequency of LLO-specific tetramer+ CD4 T cells is not impacted by Ctsl expression and this raises the question of whether this would hold across other specificities as well, or whether this happened to be the case for LLO? Additional analyses of precursor frequencies for other tetramers would address this as well as the 'repertoire hole' question that is posed but not currently answered.

(5) With regard to the LLO clonotypes identified in the presence/absence of Ctsl it would be interesting to repeat the analyses from the b-chain fixed sequencing experiment to ask whether there are patterns among Ctsl-dependent vs independent TCRs with a given antigen specificity (see also point 3). This would go towards the question of whether there is something 'unique' about the LLO-specific clonotypes or whether they are indeed a good example to dig further into what is happening in the polyclonal pool in absence of Ctsl expression.

(6) Could the apparent correlation between 'natural' CD5 levels and selection efficiency in Ctsl KO be due to cross-reactivity with CLIP-MHC? I.e. the prediction would be that Lm54 is better selected in Ctsl-KO because it can obtain positive selection signals from CLIP-MHC, which is why in the natural state it is lower in CD5 (no CLIP-MHC signals)?

(7) Experimental data beyond FSC (which is not a very good proxy for cell size) and CD44 expression to support the idea that thymocytes from LM54 Ctsl- Tec are "less poised for protein biosynthesis" would be needed to support this conclusion (perhaps specific targets from Wolf et al, 2020, Nat Immunol could be investigated in that context).

(8) Could the observation of the hyper-responsiveness, based on CD69 upregulation (Fig 6e) be a function of reduced expression of negative regulators of TCR signaling (including CD5, but perhaps others are worth looking at here too such as PD1, CTLA4, tox, btlA, CD73, etc) that were necessary to allow these cells to pass the positive selection checkpoint at all? This seems a possible explanation. Beyond CD69 upregulation, is cell division/cell cycle progression impacted as well or do cells crash out earlier after stimulation because of metabolic defects? In the context of an in vivo antigen challenge where differences in cell numbers are accounted for, is there an impact on contribution to the antigen response between the Ctsl dependent and independent TCRs?

(9) How do the authors reconcile the severe homeostatic impairment of Ctsl-independent cells with a lack of difference in precursor frequencies in LLO-specific T cells? To what extent are the Ctsl-dependent and independent cells seeing the same or different self-peptidome in the periphery and does this play a role (beyond IL7 signaling) in the competitive advantage of the Ctsl-dependent LM54 cells when they are cotransferred with Ctsl-independent LM54 cells? While the MHCII KO transfer experiment might partially get at this it could still be that the Ctsl-independent LM54 cells are more reliant on tonic MHC than the Ctsl-dependent cells, hence their more rapid loss in the MHCII KO context. Is the apoptosis of T cells in dissociated culture (Fig 6g) abrogated if IL7 is added? Is there any indication in the in vitro culture experiment that the cells are more rapidly becoming less 'poised' to respond to TCR stimulation?

Minor comments:

- Please include in the legend (fig 1) information on how the cTEC vs mTEC histograms shown were gated.
- Were analyses of the TCRs (diversity/sequence/overlap) performed at the nucleotide or amino acid level? Presumably this is amino acid level but the legend (Fig. 2) should specify.
- Could the authors include example CD5 histograms in Fig 3c,d as for Fig 2b so the shape of the distribution can also be examined, since this might be informative.

- There is no explanation in the results/legend for Fig 6a what is meant by 'base mean' or what an M/A plot is.
- It was not very clear what the significance of *Tmie*/*Slc16a5* upregulation is in the LM54 cells generated in the *Ctsl*-KO system. It seemed an odd choice of genes to highlight out of the 391 that were upregulated.
- Please ensure that a supplementary table is included with all the genes, cpm, FDR, and fold changes as relates to Figure 6a (in addition to putting the raw data on GEO).
- Please also ensure that a supplementary table is included with the TCR α sequencing results from Fig5a and the TCR β sequencing results from Fig 3. GEO number for the TCR α sequencing results was not provided.

Reviewer #3:

Remarks to the Author:

This manuscript describes that cathepsin L (CTSL)-dependent positive selection shapes TCR repertoire and function of CD4 T cells. The results show that *Foxn1*-Cre-mediated loss of CTSL in TEC causes the reduction in CD4SP thymocytes in polyclonal situation as well as in TCR-transgenic situation. TEC-specific loss of CTSL affects CD4SP development even at CD69+ DP thymocytes. CD4 T cells generated in CTSL-deficient thymus are low in CD5 and altered in TCR repertoire and antigen response. Based on these results, the manuscript concludes that CTSL-dependent peptidome in cTEC promotes the positive selection at the upper end of the permissible affinity spectrum to optimize CD4 T cell repertoire formation. Most data are clearly presented and the manuscript is elegantly written. However, the manuscript has many issues that require careful attention.

1. The major conclusion of this manuscript is the importance of cathepsin L in the positive selection and repertoire formation of CD4 T cells. This key point is essentially repetitive to the pioneering article reported by Dr. Rudensky's group, in which they concluded that the repertoire of CD4 T cells selected in CTSL-deficient thymus differs from that in wild-type thymus (ref 30). The ref 30 article showed the difference in TCR repertoire of CD4 T cells generated between control and CTSL-deficient thymus as revealed by different susceptibility to selection pressure against bone marrow derived MHC-II molecules. However, this manuscript states that the previous work only showed the impact of CTSL in the quantity and size, but not the repertoire or quality, of CD4 T cells, for example, in the abstract and introduction. The manuscript should avoid underestimation of the previous achievements.

2. Another important conclusion described in this manuscript is the contribution of CTSL-dependent peptidome in the positive selection of CD4 T cells. However, this manuscript does not demonstrate data of MHC-II-bound peptidome. Moreover, the interpretation that CTSL-dependent MHC-II-bound peptidome impacts the positive selection of CD4 T cells is not new in this manuscript and was again described in Dr. Rudensky's work (ref 30). As a lysosomal endopeptidase, CTSL cleaves a variety of proteins and the role of CTSL is not limited to the production of MHC-II-bound peptides. Indeed, it was shown that CTSL is important for the degradation of invariant chain of MHC-II molecules (ref 29), which is confirmed in this manuscript (Fig. 1e, f). Furthermore, it was shown that CTSL affects NKT cell development by regulating non-classical CD1d presentation by thymocytes (ref 31). Thus, it is possible that CTSL regulates CD4 T cell repertoire even in an MHC-II-peptidome-independent manner. The manuscript should carefully avoid overinterpretation regarding peptidome-dependency in CTSL regulation of CD4 T cell development and selection.

3. Equally important point highlighted in this manuscript is the conclusion that the thymic positive selection contributes to the optimization of T cell functionality (and CD5 expression level in T cells). This was also reported previously and extensively (Mandl, et al. *Immunity* 2013; Fulton et al., *Nat Immunol.* 2015; Takada, et al. *Nat Immunol* 2015). The link between thymic CTSL and T cell function appears novel, but the authors should give a careful attention to what was already reported in previous articles.

4. CTSL is expressed highly in cTEC but is also expressed in a variety of cells including mTEC and DC (as shown in many studies including ImmGen database). Experimental approach in this study exclusively relies on *Foxn1*-Cre-dependent floxed CTSL-deletion, which affects both cTEC and mTEC. Even though some data are shown to examine the involvement of MHC-hi subpopulation or Aire+ subpopulation of mTEC (Fig. 1d, f, h), mTEC are highly diverse. It is possible that CTSL expressed in mTEC, including functionally mature mTEC-lo subpopulations, may play a crucial role in the regulation of CD4 T cell selection. Indeed, it was previously shown that CD69+ DP thymocytes include CCR7+ cells, which predominantly localize in the medulla (e.g., Kimura, et al. *Nat Immunol.* 2016), suggesting that the reduction in CD69+ DP thymocytes shown in Fig. 3b may be due to the medullary negative selection. Consequently, data presented in this manuscript does not readily allow to draw the conclusion that CTSL in cTEC regulates positive selection of cortical DP thymocytes.

5. Data shown in Fig. 1g and h support that the loss of MHC-II in either bone marrow-derived cells or Aire+ mTEC fails to rescue CD4SP cells in TEC-specific CTSL-deficient mice, suggesting that CTSL regulation of CD4 T cell development is independent of the negative selection by bone marrow-derived cells or Aire+ mTEC subpopulation. However, it is appreciated that both bone marrow-derived cells and mTEC are potent in inducing the negative selection. The contribution of the negative selection should be examined by analyzing whether CD4SP cells are rescued by the lack of MHC-II in BOTH bone marrow-derived cells AND mTEC (not only in Aire+ mTEC subpopulation).

6. It is interesting and curious that the loss of CTSL in TEC equally and severely affects CD4 T cell development in all the seven kinds of TCR transgenic T cells, which show different levels of CD5 (Fig. 2). These data do not fit the conclusion that CTSL preferentially promotes the positive selection of CD5-hi high-affinity CD4 T cells. Instead, the results may support the possibility that CTSL regulates CD4 T cell development independent of MHC-II-bound peptidome or independent of TCR affinity to the positive selection ligands.

7. Thymocyte development in Fig. 1a,b, 2a, 3a, 3e, 5b, 5d, and S3 should be analyzed by evaluating absolute numbers, rather than the frequencies, of thymocyte subpopulations.

Version 1:

Decision Letter:

Our ref: NI-A38005A

17th Mar 2025

Dear Dr. Klein,

Thank you for submitting your revised manuscript "Cathepsin L-Dependent Positive Selection Shapes Clonal Composition and Functional Fitness of CD4 T Cells" (NI-A38005A). It has now been seen by the original referees and their comments are below. We are happy to inform you that if you revise your manuscript appropriately according to our editorial requirements, your manuscript should be publishable in Nature Immunology.

I will now pre-edit the current version of your paper. We will also perform detailed checks on your paper and will send you a checklist detailing our editorial and formatting requirements in about two weeks. Please do not upload the final materials and make any revisions until you receive this additional information from us.

If you had not uploaded a Word file for the current version of the manuscript, we will need one before beginning the editing process; please email that to immunology@us.nature.com at your earliest convenience.

In the meantime, please deposit all omic and code data into public repositories so that the accession codes are readily available to be added in the revised manuscript. We cannot accept the paper without the codes. In addition, the ORCID of ALL corresponding authors needs to be linked to their Nature account (this frequently causes delays at acceptance). Should you have any query or comments about ORCID, please do not hesitate to contact our editorial assistant at immunology@us.nature.com.

Thank you again for your interest in Nature Immunology. Please do not hesitate to contact me if you have any questions.

Sincerely,

Ioana Staicu, Ph.D.
Senior Editor
Nature Immunology

Tel: 212-726-9207
Fax: 212-696-9752
www.nature.com/ni

Reviewer #1 (Remarks to the Author):

The authors have conducted a comprehensive revision of the manuscript by adding new data that considerably strengthens their study on the role of Cathepsin D in TECs and CD4 T cell selection. Moreover, I trust they have addressed the vast majority of my comments and criticism with new data sets.

The only shortcoming that is not fully addressed pertains to the definition of the molecular mechanism needed to fully elucidate the role of Cathepsin L in MHC II processing versus peptide ligandome generation for the positive selection of CD4 T cells. The biochemical experiments are indeed challenging due to the rarity of the cells involved, and the authors acknowledge this limitation in their discussion that further methodological advances are required to further define the nature of pMHC ligandomes on cTECs and the impact of Cathepsin L in this process. Still, they also provide additional data analyzing alterations in peptide presentation in cTECs from B6 × Balb/c F1 mice, which support indirect evidence that Cathepsin L shapes, perhaps both the peptide ligandome and MHC II processing in cTECs.

Reviewer #2 (Remarks to the Author):

This is an important study that sheds light on the role of cathepsin L in positive selection and the TCR repertoire of the CD4 T cell compartment, providing a conceptual advance from prior work. Experiments are beautifully executed and draw on a variety of tools and approaches, which combined provide a really extensive set of data that highlights the impact of positive selection not only on the TCR repertoire but on the functional tuning of selected T cells. The manuscript is also very well written, which is commendable given the complexity of the data being described.

The authors have extensively revised and built on their work based on reviewer's comments and have done an excellent job

of addressing criticisms, which has further strengthened the manuscript – including quite some additional data. Of note, requesting peptidome work is quite a tall order given how challenging this is – this would be a whole investigation on its own and few labs are 'expert' at this, thus involving setting up an entire new collaboration to answer this question. In my view, this is not a reasonable, nor a necessary, ask to support their conclusions.

We thank the reviewers for the positive and fair assessment of our manuscript and appreciate the time and effort they have invested. In response to their constructive criticism and helpful comments, we have conducted additional experiments wherever feasible and incorporated them into the revised manuscript. We believe these new results significantly add to the manuscript and bolster our conclusions.

Some of the requested experiments were time-consuming, particularly the generation or expansion of mouse cohorts with rare genotypes for the new Figs. 1g and 4a. Inspired by the reviewers' comments, this also allowed us to pursue two additional aspects with major new TCRseq experimentation: whether a CD4 T cell's requirement for Ctsl relates to (i) its pMHC affinity in the thymus (as reported by CD5 expression) and (ii) its 'responsiveness' in the periphery to adjuvanted cognate antigen. These exciting findings are now presented in the new Figure 5. While this does not reflect the chronological order of experimentation, we have incorporated it at this position to maintain a coherent logical flow.

The new TCRseq data in Figure 5 have been annotated and analyzed using an optimized bioinformatics workflow. To enable cross-comparison with the previous dataset in Figure 3 and ensure a coherent TCR annotation across all datasets – including single-cell TCR $\alpha\beta$ data in Figure 6 (formerly 5) – we have reanalyzed the data in Figures 3 and 6 based on these new annotations. Moreover, we used a more stringent inclusion criterion (≥ 3 of 6) to achieve a higher robustness in the definition of TCR-classes (For instance, a 'lost' TCR seen in 2/3 WT samples and 0/3 Ctsl^{A^{TEC}} samples, or vice versa a 'newcomer', is no longer included). As a result, the total number of 'recurrent' TCRs in Fig. 3 is substantially lower now, and numbers and percentages in the updated Figures 3 and 6 differ from previous values. However, these updates do not affect the original conclusions in any way.

Additionally, the introduction and discussion have been extensively revised for clarity and to better integrate previous knowledge in response to reviewer requests; these changes are not specifically indicated. Changes in the results section are underlined.

A tabular list of new display items is provided on the last page of this document.

Point-by-point response (new display items are highlighted in bold)

Reviewer #1:

...

This study has a significant merit for its detailed analysis of the consequences of Ctsl deficiency in T cell development, employing several advanced and complementary methodologies to demonstrate its impact. Apart of some comments on T cell analysis, the manuscript does not fully address, as discussed by the authors, the molecular analysis and mechanisms controlled by Ctsl in cTECs. Although these experiments (e.g. identification of the peptidome controlled by Ctsl) are challenging, the study would benefit from a more in-depth examination of the molecular pathways to fully elucidate the role of Ctsl in T cell development. My specific comments are found below.

(1) Impact of Cathepsin L on MHC II Processing and Peptidome Generation: Previous studies have shown that the thymus of mice with combined deficiencies in Ctsl and Ii generated fewer CD4 T cells compared to Ctsl and Ii single-deficient mice. This suggests that Ctsl may also contribute to the generation of specific selecting peptides for CD4 T cells. However, the current results do not formally address whether Ctsl affects MHC II processing and/or peptidome generation. These are challenging points to address experimentally due to the complexity of proteomics and peptide-bound MHC II identification in rare cells such as cTECs. Nevertheless, an analysis of MHC II processing in sorted cTEC and mTEC populations may be provide in for example sorted cTEC and mTEC. Is there any possibility that the authors extended the analysis of peptidome, apart of the analysis with MHCII:non-CLIP Ab shown in Fig1F. These points and limitations should be attempted to be further enhanced and discussed.

As a significant additional characterization of the pMHC ligandome, we generated F1 B6 x BALB/c mice, demonstrating that a prototypical 'frequent' non-CLIP pMHC combination (Ea52-68:I-Ab) is retained in the pMHC ligandome of cTECs, albeit at moderately decreased levels (**Fig. 1g**).

This extends our flow-cytometric evidence, further consolidating the notion that:

- i) total MHCII levels of cTECs remain 'normal',
- ii) Ctsl-deficiency does not create a cTEC ligandome composed exclusively of CLIP:MHCII,
- iii) total non-CLIP ligands are moderately reduced on cTECs but remain far from absent,
- iv) all these effects are seen in cTECs, with no measurable impact on mTECs.

While points i-iii have been previously described, a rigorous side-by-side comparison of cTECs versus mTEC had not been conducted before.

Besides its role in antigen processing, it is well established that Ctsl also contributes to MHCII maturation along the antigen-loading pathway. We agree that we do not formally address the individual impact of the two functions. Given that both *bona fide* peptide processing and MHCII maturation are directly linked to Ctsl's proteolytic activity, their effects are likely inseparable. Nevertheless, Ctsl deficiency-associated disturbances in MHCII maturation will ultimately also contribute to an altered pMHC ligandome. Importantly, total MHCII surface levels are preserved.

The data in Fig. 1 serve as the starting point of our study and set the stage for three key questions:

- 1) What is the exact nature and extent of Ctsl's impact on the CD4 T cell repertoire?
- 2) How does this affect CD4 T cell responses?
- 3) Are retained clones functionally altered, and if so, how?

We fully agree that future work must aim to mechanistically link our key findings on these three aspects to Ctsl's effect on the pMHCII ligandome. We hope and are confident that our work will inspire further efforts in this direction. In the revised discussion, we elaborate on why this remains a formidable challenge. Major hurdles include the limited sensitivity of contemporary methods for characterizing MHC-bound peptides and the scarcity of cTECs, as comprehensively outlined in Ref 70. 'Heroic' efforts to elucidate the pMHCII ligandomes of cTECs and mTECs – leveraging mutant mice with an enlarged thymus – are currently being pursued by Dr. Yousuke Takahama's team (Ref 78). The results of these studies will be highly anticipated. Of note, corresponding studies on the pMHCII ligandome will likely pose an even greater challenge, given the variable length of MHCII-bound peptides and the 'open' peptide binding cleft, which complicates precise alignment of anchor residues and binding registers.

Against this background, we respectfully ask for consideration that these aspects far exceed the scope of the present work.

(2) The cellularity of thymic and splenic T cell subsets in control and Ctsl cKO mice is not shown in Figure 1. Including these data would provide a more comprehensive understanding of the effects of Ctsl deficiency on T cell populations.

We apologize for erroneously omitting this information. Corresponding cell numbers are now included here and elsewhere in the revised MS (**Extended Fig. 1 a,e; Extended Fig. 2 c; Extended Fig. 3 a,c; Extended Fig. 6 a,b**).

(3) The manuscript does not address the development of regulatory T cells (Tregs) and the maturation of SP4 cells in Ctsl cKO mice. Although Treg development is primarily dependent on mTECs, it would be interesting to examine how this process is influenced by purported altered positively selected CD4 T cells. Additionally, the analysis of peripheral T cell composition (naïve, effector, and memory subsets) in Ctsl cKO mice is limited and can be expanded. Moreover, do the mice develop any signs of disturbed immunotolerance with age? or the hyporesponsive state and imprinted abnormal selection of T cells in Ctsl cKO impairs the development of autoimmune response.

We have not observed signs of disturbed immune tolerance with age, but have not systematically kept mice beyond max. 3 months of age.

It is certainly an intriguing scenario that the impairments in the CD4 T_{conv} compartment not only impact the capacity to mount an anti-foreign response, but may also protect from autoimmunity. Indeed, Ctsl^{-/-} mice on NOD background were reported to be protected from spontaneous diabetes (Ref. 48). However, based upon (partial) 're-emergence' of diabetes upon depletion of T_{reg} cells, this was attributed to the enlarged peripheral T_{reg} compartment in Ctsl^{-/-} mice.

We have included additional new information on the thymic and peripheral Treg cell compartment (**Extended Fig. 1 c,d,g**). Indeed, there is an enlarged proportion of T_{regs} among peripheral CD4 T cells,

likely to be a consequence of lymphopenia driven expansion in the periphery rather than 'disproportional thymic selection'. In favor of this interpretation, the percentage of T_{regs} in the contracted nascent CD4SP is similar to that in WT mice. This suggests that T_{reg} differentiation is 'as affected' as T_{conv} differentiation, but likely not at the level of inductive signals (presumed to primarily arise from medullary APCs), but at the level of a smaller precursor population that – downstream of a bottleneck in positive selection – auditions for T_{reg} differentiation.

These are indeed very interesting aspects of *Ctstl*-deficiency; however, an in-depth investigation of the ' T_{reg} issue' would require extensive experimentation that, while conceptually related to the focus of the present MS, would not alter the central conclusions regarding the ' T_{conv} intrinsic' effects of selection in the absence of *Ctstl*.

To maintain the focus of the study, which we consider already quite data-heavy, we have not emphasized the T_{reg} aspect in the revised MS. The 'protection' of NOD mice is referenced in the revised discussion, and we believe that the effects on T_{conv} that we describe are not merely 'contributory', but likely central to this phenomenon. We have phrased this more cautiously, and would prefer to leave it as is.

We have included novel data on CD4SP maturation stages in the polyclonal setting and a pronounced skewing to a memory-like phenotype in the lymphopenic periphery (**Extended Fig. 1 b,e,f**).

*(4) The conclusion on page 6 states, "These findings rendered a reduced complexity of the cTEC ligandome per se or 'excessive' clonal deletion unlikely explanations for the diminished CD4SP cell compartment in *Ctstl* Δ TEC mice, corroborating the idea that 'normal' positive selection of CD4 T cells requires 'private' peptides generated through processing by *Ctstl*." This statement needs clarification. The results might favor the hypothesis of a less complex and/or available cTEC ligandome. The authors should elucidate how their findings support or contradict this hypothesis.*

We fully agree and apologize for the misleading wording, which has been revised accordingly. Rather than implying a 'reduced complexity of the cTEC ligandome', our experiments in this paragraph specifically provide evidence for an 'altered pMHCII ligandome' and examines whether 'excessive clonal deletion' accounts for the small size of the CD4SP compartment in *Ctstl*-deficient mice. We found the size of the CD4SP compartment is not more limited by negative selection in *Ctstl*-deficient mice than in *Ctstl*-sufficient mice.

(5) The results presented in Figure 2 involving several MHC II-restricted TCR transgenics are thorough. However, similar to Figure 1, the cellularity of distinct thymic populations should be included. Additionally, it would be valuable to know if the authors performed similar control experiments with some MHC I-restricted TCR transgenics.

We have included novel data demonstrating the normal selection of an MHCI-restricted transgenic TCR (**Fig. 2c,d**). Given the accumulated evidence that *Ctstl*-deficiency specifically affects CD4 T cell selection, it would be difficult to ethically justify further animal experimentation beyond this single example. Cellularity data for the TCR transgenics is now included in **Extended Fig. 2c**.

*(6) The hyporesponsive state of LLO-specific T cells and *Cts*-independent TCRs, as suggested later in the manuscript, should perhaps be expected to exist in the context of polyclonal CD4 T cells or in a TCR transgenic *Ctstl*-deficient setting (Fig2). Did the authors test the peripheral antigen-specific responses in TCR transgenics (Fig2)? Moreover, the homeostatic capacity of polyclonal peripheral CD4 T cells derived from *Ctstl* cKO and control mice can be examined in competitive homeostatic proliferation experiments, addressing these aspects in a more polyclonal setting.*

It is important to note that the Lm54 cells selected in the absence of *Ctstl* are in fact hyperresponsive with respect to proximal TCR signaling. In the revised discussion, we propose that this reflects a persistent 'tuning' effect due to diminished expression of various attenuators of TCR signaling – such as CD5, CD6, PD-1 and BTLA (**Fig. 7e**) – possibly alongside other mechanisms, to compensate diminished TCR signaling input during positive selection in the absence of *Ctstl*. We have included novel data confirming this 'early' hyperresponsive state for polyclonal M2 thymocytes (**Extended Fig. 7d**). None of the other transgenics in Fig. 2 generate sufficient CD4SP cells to allow for a meaningful assessment of their responsiveness.

We have also included additional data showing that in a co-transfer setting, polyclonal M2 thymocytes from *Ctsl*^{ΔTEC} mice exhibit defects in homeostatic survival and proliferation closely resembling those observed in Lm54 cells when selected in the absence of Ctsl (**Extended Fig. 7e**).

(7) Can the authors comment on the high level of MHC II:clip in mTECs relatively to cTEC?

This is an excellent and informative point that we missed to explicitly address. While the observation is well-documented, its mechanistic underpinnings remain largely enigmatic (Refs. 30 and 44). Nevertheless, the abundant presentation of CLIP:MHCII on mTECs has an important implication: it is expected to impose a strong 'negatively selecting' pressure on CLIP-specific cells. In the revised discussion, we elaborate on this in the context of arguments supporting the possibility that 'special' Ctsl-dependent peptides – rather than a mere Ctsl-dependent diversification of the pMHC ligandome, or prevention of a CLIP dominated pMHCII ligandome – define the crucial role of Ctsl in 'normal' selection.

Reviewer #2:

...

This is a very well-written and interesting article that provides a novel perspective on the role of the self-immunopeptidome on generating an effective T cell repertoire, a topic of key importance to better understanding how T cells are trained to be poised to respond effectively to non-self. The discussion relates the findings well to prior work, data figures are well-put together and analyses performed for the most part clear. While the data is already quite strong, some additional experiments and/or analyses of the data already in hand would strengthen the conclusions and perhaps better relate what was done at the polyclonal versus individual clonotype level with regard to the impact of loss of Ctsl expression in cTECs.

(1) The introduction is very well written and clear. However, it would be helpful to add a bit more detail on the role of cathepsin L in antigen presentation. ...

This point is very well taken, and we have considerably revised the introduction to accommodate this suggestion.

(2) From the analyses done as shown it is not addressed whether there is an affinity bias with regard to which T cells are positively selected or whether all T cells are equally impaired in absence of cTEC Ctsl expression....

To address this intriguing question, we have conducted extensive additional experimentation. We would therefore like to refer to the corresponding section in the revised manuscript ('Non-selection in....'; p. 9-10) for an extensive explanation and discussion. These new TCR sequencing data show at the whole-repertoire level that TCRs at both ends of the 'natural' CD5 spectrum are lost, strongly bolstering the conclusion that selecting interactions with self-pMHC are 'weaker across the board' (**NEW Fig 5a-c**). For clarity, we will here briefly break down the questions in comment (2) one-by-one:

(2.1) The authors interpret the CD5 decreases in the TCR transgenic CD4 T cells as interactions with self-pMHC being weaker across the board (...), but it is unclear from the histograms shown in Fig 2b whether the CD5 histograms can be compared between the TCR transgenics. In other words, the fold reduction in CD5 between the TCR transgenics could be informative (e.g. the fold reduction in CD5 expression for AND looks less than for OTII). Showing all the CD5 levels relative to the polyclonal CD4 T cell distribution, and summarizing the mean fold reduction in CD5 in absence of Ctsl might reveal if there is a bias among the repertoire with regard to which T cell clones are more impacted than others.

The histograms in Fig 2b were not acquired side-by-side (also note the use of different fluorochromes), hence should not be directly compared. We have now added a systematic side-by-side comparison of CD5 levels on mature CD4SP cells with these TCRs when normally selected (note that mature CD4SP with these TCRs are all virtually absent in *Ctsl*^{ΔTEC} mice and cannot be compared) (**Fig 2i**).

In the case of the transgenic TCRs tested in Fig. 2 (all 'not selected'), the CD5 levels on DP cells in *Ctsl*^{ΔTEC} mice likely reflect the absence of any signal (as opposed to a quantitative reduction). In support

of this, the CD5 expression on these 'stuck' DP cells is identical to that of polyclonal non-signaled CD69-TCRb- cells. Hence, we think these TCRs are representative of many 'essentially Ctsl-dependent' TCRs that are 'lost' because their selecting ligands are entirely absent.

(2.2) That the loss of Ctsl does not lead to lower self-pMHC interaction strength across all TCRs is suggested as a possibility by the TCRseq analyses, since there are Ctsl-independent sequences identified, but since CD5 expression is not shown for the polyclonal population analysed in Figure 3 from the fixed beta-chain mice, it is not clear whether selection of Ctsl-independent sequences is therefore normal, or whether those are also obtaining reduced TCR signals during thymic selection. In summary, to distinguish between population-level effects and individual clonotype effects, it would be important to analyse the data to address whether (a) among the Ctsl-dependent TCR sequences (of which the TCR transgenics were presumably a part, but this is not explicitly commented on) all TCRs are equally impacted, regardless of pMHC affinity, and (b) among Ctsl-independent TCR sequences positive selection is indeed normal (CD5 distribution is comparable to polyclonal population that is selected in the presence of Ctsl). Some of this is addressed for the specific TCRs analysed in the later part of the paper but could already be introduced for the polyclonal repertoire here.

Point well taken. We have now added additional data showing that the CD5 levels of bulk CD4SP in Fixed- β *Ctsl*^{ATEC} are indeed reduced (**Fig 3b**), suggesting that many of the retained and 'apparently Ctsl-independent' clones do receive weaker signals. Moreover, the extensive characterization of the clones Lm54 and Lm6, which we believe are representative of a substantial fraction of retained clones (see revised discussion), strongly suggest that 'many' retained clones receive weaker signals and, in turn, show functional deficits. These clones are 'functionally Ctsl-dependent'.

These intriguing comments inspired us to perform the new experimentation shown in (**Fig 5a-c**), from which we conclude that CD4 T cell selection was indeed 'weaker across the board' (see above).

(3) Given that the authors show that there is an enrichment in CLIP-specific T cells in the ctsl-DTEC CD4 T cell repertoire, are the 'newcomer' TCRs enriched for CLIP-specificity/cross-reactivity? In addition, are there features that the 'newcomer' TCRs share with regard to their amino acid usage that would suggest that they differ in pMHC binding properties (eg. Stadinski BD et al, Nat Immunol, 2016; Textor J et al, Cell Systems, 2023; Lagattuta KA et al, Nat Immunol, 2022)? A skew in regulatory T cell development among Ctsl-independent TCRs, or a greater presence of CD44hi cells in the periphery (due to lymphopenia-induced proliferation) might also suggest affinity differences.

We respectfully disagree with the interpretation that our findings demonstrate an enrichment of CLIP-specific (CLIP-selected?) T cells and hope we did not convey that conclusion; this was certainly not our intention. In the revised discussion, we further elaborate on why we believe newcomer TCRs are unlikely to be selected on MHC:CLIP.

One key argument is that, although this ligand is increased on *Ctsl*^{ATEC} cTECs, it is also present on WT cTECs (**Fig. 1e**; we have updated these panels to enhance clarity). Nevertheless, 'newcomer' TCRs are found exclusively in *Ctsl*^{ATEC} mice, suggesting an alternative mechanism.

Equally important, the abundant presentation of CLIP:MHCII on mTECs (**Fig. 1e**; see also Reviewer #1, point 7) is expected to exert a strong 'negatively selecting' pressure on CLIP-specific cells and/or cells positively selected by CLIP:MHCII. This strongly argues against 'newcomers' (but also retained clones including 'functionally Ctsl-dependent TCRs') being positively selected on or cross-reactive to CLIP:MHCII. Similarly, it suggests that clones like Lm54 or Lm6 – or any other 'functionally Ctsl-dependent TCR' – are unlikely to be cross-reactive to CLIP:MHCII (see also response to Point 6). Unless, of course, a cell positively selected on a particular pMHCII combination could somehow evade negative selection by the same pMHCII ligand presented on mTECs (or DCs). However, we are not aware of any evidence to support this possibility.

We fully agree that the properties and functional capacity of 'newcomer' TCRs are intriguing questions. However, addressing them would require in vivo re-expression and extensive experimentation – similar to our studies with Lm54 and Lm6, but notably without prior knowledge of any cognate antigen specificity. We respectfully ask for consideration that this exceeds the scope and focus of our current manuscript.

Along these lines, beyond the evidence that ‘newcomers’ as a whole have unusual V- and J-element as well as N-nucleotide features (**Extended Fig. 3**), suggesting ‘unusual’ selection, we’d rather refrain from including too much ‘deviating’ data and speculation on the nature of these TCRs into the MS. The suggested analyses of TCR features associated with autoreactivity, as far as we see it, all relate to features of the TCR β CDR3. Our TCRseq was performed on a repertoire with a Fixed- β chain, precluding a cross-comparison to these studies. We have, however, run an analysis of amino acid usage in the CDR3 α using the methodology of Textor et al. (2023). These data are shown below. In our opinion, they fail to reveal canonical AA motifs in the CDR3 α of ‘newcomers’ vs. ‘lost’ TCRs (left) or robustly differential amino acid usage when the repertoire is stratified by CDR3 α length (right), and we’d hence prefer to not include such ‘negative’ data.

(4) It is surprising that the frequency of LLO-specific tetramer+ CD4 T cells is not impacted by Ctsl expression and this raises the question of whether this would hold across other specificities as well, or whether this happened to be the case for LLO? Additional analyses of precursor frequencies for other tetramers would address this as well as the ‘repertoire hole’ question that is posed but not currently answered.

We were very surprised by this observation as well. As suggested, we have now performed additional experimentation on ‘antigenic holes’ in the repertoire, using three other I-Ab tetramers (**Fig. 4a**). Clearly, a ‘numerically maintained’ Tet⁺ population as seen for LLO is the exception, likely serendipitous. We feel that this piece of data significantly adds to the manuscript and thank the reviewer for the inspiration.

(5) With regard to the LLO clonotypes identified in the presence/absence of Ctsl it would be interesting to repeat the analyses from the b-chain fixed sequencing experiment to ask whether there are patterns among Ctsl-dependent vs independent TCRs with a given antigen specificity (see also point 3). This would go towards the question of whether there is something ‘unique’ about the LLO-specific clonotypes or whether they are indeed a good example to dig further into what is happening in the polyclonal pool in absence of Ctsl expression.

We conducted additional experimentation in Fixed- β mice to characterize the expanded Tet⁺ cells in immunized Ctsl^{+/+} mice (**Fig 5d**). The expanded population of Tet⁺ cells is both highly stereotypic between mice and remarkably diverse (**Table S3** shows a database of 127 clonotypes present in all 4 replicates). A cross-comparison to the Fixed- β M2 CD4SP thymocyte dataset (Fig. 3) shows that the Top Ten responder clones (as well as the entire Tet⁺ set) consist of nearly equal proportions of ‘essentially Ctsl-dependent’ and (seemingly) Ctsl-independent TCRs. Thus, this approach does not seem suited to ‘dig further into what is happening in the polyclonal pool in absence of Ctsl expression’. However, the findings bolster the notion that many, if not all, (seemingly) Ctsl-independent TCRs are in fact ‘functionally Ctsl-dependent’ TCRs, like Lm54 and Lm6. These TCRs are good responders when they develop in a Ctsl-sufficient thymus, but not when they develop in a Ctsl-deficient thymus.

Regarding the hypothesis that certain characteristics in the TCR α primary sequence might differentiate Ctsl-dependent from Ctsl-independent clones, we did not observe any discernable pattern (within the scope of our possibilities; see point 3). Based on the sum of the data presented in the revised MS, we presently see little reason to believe that there is ‘something inherently unique’ about LLO-specific clones (other than their, likely serendipitous, unchanged number). However, we acknowledge that hypotheses like this are difficult to formally exclude.

(6) Could the apparent correlation between 'natural' CD5 levels and selection efficiency in Ctstl KO be due to cross-reactivity with CLIP-MHC? I.e, the prediction would be that Lm54 is better selected in Ctstl-KO because it can obtain positive selection signals from CLIP-MHC, which is why in the natural state it is lower in CD5 (no CLIP-MHC signals)?

New experimentation shown in **NEW Fig 5a-c** (see above) shows at the global repertoire level that 'natural' CD5 characteristics are no predictor of loss in the Ctstl-deficient thymus. We conclude there is no correlation between 'natural' CD5 levels and Ctstl-dependency.

We cannot formally exclude the hypothetical scenario that CLIP:MHCII is involved in the selection of the TCR Lm54 (or any other Ctstl-independent TCR). However, we deem this extremely unlikely. We have performed various pilot experiments failing to reveal any evidence that the TCR Lm54 might be responsive to CLIP peptide (not an agonist). Likewise, mixing in CLIP into LLO stimulations is a 'null event' (neither co-agonist nor antagonist). These are 'negative' results, which we prefer not to include in the MS. See also the response to **(3)** concerning the significance of abundant presentation of CLIP:MHCII on tolerogenic mTECs in this context. In the revised discussion, we provide additional arguments in favor of 'rare' Ctstl-dependent peptides, rather than CLIP:MHCII, being critical.

(7) Experimental data beyond FSC (which is not a very good proxy for cell size) and CD44 expression to support the idea that thymocytes from LM54 Ctstl-DTEC are "less poised for protein biosynthesis" would be needed to support this conclusion (perhaps specific targets from Wolf et al, 2020, Nat Immunol could be investigated in that context).

Newly included measurements of 'true' cell size confirm the difference (**Extended Fig. 7a**). Most significantly, additional new data show a direct assessment of basal translation and directly confirm the presumed differences (**Fig. 7c**).

(8) Could the observation of the hyper-responsiveness, based on CD69 upregulation (Fig 6e) be a function of reduced expression of negative regulators of TCR signaling (including CD5, but perhaps others are worth looking at here too such as PD1, CTLA4, tox, btlA, CD73, etc) that were necessary to allow these cells to pass the positive selection checkpoint at all? This seems a possible explanation. Beyond CD69 upregulation, is cell division/cell cycle progression impacted as well or do cells crash out earlier after stimulation because of metabolic defects? In the context of an in vivo antigen challenge where differences in cell numbers are accounted for, is there an impact on contribution to the antigen response between the Ctstl dependent and independent TCRs?

Excellent suggestion. We deem this indeed a very plausible scenario that we have incorporated into the revised discussion. We have now included additional analyses of other negative regulators of TCR signaling. Indeed, PD-1 and BTLA show a strikingly similar reduced amplitude as CD5 downstream of positive selection (**Fig 7e**). Moreover, CD6 shows the very same pattern (**Extended Fig. 6c**). With regards to whether Ctstl-dependent or -independent clones in the 'normal' repertoire differentially contribute to an antigen response, we have added major new TCRseq data on the contribution of both classes of TCRs to the LLO response (**Fig 5d**).

(9) How do the authors reconcile the severe homeostatic impairment of Ctstl-independent cells with a lack of difference in precursor frequencies in LLO-specific T cells? To what extent are the Ctstl-dependent and independent cells seeing the same or different self-peptidome in the periphery and does this play a role (beyond IL7 signaling) in the competitive advantage of the Ctstl-dependent LM54 cells when they are cotransferred with Ctstl-independent LM54 cells? While the MHCII KO transfer experiment might partially get at this it could still be that the Ctstl-independent LM54 cells are more reliant on tonic MHC than the Ctstl-dependent cells, hence their more rapid loss in the MHCII KO context. Is the apoptosis of T cells in dissociated culture (Fig 6g) abrogated if IL7 is added? Is there any indication in the in vitro culture experiment that the cells are more rapidly becoming less 'poised' to respond to TCR stimulation?

This is a point that has intrigued us as well. The size of the peripheral precursor pool as assessed by tetramer-staining is a snapshot in time; the size of this population in steady state is influenced by numerous parameters such as thymic output (note that LLO Tet+ are present in the Ctstl^{ΔTEC} thymus in similar absolute numbers, corresponding to a substantially higher relative abundance in the nascent

repertoire; **Extended Fig 4**), homeostatic survival, and last not least, reflects the sum of these effects on an in-all-likelihood heterogeneous population of Tet⁺ cells that is composed of cells that are affected in their homeostatic behavior upon selection in Ctsl-deficient thymus (as we conclusively show), and others that may not be affected (speculation). The latter may compensate. We elaborate on the 'mixed' nature of the LLO-Tet⁺ population in the revised discussion.

The question of whether Lm54 cells selected in the presence or absence of Ctsl see the same or different pMHC ligandome in the periphery is intriguing, but we unfortunately don't see how this can conclusively be answered. They have the very same TCR, and compete in the very same host, so they at least have the chance to encounter the same ligands (we do not find evidence for altered homing upon transfer, but this is certainly a formal possibility). We have added new experimentation regarding the in vitro rescue by IL-7, which indeed occurs at very high concentrations (likely supraphysiological) (**Fig. 7i**). It certainly remains possible that the diminished survival of cells selected in Ctsl^{ΔTEC} donors in MHCII^{-/-} recipients reflects an increased dependency on tonic signals rather than a decreased responsiveness to non-MHC input. We deem it in fact very likely that these effects act in concert. We elaborate on putative intersections of CD5, TCR signal and responsiveness to IL-7 in the revised discussion.

Minor comments:

- Please include in the legend (fig 1) information on how the cTEC vs mTEC histograms shown were gated.

A new supplementary figure (**Fig S2**) has been included depicting the gating strategies used in Fig. 1 (TECs) and in Fig 4 (Tet⁺ cells).

- Were analyses of the TCRs (diversity/sequence/overlap) performed at the nucleotide or amino acid level? Presumably this is amino acid level but the legend (Fig. 2) should specify.

Throughout the study, a 'clonotype' was defined at the level of amino-acid sequence (V-region, CDR3, J-region). This is now specified in the legends to Fig. 3 and NEW Fig. 5.

- Could the authors include example CD5 histograms in Fig 3c,d as for Fig 2b so the shape of the distribution can also be examined, since this might be informative.

Representative CD5 stainings have now been included, now **Fig. 2g,h** (formerly Fig. 3c,d). As a note of caution, and although it may be tempting, we would be very hesitant to read too much into the shapes of these distributions with regards to which clones might be affected versus how signaling might be affected (even more so across experiments). This is one of the reasons why we performed the extensive new analyses of the loss of clones in the 'natural' CD5^{lo} vs 'natural' CD5^{hi} sub-repertoires (**Fig. 5a-c**).

- There is no explanation in ... Fig 6a what is meant by 'base mean' or what an M/A plot is.

We apologize for this; the explanation fell victim to our attempts to limit the word count. A clarification is now included as part of the legend to Fig. 7a (formerly Fig. 6a).

- It was not very clear what the significance of Tmie/Slc16a5 upregulation is in the LM54 cells generated in the Ctsl-KO system. It seemed an odd choice of genes to highlight out of the 391 that were upregulated.

We fully agree that highlighting certain genes in such analyses is highly subjective, which is why we performed GSE analyses. Nonetheless, visual inspection of these datasets revealed a number of parallels to observations in CD8 T cell selected on 'weak' interactions (Lutes et al, 2021; Ref 55). We find this intriguing. We have rephrased the respective section, hoping to improve clarity.

- Please ensure that a supplementary table is included with all the genes, cpm, FDR, and fold changes as relates to Figure 6a (in addition to putting the raw data on GEO).

A corresponding new supplementary table (**Table S2**) has been added (Fig6a is now Fig 7a).

- Please also ensure that a supplementary table is included with the TCRab sequencing results from Fig5a and the TCRa sequencing results from Fig 3. GEO number for the TCRab sequencing results was not provided.

A new supplementary table (**Table S1**) has been added to provide the TCR $\alpha\beta$ sequencing results. Data in Table S1 and the underlying nucleotide sequences will be made publicly available via ImmuneACCESS or a similar Immune receptor-specific repository with acceptance of the manuscript for publication. GEO is unfortunately less suited for this purpose. The full TCR α sequence dataset from Fixed- β samples, and accompanying computer code for all analyses, has been uploaded to a github repository that will be made public upon acceptance of the manuscript.

Reviewer #3:

...

Most data are clearly presented and the manuscript is elegantly written. However, the manuscript has many issues that require careful attention.

1. The major conclusion of this manuscript is the importance of cathepsin L in the positive selection and repertoire formation of CD4 T cells. This key point is essentially repetitive to the pioneering article reported by Dr. Rudensky's group, in which they concluded that the repertoire of CD4 T cells selected in CTSL-deficient thymus differs from that in wild-type thymus (ref 30). The ref 30 article showed the difference in TCR repertoire of CD4 T cells generated between control and CTSL-deficient thymus as revealed by different susceptibility to selection pressure against bone marrow derived MHC-II molecules. However, this manuscript states that the previous work only showed the impact of CTSL in the quantity and size, but not the repertoire or quality, of CD4 T cells, for example, in the abstract and introduction. The manuscript should avoid underestimation of the previous achievements.

We couldn't agree more that the studies (now Refs. 31 and 34) from Sasha Rudensky's team represent landmark papers; over many years, we have consistently recognized these breakthroughs by citing them in original work as well as in reviews on thymic selection. We have now reintroduced an introductory paragraph on cathepsins in antigen presentation that we omitted for length limitations, and we are confident to now provide a better reflection of the state-of-the-art at the beginning of our study.

Against this background, we find the assessment that our work is 'repetitive' of existing knowledge – implying a lack of conceptual novelty – somewhat perplexing!

We therefore reiterate the state-of-knowledge on Ctsl's role in CD4 T cell selection as of 2002 and stand by our statement that there has been little to no progress since – without diminishing the significance of prior achievements:

- i) Ctsl-deficiency results in a dramatic contraction of the thymic CD4 T cell compartment (Ref. 31)
- ii) This is independent of Ctsl's role in li-degradation (Ref. 34)
- iii) One transgenic CD4 TCR is not selected in Ctsl-deficient mice (later shown to apply to a second one; Ref 48), while a CD8 TCR is (Ref. 34)
- iv) There is a disproportionally high 'rebound' of the CD4SP compartment when negative selection by hematopoietic APCs is eliminated (Ref. 34)
- v) The most likely explanation for this is an altered pMHCII ligandome on cTECs (Refs. 31 and 34)

These exciting observations, which we by no means underestimate, motivated us to embark on this study.

We fully agree that Ref. 34 established that the CD4 repertoire selected in the absence of Ctsl must somehow be 'altered'. However, since then, little – if any – progress has been made in defining the nature of these 'alterations': Is the CD4 repertoire entirely different? If not, which fraction of clones is lost? Is there a discernable pattern in the clones that are lost? Do new TCRs emerge concomitant with the loss of 'normal' TCRs (we were stunned to observe this...)? If some clones persist, are they functionally still 'the same'? If not, how are they functionally different? How do these changes shape immune responsiveness at the repertoire level? And how do they impact CD4 T cell responsiveness and homeostasis at the clonal level?

Fig. 1a-f is, if one wishes to call it that way, 'repetitive'; but it was essential to validate key observations from previous studies with the conditional model. While we will not dwell too deeply on a 'repetitive' experiment that yielded a diametrically opposite outcome (and also do not do so in the manuscript), we note that neither in our conditional model, nor in the exact genetic constellation previously used in Ref.

34, do we observe any disproportionate rebound of the CD4 compartment upon ablation of MHCII in hematopoietic APCs (Fig. 1h and Ext. Fig. 1j; compare Ref. 34, Fig. 2). It is not our role to reconcile this discrepancy. However, this has major implications in terms of Ctsl-deficiency causing ‘excessive’ susceptibility to negative selection (reminiscent of the peptide-switch scenario, but not synonymous...) versus a genuine quantitative bottleneck in positive selection.

From Fig. 2 on, our work addresses the open questions listed above. We would respectfully ask for consideration that elucidating these open issues represents a significant conceptual advancement. It also opens exciting new perspectives on the ‘parallel universe’ of $\beta 5t$'s role in CD8 T cell selection – a field that has long been, and remains, years ahead of our understanding of Ctsl in CD4 T cell selection. In the revised discussion, we extensively explore these parallels, but also (apparent?) distinctions that we hope will inspire further investigation.

2. Another important conclusion described in this manuscript is the contribution of CTSL-dependent peptidome in the positive selection of CD4 T cells. However, this manuscript does not demonstrate data of MHC-II-bound peptidome. Moreover, the interpretation that CTSL-dependent MHC-II-bound peptidome impacts the positive selection of CD4 T cells is not new in this manuscript and was again described in Dr. Rudensky's work (ref 30). As a lysosomal endopeptidase, CTSL cleaves a variety of proteins and the role of CTSL is not limited to the production of MHC-II-bound peptides. Indeed, it was shown that CTSL is important for the degradation of invariant chain of MHC-II molecules (ref 29), which is confirmed in this manuscript (Fig. 1e, f). Furthermore, it was shown that CTSL affects NKT cell development by regulating non-classical CD1d presentation by thymocytes (ref 31). Thus, it is possible that CTSL regulates CD4 T cell repertoire even in an MHC-II-peptidome-independent manner. The manuscript should carefully avoid overinterpretation regarding peptidome-dependency in CTSL regulation of CD4 T cell development and selection.

Point well taken; we hope that in the revised manuscript, we have now done due justice to the suggestion of avoiding overinterpretations. We never intended to claim being the first to conclude that ‘*CTSL-dependent MHC-II-bound peptidome impacts the positive selection of CD4 T cells*’ (see point 1).

The pMHC ligandome of TECs is far beyond the scope of our work. In the revised introduction and discussion, we elaborate on why elucidating the pMHC ligandome of TECs remains a formidable challenge in itself, let alone linking such insights to T cell fate, as comprehensively outlined in Ref 70. ‘Heroic’ efforts towards elucidating the pMHCI ligandomes of cTECs and mTECs are currently being pursued by Dr. Yousuke Takahama's team (Ref 78). The results of these studies will be highly anticipated in the field. Notably, corresponding studies on the pMHCII ligandome will likely pose an even greater challenge, given the variable length of MHCII-bound peptides and uncertain MHCII-binding registers.

Against this background, we cannot reiterate here all the accumulated ‘circumstantial’ evidence put forward in the landmark study by Dr. Rudensky (Ref. 34) to support that defective CD4 T cell selection in Ctsl-knockout mice is indeed driven by alterations in the pMHCII ligandome of positively selecting cells. However, we would like to highlight one key additional argument that emerges from our investigation:

Our global repertoire analysis reveals a highly selective loss of clonotypes, while others are retained. At the same time, there is no discernable overarching pattern distinguishing lost from retained TCRs – for example, in terms of V- or J-region usage, as might be expected in a superantigen-like scenario. Perhaps even more significant, a vast number of ‘newcomer’ TCRs emerge exclusively in the absence of Ctsl. It is difficult to envision a scenario where a Ctsl-dependent accessory molecule selectively orchestrates the ‘disappearance’ of one highly diverse subset of TCRs while simultaneously promoting the emergence of another. That said, perhaps two Ctsl-controlled accessory molecules could, if they operated in opposite directions.

Applying Occam's razor, we feel that the accumulated evidence – including insights from our repertoire analyses – supports the conclusion that perturbations in the CD4 T cell compartment stem from (yet unknown) alterations in the pMHCII ligandome. We respectfully ask for consideration that scientific discussion becomes impractical if every reasonable deduction must be accompanied by formal caveats.

3. Equally important point highlighted in this manuscript is the conclusion that the thymic positive selection contributes to the optimization of T cell functionality (and CD5 expression level in T cells). This

was also reported previously and extensively (Mandl, et al. *Immunity* 2013; Fulton et al., *Nat Immunol.* 2015; Takada, et al. *Nat Immunol* 2015). The link between thymic CTSL and T cell function appears novel, but the authors should give a careful attention to what was already reported in previous articles. We refer to these previous findings on the 1st and 2nd page of the introduction, respectively, and revisit them throughout our discussion – particularly in highlighting commonalities between our novel findings on Ctsl's role in CD4 T cell selection and the well-established role of $\beta 5t$ in CD8 T cell selection. We certainly do not claim to be the first to describe functional tuning: 'There is growing evidence that positive selection not only determines whether a given T cell clone survives but also imprints its intrinsic functionality. A well-characterized example....' (2nd paragraph of the introduction).

4. CTSL is expressed highly in cTEC but is also expressed in a variety of cells including mTEC and DC (as shown in many studies including ImmGen database). Experimental approach in this study exclusively relies on Foxn1-Cre-dependent floxed CTSL-deletion, which affects both cTEC and mTEC. Even though some data are shown to examine the involvement of MHC-hi subpopulation or Aire+ subpopulation of mTEC (Fig. 1d, f, h), mTEC are highly diverse. It is possible that CTSL expressed in mTEC, including functionally mature mTEC-lo subpopulations, may play a crucial role in the regulation of CD4 T cell selection. Indeed, it was previously shown that CD69+ DP thymocytes include CCR7+ cells, which predominantly localize in the medulla (e.g., Kimura, et al. *Nat Immunol.* 2016), suggesting that the reduction in CD69+ DP thymocytes shown in Fig. 3b may be due to the medullary negative selection. Consequently, data presented in this manuscript does not readily allow to draw the conclusion that CTSL in cTEC regulates positive selection of cortical DP thymocytes.

We acknowledge that some influence of medullary 'negatively selecting events' is difficult to formally rule out. However, accumulated evidence ought to be considered as a whole, and 'formal possibilities' weighed against the available data to identify the most likely scenario. In light of this, we have incorporated a passage into the revised discussion to summarize the evidence in favor of a genuine bottleneck in positive selection (p. 19; 'While we cannot formally exclude that negative selection accounts for some of the TCR loss,....').

Accumulated evidence – each piece with its own caveats – supporting that the diminished CD4SP compartment in *Ctsl*^{ΔTEC} mice primarily reflects a genuine bottleneck in positive selection arises from the following observations (assuming that only cTECs mediate positive selection and that this is a cortical event):

- 1) Protein and catalytic activity (active site labeling) are detected exclusively in cTECs (Fig. S1).
- 2) Significant alterations in the cTEC ligandome are observed with various antibodies targeting distinct pMHC combinations, with no corresponding changes in mTEC^{hi} or mTEC^{lo} cells (Fig. 1d-g and Ext. Fig. 1h,i).
- 3) There is no rebound of the CD4 compartment upon ablation of MHCII in hematopoietic APCs and knockdown of MHCII in mTEC^{hi}. Notably, mTEC^{lo} inherently express far less MHCII (Fig. 1h,i and Ext. Fig. 1j,k). Note the distinction from Ref. 34!
- 4) Multiple TCR transgenic clones, which we believe represent a significant fraction of 'lost' TCRs in the polyclonal repertoire, do not give rise to CD4SP cells at all (Fig. 2). In all of these cases, these clones appear 'stuck' at the DP stage, 'frustratedly' auditioning for positive selection.
- 5) There is a reduced fraction of 'signaled' TCR β ^{lo}CD69⁺ cells in MHC^{-/-} *Ctsl*^{ΔTEC} mice (now Fig. 2f, formerly 3b). Concerning the proposal that this observation reflects medullary negative selection: there is convincing recent evidence that CCR7+ CD69⁺DP cells are CD8 lineage intermediates; for instance, these cells are extremely sparse in MHC1^{-/-} mice. The 'true' CD4-lineage TCR β ^{lo}CD69⁺ DP cells are CCR7-, hence (by inference) located in the cortex (PMID: 37266571).
- 6) Lower Nur77 expression in 'functionally Ctsl-dependent' clones is detectable as early as the CD69⁺ DP stage, indicating reduced acute or very recent signaling during positive selection (see 5).
- 7) Last not least: how would disproportionately high negative selection explain the emergence of 'newcomers'?

In the respective passage in the discussion, we acknowledge that 'a substantial body of evidence supports that $\beta 5t$'s essential role in CD8⁺ T cell selection similarly – but with even clearer mechanistic support – primarily reflects a requirement in bona fide positive selection' (p. 19) (Refs. 23 and 25).

5. Data shown in Fig. 1g and h support that the loss of MHC-II in either bone marrow-derived cells or Aire+ mTEC fails to rescue CD4SP cells in TEC-specific CTSL-deficient mice, suggesting that CTSL regulation of CD4 T cell development is independent of the negative selection by bone marrow-derived cells or Aire+ mTEC subpopulation. However, it is appreciated that both bone marrow-derived cells and mTEC are potent in inducing the negative selection. The contribution of the negative selection should be examined by analyzing whether CD4SP cells are rescued by the lack of MHC-II in BOTH bone marrow-derived cells AND mTEC (not only in Aire+ mTEC subpopulation).

This point is closely related to our response to point 4. We acknowledge that, despite all the evidence in favor of a bottleneck in positive selection, the formal possibility of additional contributing factors cannot be entirely excluded. Furthermore, we fully agree that the suggested experimental approach presents an elegant way to test whether ‘excessive susceptibility to negative selection’ – redundantly imposed by DCs or mTECs – plays a role.

However, we respectfully ask for consideration of whether these experiments are truly essential to support the conclusions of our work. Addressing this question would require multiple generations of mouse-breeding to obtain complex compound genotypes (e.g., *Ctsl*^{ΔTEC} mice lacking a medulla, etc.), a significant undertaking. Notably, resolving a similar question in the β5t field was the focus of an entire recent study (Ref 23). Expecting us to recapitulate over fifteen years of groundbreaking β5t research within a single study sets an exceptionally high bar.

6. It is interesting and curious that the loss of CTSL in TEC equally and severely affects CD4 T cell development in all the seven kinds of TCR transgenic T cells, which show different levels of CD5 (Fig. 2). These data do not fit the conclusion that CTSL preferentially promotes the positive selection of CD5-hi high-affinity CD4 T cells. Instead, the results may support the possibility that CTSL regulates CD4 T cell development independent of MHC-II-bound peptide or independent of TCR affinity to the positive selection ligands.

This point is well taken. We already elaborated on accumulated evidence against ‘the possibility that CTSL regulates CD4 T cell development independent of MHC-II-bound peptide’ (i.e., in favor of the peptide scenario) in the previous points – most notably, in our view, through the TCR-selectivity observed in the clonal loss and the new emergence of clones.

The second scenario is an intriguing question. These remarks together with a related comment by Rev #2, prompted us to conduct extensive additional experiments, now presented in **NEW Fig 5a-c**. For brevity, we refer to the corresponding section in the revised manuscript (‘Non-selection in....’; p. 9-10) for an extensive explanation and discussion.

These new TCR sequencing data show at the whole-repertoire level that TCRs at both ends of the ‘natural’ CD5 spectrum are lost, strongly bolstering the conclusion that positively selecting interactions with self-pMHC are ‘weaker across the board’. We believe these additional insights significantly add to the study, and appreciate the reviewer’s input, which motivated us to explore these aspects in greater depth.

7. Thymocyte development in Fig. 1a,b, 2a, 3a, 3e, 5b, 5d, and S3 should be analyzed by evaluating absolute numbers, rather than the frequencies, of thymocyte subpopulations.

We apologize for erroneously omitting this information. Corresponding novel data are included in the revised manuscript (**Extended Fig. 1 a,e; Extended Fig. 2 c; Extended Fig. 3 a,c; Extended Fig. 6 a,b**).

List of new or updated display items

Items related to major new experimentation are highlighted in gray.

Figure		Caption	
1	g	Presentation of MHCII: Ea52-68 on cTECs is maintained, but reduced, in Ctsl Δ TEC mice	NEW
2	c	Normal selection of OT-I CD8 T cells in Ctsl Δ TEC mice	NEW
	d	Normal CD5 on OT-I DP cells in Ctsl Δ TEC mice	NEW
	i	Natural' CD5 levels on various TCR-tg CD4SP cells when normally selected	NEW
3	b	Reduced CD5 expression on CD4SP cells in Fixed- β Ctsl Δ TEC mice	NEW
	c-g	Altered repertoire in Ctsl Δ TEC mice	*
4	a	Diminished size of three antigen-specific CD4 T cell cohorts in Ctsl Δ TEC mice	NEW
5 NEW	a	Sorting strategy 'natural' CD5lo and hi	NEW
	b	Minimal repertoire overlap between 'natural' CD5lo and CD5hi	NEW
	c	TCR loss affects both ends of the 'natural' CD5 spectrum	NEW
	d	5 of the Top Ten responder TCRs are 'lost' from the Ctsl Δ TEC repertoire.	NEW
	e	Contribution of Ctsl-dependent and -independent clones to Top Ten LLO-responders mirrors the overall contribution of these two categories to the total repertoire	NEW
6 (old 5)	a	Shared LLO Tet+ clones between Ctsl Δ TEC and Ctsl+/+ mice	#
7 (old 6)	c	Diminished basal translation in CD4 T cells at all post-selection stages in Ctsl Δ TEC mice	NEW
	e	Diminished PD-1 and BTLA in CD4 T cells at all post-selection stages in Ctsl Δ TEC mice	NEW
	i	High concentrations of IL-7 'rescue' in vitro survival of CD4 T cells selected in Ctsl Δ TEC mice	NEW
Ext. 1 NEW	a	Thymocyte cell numbers	NEW
	b	CD69 vs MHC I on CD4SP cells	NEW
	c	Foxp3+ Treg cells in thymic CD4 SP cells	NEW
	d	Re-immigrants among thymic Foxp3+ Treg cells	NEW
	e	Peripheral CD4 T cell numbers	NEW
	f	CC44 vs CD62L on peripheral CD4 T cells	NEW
	g	Foxp3+ Treg cells in peripheral CD4 T cells	NEW
	h	Unchanged MHCII:CLIP on immature mTECs	NEW
	i	Unchanged MHCII:non-CLIP on immature mTECs	NEW
Ext. 2 NEW	c	Cell numbers related to Fig. 2	NEW
Ext. 3 NEW	a	Cell numbers related to Fig. 3a	NEW
	c	Number of peripheral CD4 T cells in Fixed- β Ctsl Δ TEC	NEW
Ext. 6 NEW	a	Thymocyte cell numbers related to Fig. 6b	NEW
	b	Thymocyte cell numbers related to Fig. 6c	NEW
Ext. 7 NEW	a	Reduced cell size of CD4SP cells in Ctsl Δ TEC mice	NEW
	c	Unchanged pERK upon PMA stimulation in CD4SP cells in Ctsl Δ TEC mice	NEW
	d	CD69 and CD25 in polyclonal M2 thymocytes upon in vitro anti-CD3 and anti-CD28 stimulation.	NEW
	e	Impaired survival of polyclonal CD4 T cells selected in Ctsl Δ TEC mice	NEW
* Reanalyzed with updated TCR annotations and more stringent definition of recurrent TCRs. Conclusions not affected.			
# Updated with new TCR annotations for consistency with updated dataset in Fig. 3 and new Fig. 5. Conclusions not affected.			
Essentially all FACS plots have been updated to meet Nat Immunol formatting guidelines			

We thank the reviewers for the positive and fair assessment of our manuscript and appreciate the time and effort they have invested. We agree with Reviewer #1 that future work must aim to mechanistically link our key findings more 'peptide-specifically' to Ctsl's effect on the pMHCII ligandome. We have previously elaborated on why this remains a formidable challenge and asked for consideration that these aspects far exceed the scope of the present work, as also pointed out by Reviewer #2.

The MS has now been substantially condensed to meet the word count. A version with tracked changes can unfortunately not be provided owing to erroneous acceptance of all changes at some point in the editing process. None of the modifications that have been made to meet the word count affect the conclusions or message of the original MS as seen by the reviewers.

Some changes were made to the order of display items, mostly in order to enhance the flow, but also to avoid redundancies and save space:

Changes in figure items:

- Fig. 1 j and k were formerly Fig. 2 f and h
- Fig. 2 former d has been removed to save space (CD5 on OT-I)
- Fig. 2 former e and g have been moved to Ext Data Fig. 1 (now l and m)
- Fig. 2 former f and h is now Fig. 1 j and k
- Fig. 5 d has been added (NEW) per editorial request to show definition of top-ten responder clones
- Fig. 5 former e has been removed to save space (largely redundant with panel c)
- Fig. 6 order of f and g has been swapped for flow in results section
- Fig. 7 order of g and h has been swapped for flow in results section
- Fig. 7 order of j and k has been swapped for flow in results section
- Fig. 7 former l was split into l and m